# Cerebral organoids display dynamic clonal growth and tunable tissue replenishment

Dominik Lindenhofer [1,2,9,11], Simon Haendeler [2,3,11], Christopher Esk [1,4,11] ✉, Jamie B. Littleboy [1,2,11], Clarisse Brunet Avalos [5], Julia Naas [2,3], Florian G. Pflug [3,10], Eline G. P. van de Ven[1], Daniel Reumann[1], Alexandre D. Baffet [5,6], Arndt von Haeseler [2,7] & Jürgen A. Knoblich [1,8] ✉

During brain development, neural progenitors expand through symmetric divisions before giving rise to differentiating cell types via asymmetric divisions. Transition between those modes varies among individual neural stem cells, resulting in clones of different sizes. Imaging-based lineage tracing allows for lineage analysis at high cellular resolution but systematic approaches to analyse clonal behaviour of entire tissues are currently lacking. Here we implement whole-tissue lineage tracing by genomic DNA barcoding in 3D human cerebral organoids, to show that individual stem cell clones produce progeny on a vastly variable scale. By using stochastic modelling we find that variable lineage sizes arise because a subpopulation of lineages retains symmetrically dividing cells. We show that lineage sizes can adjust to tissue demands after growth perturbation via chemical ablation or genetic restriction of a subset of cells in chimeric organoids. Our data suggest that adaptive plasticity of stem cell populations ensures robustness of development in human brain organoids.

Clonal heterogeneity, the differential distribution of the number of cells derived from individual tissue stem cells, is characteristic for individual species as well as different tissues. The human brain is one of the most expanded organs when compared to rodents in terms of cell numbers, but the underlying lineage relationships accompanied with this expansion are unclear. Labour-intensive imaging-based lineage tracing studies in the mouse brain suggest a variable clonal output of individual stem cells[1–3]. In human brain only limited data of clonal output heterogeneity are available, suggesting potentially expanded clone size dispersion compared to mouse[3–6]. These pioneering studies rely on whole-genome sequencing of individual neurons from post-mortem brains and back-tracking lineage relationships based on single nucleotide polymorphisms accumulated during a donor's brain development. These approaches are time- and cost-intensive and have so far been limited to the study of several thousand cells. Human cerebral organoids have emerged as a 3D self-organizing model system recapitulating key aspects of human brain development in a tissue context[7–13]. They recapitulate the initial expansion of neural stem cells by symmetric divisions before producing non-cycling, differentiating neuronal progeny in an asymmetric division mode[14,15]. Functionally, cerebral organoids resemble human brain development regarding morphology, cell type composition, gene expression, neuronal migration behaviour and electrical activity[16–24]. Here we use cerebral organoids to assess clonal heterogeneity using a population-based whole-tissue lineage tracing approach.

[1]Institute of Molecular Biotechnology of the Austrian Academy of Science, Vienna BioCenter, Vienna, Austria. [2]Vienna Biocenter PhD Program, University of Vienna and the Medical University of Vienna, Vienna, Austria. [3]Center of Integrative Bioinformatics Vienna, Max Perutz Labs, University of Vienna and Medical University of Vienna, Vienna BioCenter, Vienna, Austria. [4]Institute of Molecular Biology, University of Innsbruck, Innsbruck, Austria. [5]Institut Curie, PSL Research University, CNRS UMR144, Paris, France. [6]Institut national de la santé et de la recherche médicale, Paris, France. [7]Faculty of Computer Science, Bioinformatics and Computational Biology, University of Vienna, Vienna, Austria. [8]Department of Neurology, Medical University of Vienna, Vienna, Austria. [9]Present address: Genome Biology Unit, European Molecular Biology Laboratory, Heidelberg, Germany. [10]Present address: Biological Complexity Unit, Okinawa Institute of Science and Technology Graduate University, Onna, Okinawa, Japan. [11]These authors contributed equally: Dominik Lindenhofer, Simon Haendeler, Christopher Esk, Jamie B. Littleboy. ✉e-mail: christopher.esk@uibk.ac.at; juergen.knoblich@imba.oeaw.ac.at

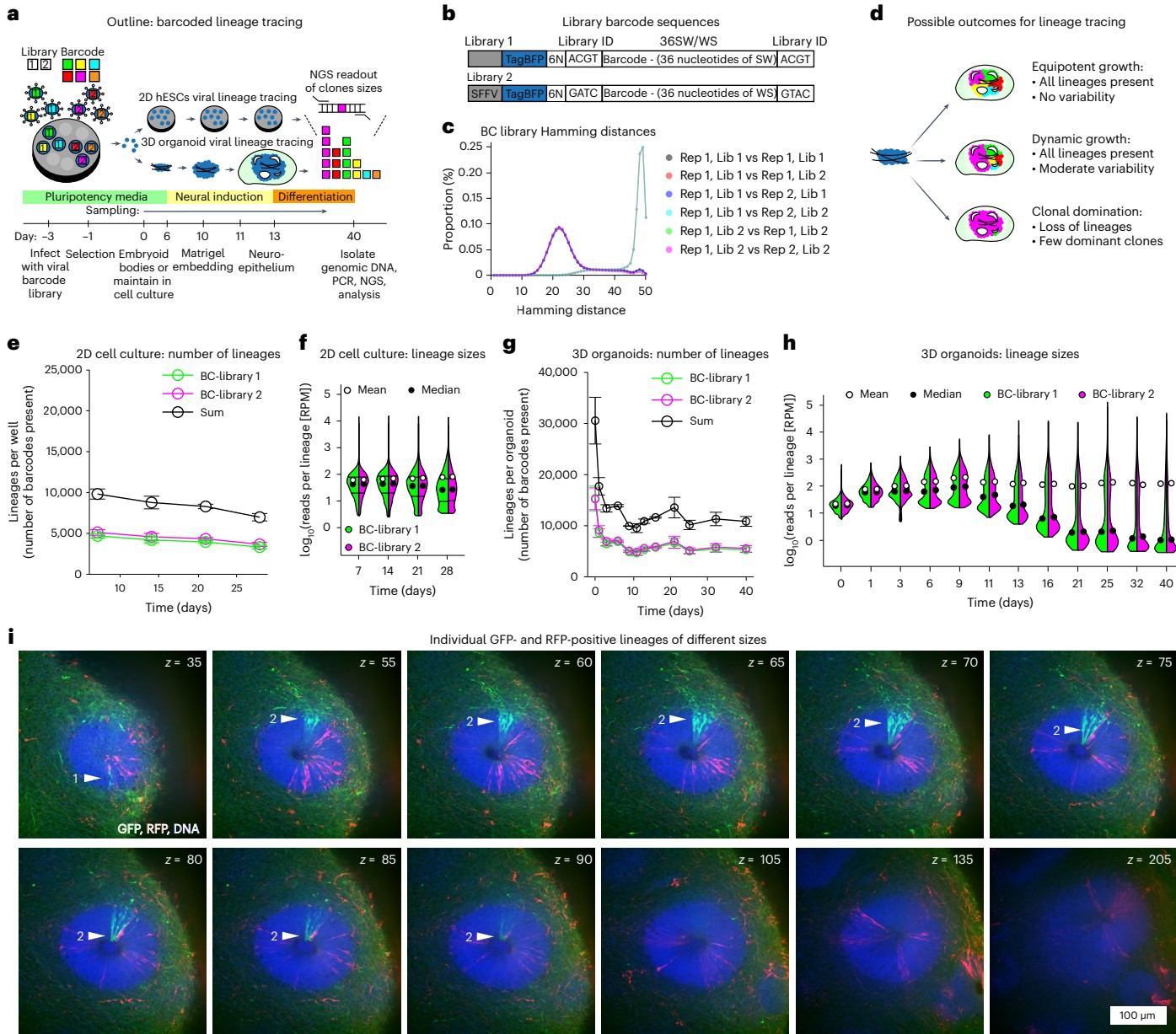

**Fig. 1 | Whole organoid lineage tracing. a**, Outline of whole-tissue lineage tracing experiment. NGS, next generation sequencing. **b**, Schematic of barcodes used for lineage tracing. Two barcode libraries with distinct semi-random barcode sequences that are distinguished by library identifiers. SFFV, spleen focus-forming virus. **c**, Hamming distances between barcode libraries within replicates and inter-replicates at day 0. **d**, Possible outcomes of lineage tracing experiment with characteristics for equipotent, dynamic and clonal domination growth modes. **e**, Number of measured barcodes in 2D lineage tracing ($n = 3$ biological replicates from 1 experiment at each time point). Data points represent mean ± s.e.m. **f**, Lineage size distributions in 2D. Normalized reads per lineage (in reads per million, RPM) for each time point. ($n$ = individual lineages from 3 biological replicates from 1 experiment at each time point). **g**, Number of measured barcodes in 3D lineage tracing ($n = 3$ organoids as biological replicates from 1 differentiation experiment at each time point). Data points represent mean ± s.e.m. **h**, Lineage size distributions in 3D. Normalized reads per lineage (RPM) for each time point ($n$ = individual lineages from 3 organoids as biological replicates from 1 differentiation experiment at each time point). **i**, Sparsely labelled organoids containing 98% WT, 1% GFP-WT and 1% RFP-WT cells were cleared and imaged at day 40. Z-series shows a neural rosette containing two GFP-WT clones of small (1) and large sizes (2).

## Results

### Whole-tissue barcoded lineage tracing

To assess clonal growth behaviour in human cerebral organoids we grew organoids in which each of the 24,000 starting cells had been labelled with a unique, functionally inert and genomically integrated DNA barcode (Fig. 1a)[25]. H9 human embryonic stem cells (hESCs) were infected with a mix of two independently cloned highly variable retroviral DNA barcode libraries (BC-library 1 and 2; Fig. 1b) for internal control and sorted by flow cytometry ensuring that each starting cell of an organoid contained a barcode. Infection rates were kept low resulting in predominantly unique barcode integrations, while comparing Hamming distances of replicates at day 0 confirmed a highly variable and complex barcode library (Fig. 1c)[25]. Whole organoids were sampled at various timepoints by direct lysis in DNA extraction buffer and their genomic DNA isolated. DNA barcodes were PCR amplified and sequenced to determine lineage sizes of each individual remaining starting cell within the tissue. There are three potential modes for clonal growth in tissues (Fig. 1d): (1) equipotent growth—all starting cells produce equal progeny; (2) dynamic growth—all starting cells contribute progeny albeit at differing levels; and (3) clonal domination—few starting cells contribute

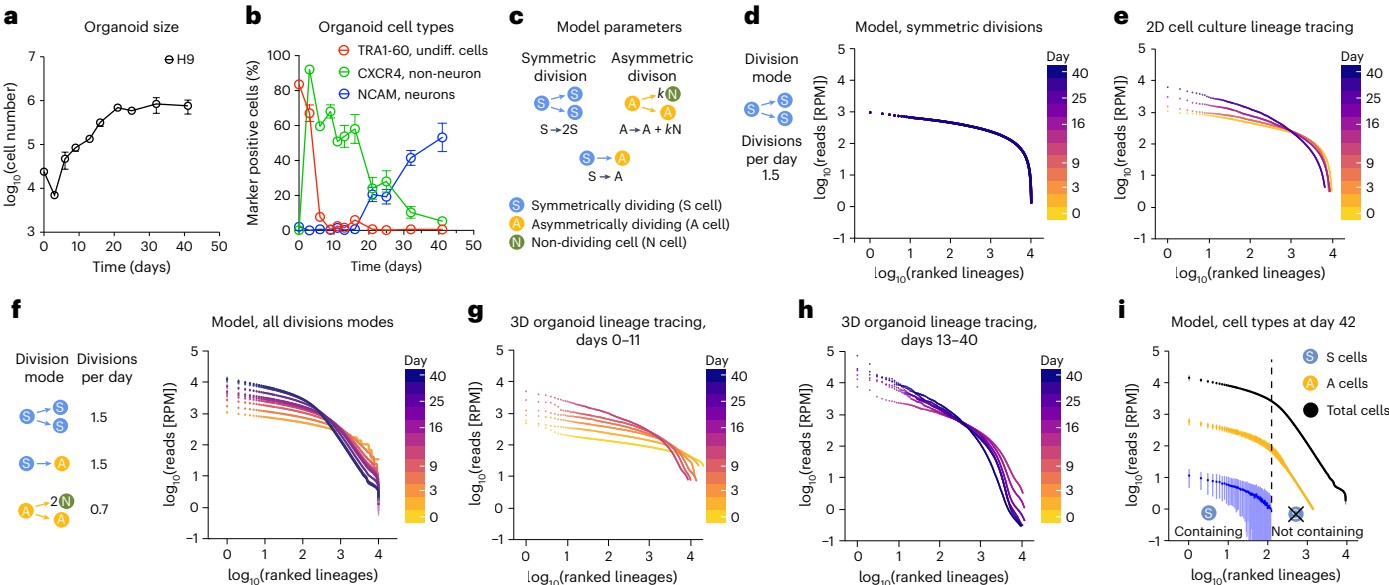

**Fig. 2 | Characterization of cerebral organoid growth. a,** Organoid sizes by cell numbers over time (*n* = 3 organoids as biological replicates from 1 differentiation experiment at each time point). Data points represent mean ± s.e.m. **b,** Organoid cell type composition over time. Cell types (undifferentiated, non-neuronal and neurons) and markers used as indicated (*n* = 9 organoids as biological replicates from 3 differentiation experiments at each time point). Data points represent mean ± s.e.m. **c,** Possible cell division behaviours used to model lineage distribution in organoid growth. Rates in events per day were subsequently modelled to fit organoid lineage read distribution and number. **d,** Model of lineage size distribution encompassing only symmetric division over time leading to generation of S cells. **e,** 2D stem cell lineage tracing over time. **f,** Model of lineage distribution encompassing symmetric, S-to-A transitions and asymmetric divisions leading to generation of S, A and N cells over time. Rates for S-cell division and S-to-A transition are equal. **g,h,** 3D organoid lineage tracing (days 0–11 (**g**); days 13–40 (**h**)). **i,** Cell-type composition resulting from lineage size model encompassing S, A and N cells. Lineages still containing and not containing S cells are separated by a dashed line.

the vast majority of progeny. Analysis of control cultures grown in 2D showed equipotent growth as neither the number of individual lineages (number of barcodes) detected nor their relative contribution to cultures varied, in line with previous data from 2D grown cancer cells[26] (Fig. 1e,f and Extended Data Fig. 1a). In 3D, cerebral organoid lineages were lost during embryoid body (EB) formation, representing a selective pressure during this in vitro process. However, similar to 2D cells, from day 6 the number of lineages in the growing organoid remains constant, indicating that cells that successfully integrate in an EB continue to contribute to the organoid (Fig. 1g). In contrast to 2D cultures, 3D organoids showed an increasingly differential lineage size distribution with a range over five orders of magnitude at day 40 resulting in 80% of progeny originating from only 5% of lineages. This indicates substantial clone size heterogeneity and dynamic growth in organoids (Fig. 1h and Extended Data Fig. 1b–d). Triplicate measurements showed high inter-organoid reproducibility, while the two independent retroviral libraries acted nearly identically, indicating high intra-organoid reproducibility (Extended Data Fig. 1a,b). Variable clone size output in organoids was confirmed by imaging green fluorescent protein (GFP)-expressing lineages in sparsely labelled organoids (Fig. 1i).

We sought to verify the whole organoid lineage tracing method to accurately measure lineage sizes. We used both an alternative technique in cerebral organoids and the same technique in a well-studied model system, the mouse cortex. Firstly, in silico subsampling of reads of random lineages from the measured whole organoid lineage tracing data predicts that a subset of 1–2% of starting cells recapitulates whole-organoid growth with high accuracy indicated by the small errors within the 90% confidence range of subsampled lineages (Extended Data Fig. 2a). To test this prediction, we mixed six fluorescently labelled cell lines into growing organoids at 1–2% ratios and followed their maintenance over time by flow cytometry (Extended Data Fig. 2b–d). Indeed, labelled cell lines were maintained with a variability comparable to the 90% confidence interval from in silico subsampling. Secondly, we used our sequencing-based lineage tracing approach in the well-studied

mouse cortex and compared it to published low-throughput and high-accuracy imaging-based data[1,2]. We injected DNA barcoded retrovirus at E9.5, E10.5 and E13.5 in the ventricle of developing mice, infecting progenitors and analysed at E18.5 (Extended Data Fig. 2e–g). Our high-throughput data consisting of ten-thousands of lineages confirms the published imaging-based range of clonal heterogeneity in mice, ranging around two orders of magnitude from beginning to the end of neurogenesis, and validates our sequencing-based lineage tracing approach[1,2,27]. This data highlights smaller clonal heterogeneity in mouse compared to human cortex, establishing a clear species difference. To test if the observed clonal heterogeneity is an effect specific to tissue development or a consequence of stem cell differentiation, we repeated the cerebral organoid tissue lineage tracing experiment and compared it to a 2D neuronal differentiation protocol and symmetrically dividing 2D hESCs (Extended Data Fig. 3a–g). Neurons differentiated in 2D showed increased clonal heterogeneity compared to symmetrically dividing 2D hESCs but did not reach the highly divergent distribution observed in 3D cerebral organoids. This shows that simultaneous proliferation and differentiation are key factors in driving lineage size heterogeneity. Increased lineage size heterogeneity in 3D cerebral organoids might be attributed to an extended presence of proliferative neural rosettes in a tissue context. Here, we established sequencing-based whole-tissue lineage analysis demonstrating that lineage size distribution of human cerebral organoids follows a dynamic growth mode and differs from distributions found in mouse, 2D differentiated neurons and symmetrically dividing hESCs.

## A model for organoid growth based on lineage populations

We noted that lineage size distribution increasingly diverged from day 11 onwards in cerebral organoids indicated by an increased difference of median and mean, increased relative contribution of the top percentile and the cumulative lineage contribution to the total organoid (Fig. 1h and Extended Data Fig. 1c,d). This timeframe coincides with a decrease in organoid growth (Fig. 2a). Immunofluorescence of cerebral organoids

at day 11 showed that apical–basal polarity is inverted after matrigel embedding generating ventricular-like progenitor zones that correspond to neural epithelium (Extended Data Fig. 4a,b). Neurogenesis in cerebral organoids starts around day 16 (Fig. 2b and Extended Data Fig. 4c). Thus, the increasingly divergent lineage size distribution from day 11 onwards coincides with progenitors switching from a symmetric to an asymmetric division mode and the generation of neurons or non-dividing cells. As whole organoid lineage tracing does not provide information on cell division behaviour, we tested in silico for different cell division modes and resulting lineage size distributions. We devised a stochastic model in which cell division behaviour and fate transitions within lineages are represented as discrete events over time. These include symmetrically dividing (S) cells that can switch to asymmetrically dividing (A) cells that self-renew and generate non-dividing (N) cells (Fig. 2c). The factor $k$ represents A cells generating intermediate progenitor cells (IPs) that divide a finite number of times to generate $k$N cells. Parameters were varied to model 10,000 lineages separately resulting in simulated lineage sizes including these cell types. We explored two simplified stochastic growth models: (1) a symmetric division model containing exclusively S cells and (2) a combined model containing S, A and N cells and their respective division modes. The symmetric division model showed limited clonal heterogeneity, which resembles the lineage size distribution of 2D stem cells (Fig. 2d,e). The combined model showed large clonal heterogeneity, particularly of small lineages (size rank 300–10,000) growing to diverse lineage sizes while large lineages (rank 1–300) displayed more homogeneous lineage sizes (Fig. 2f). The 3D organoid lineage tracing data up to day 11 resemble the symmetric division model, whereas later stages of the organoid data are increasingly similar to the combined model that includes A and N cells (Fig. 2g,h). This distribution was only observed when the S cell division rate was equal to the S to A differentiation rate, but not if either of them was bigger or smaller (Extended Data Fig. 5). In simulations where the S cell division rate was higher than S to A differentiation, S cells dominated, and the model resembled the symmetric division model. On the contrary, if S cell division rate was lower than S to A differentiation, S cells were depleted and lineage sizes more uniform. Varying IP divisions, modelled by varying the factor $k$ from $k=1$ (direct neurogenesis) up to $k=8$ (neurogenesis via seven IP divisions), did not substantially change lineage size distributions (Extended Data Fig. 5). Similarly, direct neurogenesis of S cells into 2N cells did not change the requirement of equal rates of S cell division and S to A transition rates to fit the measured data (Extended Data Fig. 5). In the combined model most similar to day 40 organoids, two results emerge: (1) the largest lineages in a growing organoid contain cells that divide symmetrically and (2) lineage size heterogeneity is driven by lineages losing symmetrically dividing cells, while lineages that contain symmetrically dividing cells grow homogeneously in size. This prolonged presence of symmetrically dividing neural stem cells well into neurogenesis marks a clear difference of human to murine development[1,2].

## Prolonged presence of symmetrically dividing cells

We sought to verify our model predictions. To this end, we wanted to survey the cell type composition in individual lineages of organoids by barcoded scRNA-seq (single-cell RNA sequencing). We generated organoids that contained 3–5% lineage-barcoded cells labelled with GFP (Fig. 3a and Extended Data Fig. 6a,b), a proportion that faithfully represents the total lineage size population in organoids (Extended Data Fig. 2a–d). To dissect cell type composition within lineages we sorted the barcoded cells enriching for individual lineage profiles and performed scRNA-seq. Cells were harvested at the onset of neurogenesis (day 18) and the latest timepoint surveyed (day 42). On average, 65% of 8,111–15,833 cells per organoid could be associated with a barcode resulting in 140–175 lineages per organoid. We identified cycling progenitors as well as non-dividing cells such as neurons (Fig. 3b–d). Importantly, to our knowledge, symmetrically and asymmetrically

dividing progenitors cannot be distinguished based on gene expression. To directly compare model predictions, we plotted the barcoded scRNA-seq data into ranked lineage size plots and indicated their cell type composition (Fig. 3e,f and Extended Data Fig. 6c–i). Lineage size distribution was similar to whole organoid lineage tracing data with the range of lineage sizes about tenfold lower in day 18 organoids compared with day 42 organoids. As expected, day 18 organoids contained the first neurons, while the majority of cells were progenitors. At day 42 the lineage size rank distribution appeared similar to the measured lineage size distribution from the whole tissue lineage tracing and our model predictions (Fig. 2i). Individual lineage composition regarding progenitors and neurons could be qualitatively recapitulated. Large lineages contained both progenitors and neurons (Fig. 3b–f), rather than exclusively progenitor cells at the expense of neurons. Upon close examination of the data we find overall larger numbers of progenitors as predicted by the sum of S and A cells in the model. This may be due to our model underestimating the number of S and A cells in organoids or an scRNA-seq sample preparation bias towards progenitors at the expense of neurons. Overall, the barcoded scRNA-seq lineage tracing data are consistent with our model predicting the existence of S cells in organoids at least 25 days after the onset of neurogenesis.

To directly identify symmetrically dividing cells in day 42 organoids we turned to microscopy. Indeed, we found examples of symmetrically dividing cells in day 42 organoids based on BrdU-labelled progenitors with a spindle orientation perpendicular to the apical side of the ventricle (Fig. 3g,h and Extended Data Fig. 6j,k). Furthermore, fate mapping of dividing cells in ventricular-like zones in live slice cultures of organoids identified SOX2+ radial glia dividing symmetrically to generate two SOX2+/EOMES−/NEUROD2− cells (Fig. 3i–m and Extended Data Fig. 6l), a result mirrored in older cerebral organoids and human brain slice cultures[28]. This is consistent with the division behaviours of S cells in our model. Taken together, our whole organoid lineage tracing approach allows for size characterization of all lineages within developing brain organoids and reveals large lineage size heterogeneity. This heterogeneity is driven by symmetrically dividing cells that remain in a subset of organoid lineages during organoid development as predicted by our stochastic growth model and confirmed by barcoded scRNA-seq and microscopy.

## Tissue-dependent lineage replenishment

Our data and modelling strongly suggest that a symmetrically dividing cell population is maintained for an extended period of time after the onset of neurogenesis in cerebral organoids. In some adult tissues, symmetrically dividing stem cells allow for adaptive turnover of cells depending on tissue requirements[29,30]. We hypothesized that also in organoid development, symmetrically dividing stem cells could modulate their cellular output depending on tissue needs and possibly replenish depleted lineages and compensate for their loss during organoid growth. To test this, we grew chimeric organoids from wild-type (WT) cells (100, 99, 95, 90, 50, 0%) mixed with increasing amounts of GFP-labelled puromycin-resistant cells (0, 1, 5, 10, 50, 100%; Fig. 4a)[16]. Upon puromycin treatment from day 11, WT cells get selectively depleted, while GFP-labelled puromycin-resistant cells continue to grow. Chimeric organoids without puromycin treatment served as control. We measured both organoid cell number and relative contribution of the puromycin-resistant GFP-positive cells over time (Fig. 4b,c). Organoid populations consisting of either 100% WT or 100% GFP-puromycin-resistant cells grew as expected, with growth inhibition and organoid disruption upon puromycin treatment only in 100% WT organoids. Chimeras containing 1% GFP-puromycin-resistant cells were insufficient to maintain organoid integrity upon puromycin treatment, whereas 5% of GFP-puromycin-resistant cells maintained organoid growth with a majority of GFP-positive cells, albeit at lower overall organoid size. By contrast, chimeras containing 10% GFP-puromycin-resistant cells resulted in near complete rescue of

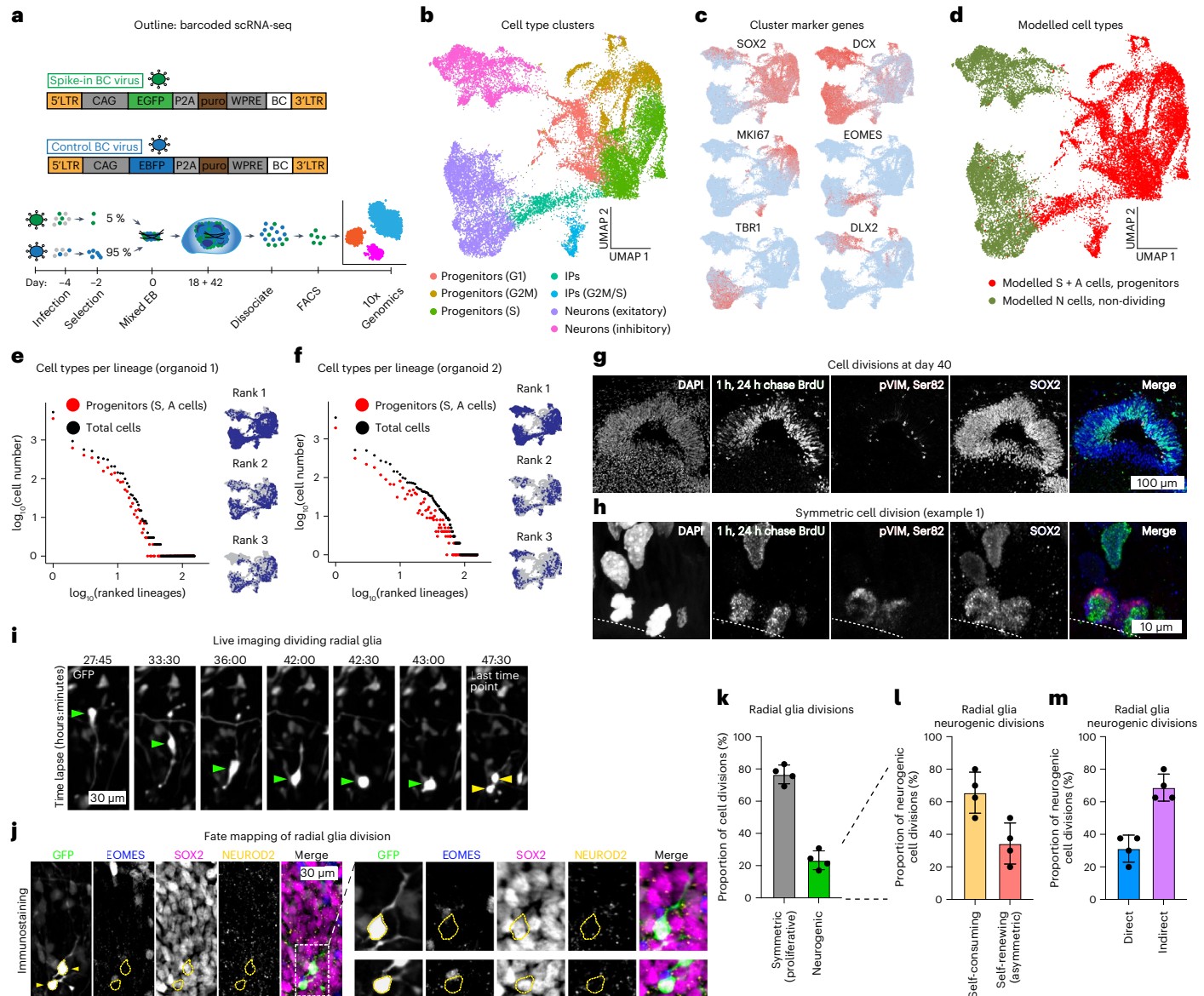

**Fig. 3 | Prolonged symmetric cell divisions in organoids. a**, Scheme of barcoded lineage tracing scRNA-seq experiment. **b–d**, UMAPs of day 42 organoid cells in scRNA-seq indicating cell types (**b**), select markers for cell type annotation (**c**) and cell classifications in S + A and N cells (**d**) (*n* = 2 organoids as biological replicates from 1 differentiation experiment). **e,f**, Lineage size rank plots for day 42 organoids 1 (**e**) and 2 (**f**) from barcoded scRNA-seq. Insets show cell distribution of the three largest lineages per organoid overlaid on UMAP. **g,h**, Immunofluorescent characterization of cerebral organoids at day 40 for nuclei (DAPI), incorporated BrdU (1 h pulse 24 h prior to fixation), dividing cells (pVIM, Ser 82) and neural stem cells (SOX2). Overview (**g**) and individual example 1 (**h**). For additional examples see Extended Data Fig. 6. **i,j**, Cell fate mapping of dividing radial glia. Example of a dividing cell at the ventricular surface of day 42 organoids (**i**) followed by staining for markers of radial glia (SOX2), IPs (EOMES) and neurons (NEUROD2) (**j**). **k–m**, Quantification of cell divisions analysed by division type. Symmetric proliferative and neurogenic divisions (**k**), subset of neurogenic divisions into self-consuming and self-renewing divisions (**l**) and subset self-renewing neurogenic divisions into direct and indirect divisions (**m**) (*n* = 195 cell divisions from 4 organoids as biological replicates from 2 differentiation experiments). Data points represent mean ± s.d.

organoid sizes upon treatment, while 50% puromycin-resistant cells fully rescued WT cell loss. In all chimeric organoids without puromycin treatment, GFP-puromycin-resistant cells grew similarly to WT cells, indicating that these cells do not have an intrinsic growth advantage. This suggests that GFP-puromycin-resistant cells can tune their cellular output under puromycin treatment depending on tissue demand. Successful replenishment can be achieved with around 10% of resistant cells, revealing a substantial replenishment capacity in a tissue context. Besides organoid size, compensation occurred also qualitatively as progenitors and neurons were similarly arranged in puromycin-treated and control organoids and cell fate marker expression was highly similar (Extended Data Fig. 7a–g). Next, we asked whether all resistant cells or only a subset contribute to this replenishment effect. To test this, we performed

whole organoid lineage tracing using low chimeric organoid mixtures near maximum replenishment (10% and 20% GFP-puromycin-resistant cells), labelling WT and GFP-puromycin-resistant cells with independent DNA barcode libraries. GFP-puromycin-resistant cells expanded upon puromycin treatment leading to a population wide shift over time such that both median and mean lineage sizes were enlarged by an entire order of magnitude over WT cells (Fig. 4d–f and Extended Data Fig. 7h–k). To quantify changes in lineage size distributions on a population level we applied a statistical framework termed optimal transport theory[31,32] (Extended Data Fig. 7l). This allows for the comparison of distributions such that the change between them is minimized. Here, we apply optimal transport calculations to define the minimal lineage size change of GFP-puromycin-resistant lineages to fully compensate

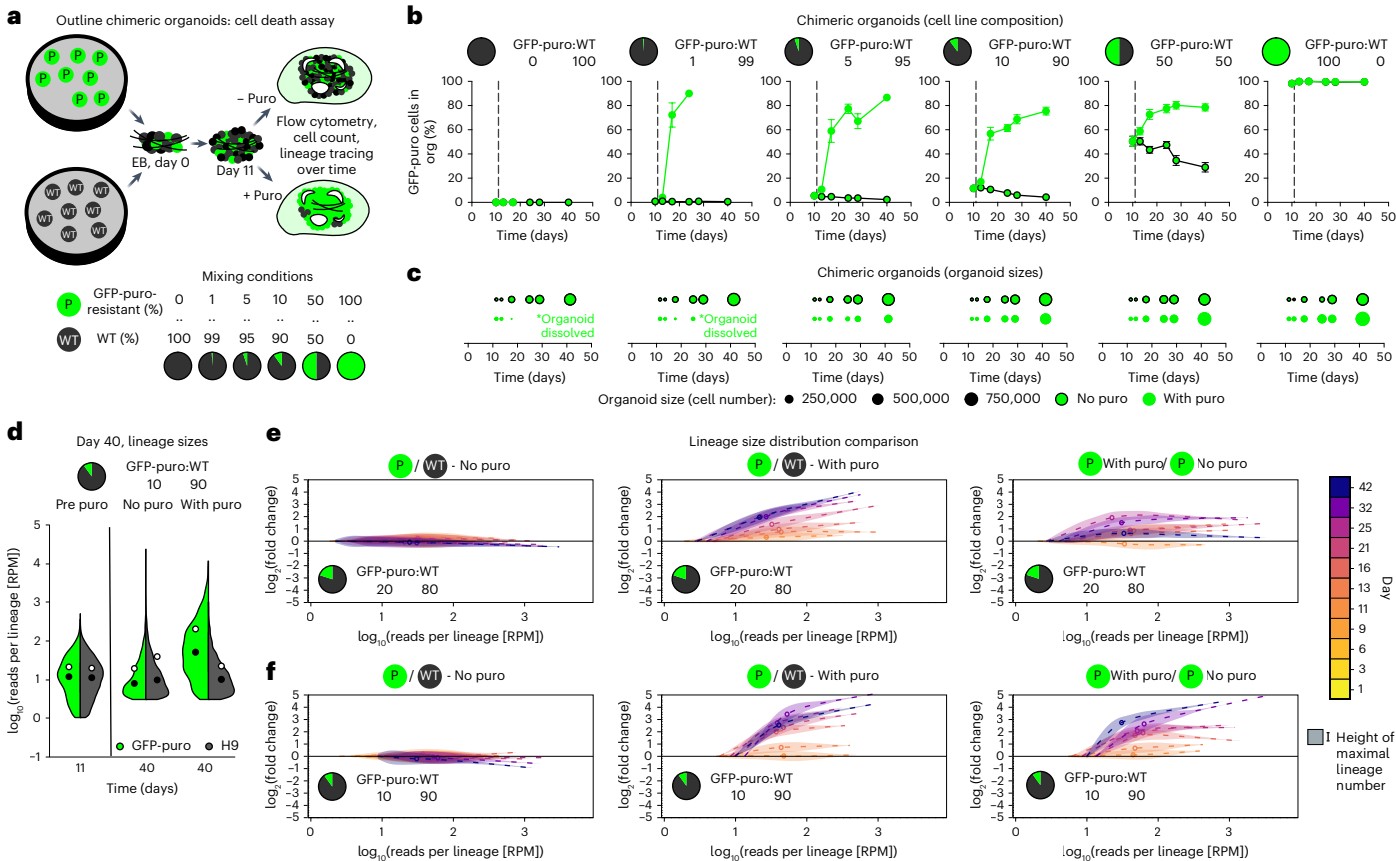

**Fig. 4 | WT to cell death competition assays in cerebral organoids. a**, Scheme of experimental setup of chimeric organoid formation and readout containing WT cells and GFP-labelled puromycin-resistant cells of different mixtures. **b,c**, GFP-puromycin-resistant cell percentages (**b**) and overall cell numbers (**c**) of chimeric organoids of the indicated composition with and without puromycin treatments over time (*n* = 6 organoids as biological replicates from 2 differentiation experiments at each time point for each ± puromycin treatment). Data points in **b** represent mean ± s.e.m.; area in **c** represents relative organoid size by cell number. **d**, Whole organoid lineage tracing of WT (90%) and GFP-puromycin-resistant (10%) cells. Lineage size distribution in conditions indicated on the top (*n* = individual lineages from 3 biological replicates from 1 experiment

for each ± puromycin treatment). **e,f**, Optimal transport comparison of lineage populations of WT and GFP-puromycin-resistant cells in chimeric organoids containing WT:GFP-puro in 80:20 (**e**) and 90:10 (**f**) ratios. Left: GFP-puromycin-resistant to WT comparison in the absence of puromycin. Middle: GFP-puromycin-resistant to WT comparison in the presence of puromycin from day 11 onwards. Right: GFP-puromycin-resistant cells in chimeric organoids treated with puromycin in comparison to the same GFP-puromycin-resistant population in chimeric untreated organoids. Thickness of curves indicates number of lineages of certain sizes, dashed line indicates mean (*n* = individual lineages from 3 biological replicates from 1 experiment at each time point ± puromycin treatment).

for perturbed WT lineages in the presence of puromycin. Comparing WT and GFP-puromycin-resistant populations without puromycin treatment indicated no changes at any timepoint, whereas GFP-puromycin-resistant lineages of all sizes outgrew their WT counterparts in the presence of puromycin (Fig. 4e,f). We observed increasing replenishment over time and a greater replenishment effect size in chimeras containing 10% GFP-puromycin-resistant cells compared with 20%. Importantly, optimal transport analysis also allows for cross comparison between lineage populations of the same cell line across experimental conditions. We find that the exact same population of GFP-puromycin-resistant lineages increases their clonal output dependent on the growth-inhibition of surrounding WT cells. This indicates that the heterogeneous clonal output of neural stem cells is not predetermined but plastic and responsive to the state of surrounding tissue to generate normal-sized organoid tissue, efficiently buffering growth defects.

**Replenishment of genetic defects**

Next, we asked whether replenishment in a tissue also occurs when subsets of cells have growth or fate alterations. During development this can be caused by genetic defects occurring randomly in individual cells posing a threat to normal development. We tested the ability of RFP-expressing WT cells (RFP-WT) to compensate for

GFP-expressing cells with homozygous loss-of-function (GFP-KO) mutations in genes crucial for brain development in equally mixed organoid chimeras (Fig. 5a and Extended Data Fig. 8a–f). Genes knocked out in GFP-positive cells included *TP53* overcoming initial cell stress-induced apoptosis in EB formation[33], the microcephaly gene *ASPM* modelling a consistently weak proliferation defect[34] and the dorsal forebrain specification-determining factor *PAX6*[35–38]. All chimeric organoids showed the expected changes in RFP-WT and GFP-KO cell contribution to organoids (Fig. 5b) but no defects in organoid cell numbers, suggesting full compensation by RFP-WT cells independent of the genetic ablation (Fig. 5c). Immunohistochemistry and whole organoid lineage tracing with optimal transport analysis enabled comparison of RFP-WT cells versus GFP-KO cells. Control organoids showed consistent contribution of both GFP- and RFP-WT cells to progenitors and neurons with no difference in lineage size distribution between the two control lines (Fig. 5d,e and Extended Data Figs. 8g and 9a,e,i). The few RFP-WT cells in GFP-KO-TP53 chimeras surviving EB formation kept a relatively consistent contribution to both progenitors and neurons. Accordingly, optimal transport analysis indicated differential lineage populations already early in the protocol with the differences remaining relatively stable (Fig. 5b–e and Extended Data Fig. 9b,f,i). GFP-ASPM-KO cells in chimeric organoids showed consistently slower growth with

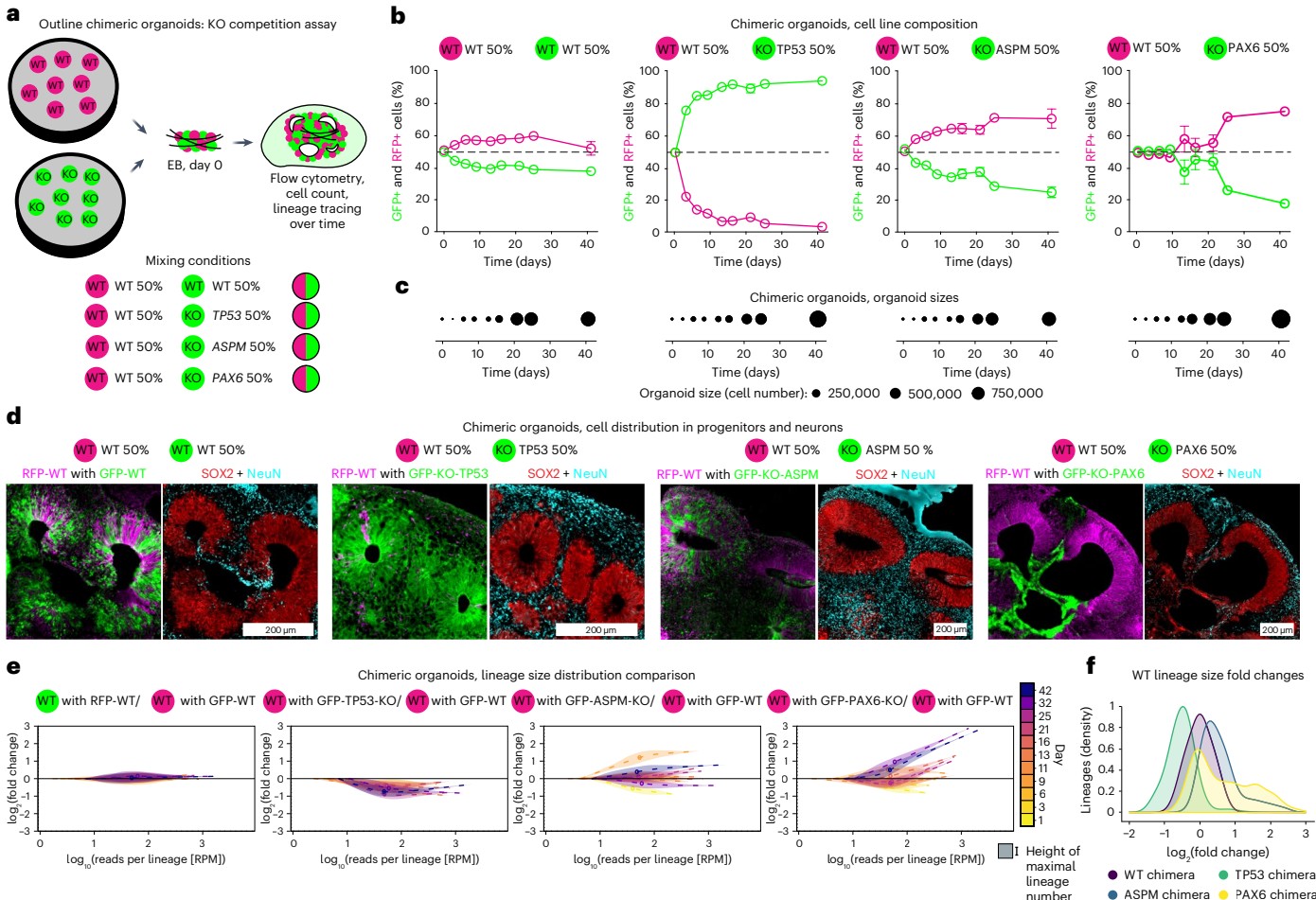

**Fig. 5 | WT to knockout competition assays in cerebral organoids. a**, Scheme of experimental setup of chimeric organoid formation and readout containing 50% RFP-labelled WT cells and 50% GFP-labelled WT cells or cells with *TP53*, *ASPM* or *PAX6* genes knocked out. **b**,**c**, RFP-WT-positive and GFP-KO-positive cell percentages of chimeric organoids (**b**) and overall cell numbers (**c**) over time (*n* = 3 organoids as biological replicates from 1 differentiation experiment at each time point for each chimeric condition). Data points in **b** represent mean ± s.e.m.; area in **c** represents relative organoid size by cell number.
**d**, Immunohistochemistry of RFP-WT-positive and GFP-KO-positive cells alongside for neural stem cells (SOX2) and neurons (NeuN) at day 40. Zoom-in of Extended Data Fig. 9a. **e**, Optimal transport comparison of lineage populations of RFP-WT to either chimeric GFP-WT or to lineage populations of RFP-WT chimeric with selected GFP-KO cells over time. Colour code for time indicated at the right, thickness of curves indicates number of lineages of certain sizes, dashed line indicates mean (*n* = individual lineages from 3 organoids as biological replicates from 1 differentiation experiment at each time point and for each chimeric condition). **f**, Lineage size density of fold changes in optimal transport. WT lineage size fold changes dependent on co-cultured GFP-KO cells in chimeric organoids at day 40. Note that WT lineage size populations are either uniformly unchanged as a population (chimeric with GFP-WT cells, dark purple), smaller (chimeric with GFP-KO-TP53, green), larger (chimeric with GFP-KO-ASPM, blue) or differentially larger across the population (chimeric with GFP-KO-PAX6, yellow) (*n* = individual lineages from 3 organoids as biological replicates from 1 differentiation experiment for each chimeric condition).

minor changes in stem cell/neuron ratios. Lineage tracing and optimal transport analysis revealed a uniform increase in RFP-WT lineage sizes across all lineages indicating that RFP-WT lineages replenished low numbers of GFP-KO-ASPM progeny uniformly across all timepoints (Fig. 5b–e and Extended Data Fig. 9c,g,i). GFP-KO-PAX6 cells initially retained consistent ratios compared to RFP-WT cells in chimeric organoids before a sharp drop in their contribution around day 20, coinciding with dorsal forebrain specification. Immunofluorescent analysis revealed that all progenitors lining neural rosettes and the majority of neurons produced were composed of RFP-WT cells in mature organoids, indicating complete rescue of this tissue fate. Lineage tracing and optimal transport analysis showed that WT-lineages compensated for PAX6-KO lineage loss with some lineages contributing to a high degree (Fig. 5b–e and Extended Data Fig. 9d,h,i). Accordingly individual neural rosettes were completely composed of RFP-WT cells in GFP-KO-PAX6 chimeras (Fig. 5d and Extended Data Fig. 9d). GFP-KO-PAX6 cells were confined to small remnants of neurons in cortical plate areas and discrete regions morphologically resembling choroid plexus, and positive for the choroid plexus marker TTR (Extended Data Fig. 10). Chimeric organoids of PAX6-GFP-KO cells and RFP-WT cells were of similar or even larger size than chimeric WT organoids as were neural rosettes potentially due to signalling mediated by PAX6-GFP-KO choroid plexus tissue (Fig. 5c,d). Importantly, RFP-WT lineages in TP53 and ASPM-KO chimeric organoids responded in a predominately uniform manner compared to RFP-WT lineages of WT chimeric organoids (Fig. 5f). This contrasts with RFP-WT lineages found in GFP-KO-PAX6 chimeras with lineages compensating heterogeneously—a subset of lineages did not react while others replenished by increasing their lineage size up to eightfold. These data are consistent with replenishment of PAX6-KO through RFP-WT lineages only occurring in areas destined for neural progenitor fate. Tissues developing independently of PAX6 function do not require replenishment, resulting in unchanged RFP-WT lineage sizes in these areas. Our data indicate the plasticity of WT cells to modulate neural stem cell output across different lineage sizes, different timepoints and stages of organoids as well as differential tissue demands to ensure robust generation of human brain tissue.

Here we establish whole-tissue lineage tracing in cerebral organoids to assess clonal growth behaviour of entire lineage populations in developing human cerebral organoids. Cerebral organoids grow much more heterogeneous than 2D stem cell cultures, 2D neuronal cultures and mouse brain, potentially mimicking brain tissue growth in vivo[3,5]. Lineage size heterogeneity is enabled by the extended presence of symmetrically dividing progenitors. Our model suggests that IPs producing a defined number of progenies have little effect on lineage size heterogeneity at this stage of development. Furthermore, clonal output of neural stem cell lineages is tunable depending on tissue needs and can compensate for cell loss induced by either chemical or genetic ablation. Thus, our data show that neural stem cell lineage growth is not predetermined but plastic. The experimental approach for whole organoid lineage tracing based on viral barcoding of cells is not feasible in humans; however, this study complements lineage tracing data from post-mortem human brain based on single nucleotide polymorphisms arising during brain development[3–6]. In contrast with labour-intensive imaging approaches, the presented whole-tissue lineage tracing method enables measurement of clonal growth behaviour of all cells in either monogenetic or chimeric in vitro differentiation systems. It can be used to test for the influence of genetic backgrounds and robustness of the whole tissue lineage tracing approach by using other pluripotent stem cell lines beyond the H9 hESCs used in this study. Limitations of this technology are that measurements of clonal behaviour are static as tissue is harvested at the time of interest and that it is cell-type agnostic. Recent single-cell approaches combine a modifiable and dynamic barcode with gene expression data but measure limited cells per sample and face cell type biases due to the need for prior dissociation of the tissue[39,40]. The presented method for whole organoid lineage tracing complements other approaches as it characterizes tissue development as a consequence of entire lineage populations. The approach of lineage tracing an entire developing tissue may serve as a template for other organoid systems where growth dynamics and compensation capacities are unknown and it will be interesting to compare compensation capacity between different tissues and species.

## Online content

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

## Methods

### Cell culture

Feeder-free hESCs line WA09 (H9) was obtained from WiCell, verified to display a normal karyotype and regularly checked for contamination and mycoplasma. All hESCs were cultured on hESCs-qualified Matrigel (Corning, cat. no. 354277) coated plates with Essential 8 Medium (Thermo Fisher Scientific, A1517001). All stem cells were maintained in a 5% CO$_2$ incubator at 37 °C. Cells were split using DPBS (Dulbecco's phosphate-buffered saline) −/− (Gibco, 14190-250) to wash once, followed by 5 min incubation at 37 °C, washed off and plated in Essential 8 Medium supplemented with RevitaCell Supplement (Thermo Fisher Scientific, A2644501).

### Cloning, molecular biology and generating transgenic and KO hESCs lines

For cell lines expressing fluorophore combination transgenes carrying a CAG-TagBFP-P2A-TagBFP, CAG-TagBFP-P2A-EGFP, CAG-TagBFP-P2A-dTomato, CAG-EGFP-P2A-EGFP, CAG-EGFP-P2A-dtomato or CAG-dTomato-P2A-dTomato cassette were inserted into the AAVS1 locus in H9 cells as previously described[41]. The modified AAVS1 targeting vector AAVS-SA-2A-puro was a kind gift from R. Jaenisch (Whitehead Institute for Biomedical Research). Nucleofection was carried out as previously described[16] using the Amaxa nucleofector (Lonza) with the Stem Cell Kit 1. Cells were grown for 7 days, sorted for fluorophore expression and plated as single cells. Colonies were picked and genotyped to confirm transgene presence. For KO cell line generation, the pSpCas9(BB)-2A-GFP (PX458, Addgene: 48138) plasmid, modified to express eCas9 instead of WT-Cas9 and dTomato instead of GFP, was used carrying a gRNA targeting ASPM, TP53 or PAX6. H9-CAG-EGFP-P2A-EGFP cells were nucleofected as described above, sorted by flow cytometry for dTomato expression after 3 days and plated at low density on a 10 cm dish. Single-cell colonies were picked and verified to carry frame-shift mutations.

### Cerebral organoid generation

Cerebral organoids were generated in our lab as previously developed[19,20]. Briefly, hESC cells were grown to 60–80% confluency and single-cell suspensions were obtained using Accutase. Pelleted cells were resuspended in hESC media supplemented with RevitaCell and counted. In total, 24,000 cells and 5–10 poly(lactide-co-glycolide) copolymer (PLGA) fibre microfilaments were seeded to form embryoid bodies in 96-well ultra-low-attachment U-bottom plate (Sigma, CLS7007) in 150 µl E8 supplemented with RevitaCell. On day 3, media was changed to E8, and from day 6 on organoids were grown as previously described[4,6]. In experiments that included puromycin treatment, media was supplemented with 0.3 µg/ml Puromycin from day 11 on (Sigma-Aldrich, P8833). For some experiments, organoids were incubated with 100 µM BrdU (Sigma, B5002) for 1 h followed by washing 24 h prior to analysis.

### Differentiation of neurons in 2D

At day 0, EBs were formed as described above directly in N2B27 differentiation media supplemented with 10 µM SB 431542 (Tocris, 1614), 0.25 µM LDN193189-hydrochlorid (Sigma-Aldrich, SML0559) and 10 µM RevitaCell in 96-well ultra-low-attachment U-bottom plates. N2B27 differentiation media was 1:1 Dulbecco's Modified Eagle Medium:Nutrient Mixture F-12 (DMEM/F12, Gibco, 12330-057):Neurobasal media (Gibco, 21103049) supplemented with N-2 Supplement (100×, Thermo Fisher Scientific, 7502048), B-27 Supplement (50×, Thermo Fisher Scientific, 17504044), NEAA (Sigma-Aldrich, M7145) and GlutaMAX (Thermo Fisher Scientific, 35050-038). On days 2, 4 and 6 media was changed with N2B27 media supplemented with 10 µM SB, 0.25 µM LDN. On day 7 EBs were plated on 96-well plates coated with growth factor reduced Matrigel (Corning, 354230) in N2B27 media. Media was changed every second day with N2B27 media. On day 21 plated EBs were dissociated using Accutase,

and pelleted cells were resuspended in N2B27 media supplemented with 10 ng ml$^{-1}$ human recombinant brain-derived neurotrophic factor (BDNF, Stemcell Technologies, 78005), 10 ng ml$^{-1}$ human recombinant glial cell line-derived neurotrophic factor (GDNF, Stemcell Technologies, 78058) and 10 µM DAPT (Stemcell Technologies, 72082) on poly-L-ornithine (Sigma-Aldrich, P3655) and laminin (Thermo Fisher Scientific, 23017015) coated 24-well plates. Plates were coated with poly-L-ornithine for 1 h at 37 °C, washed three times with DPBS and recoated with 10 µg ml$^{-1}$ laminin for 2 h. Media was changed every second day with N2B27 media supplemented with 10 ng ml$^{-1}$ BDNF, 10 ng ml$^{-1}$ GDNF and 10 µM DAPT (Sigma-Aldrich, D5942). From day 31 on DAPT is left out.

### In utero injection

For this study, we used 10–12-week-old timed pregnant C57BL/6J mice (Jackson Laboratory). Mice were kept at 22 °C +/− l °C, 55% +/− 5 % humidity in a 14 h light/10 h dark cycle under the care of the Institute of Molecular Biotechnology (IMBA) animal care facility. For in utero injections mice were anaesthetized, uterine horns were exposed, and concentrated, Eco-coated retrovirus was injected into the ventricle of E9.5, E10.5 or E13.5 embryos followed by suturing. At E17.5 left and right hemispheres of the cortex and midbrain were harvested. Injections at days E9.5 and E10.5 were guided by ultrasound visualization (Vevo 770, scanhead RMV711, Visualsonics). Embryos were not tested for sex. All procedures were performed in accordance with protocols approved by the Austrian Federal Ministry of Education, Science and Research under the animal experiment licence BMWF.66.015/0023.WF/3b/2017. 3R principles (replacement, reduction and refinement) were followed and monitored by the IMBA Ethics and Biosafety Department.

### Virus construction and viral library preparation

Two retroviral libraries based on pRSF retrovirus expressing TagBFP and containing one of two semi-random barcode libraries were used for lineage tracing. To generate the random barcode libraries, library oligos (Supplementary Table 1) were amplified until they reached logarithmic phase (6–8 cycles, monitored in qPCR machine, stopped in the extension phase). The purified PCR product was cloned into a modified pRSF retroviral vector using XhoI and EcoRI restriction enzymes. Purified ligation reactions were transformed into MegaX DH10 T1R Electrocomp Cells (Thermo Fisher Scientific, C640003) according to the manufacturer's instructions and plated onto lysogeny broth (LB) media plates containing ampicillin. Using dilution series libraries were determined to contain at least 20 × 10$^6$ colonies. Overnight colonies were scraped and recovered for 1 h in LB media containing ampicillin. Plasmid DNA was extracted using a QIAGEN EndoFree Plasmid Maxi Kit (12362).

### Virus production

For WT lineage tracing experiments, plasmid libraries were mixed prior to making retrovirus. As both libraries acted highly similarly in WT they were used separately to distinguish individual genotypes in chimeric organoid experiments. Retrovirus was produced in HEK293 grown in DMEM/10% fetal bovine serum/2 mM L-glutamine/100 units ml$^{-1}$ penicillin/0.1 mg ml$^{-1}$ streptomycin and coated using vesicular stomatitis virus G (human cells) or Eco-R (mouse). Supernatant was collected for 72 h every 12–24 h, pooled, pelleted at 28,000$g$ and resuspended in stem cell medium containing 4 µg ml$^{-1}$ polybrene (Sigma-Aldrich, H9268) and 100 µg ml$^{-1}$ Synperonic F108 (Sigma-Aldrich, 07579) (hESCs) or Hanks' balanced salt solution (mouse). Human cells were split 1:2.5 1 h prior to infection using Accutase. Infections were performed for 3 h, followed by two washes using DMEM/F12 and addition of fresh Essential 8 medium supplemented with RevitaCell.

### Lineage tracing with genomic barcoding

Virus was produced and hESCs and mice infected. One to two days after infection single stem cell suspensions were prepared, sorted by flow cytometry for TagBFP (4–7% TagBFP-positive, 1.3–1.8 × 10$^6$ cells

total sorted, actual infection rates were determined to be up to 19% on not previously sorted cells three days after infection) and plated in E8 supplemented with RevitaCell to recover for 1–2 days before organoid setup. Minimal times between viral library infection, sorting and organoid setup were kept to prevent expansive clone growth. Organoids, stem cells and mouse brains (E17.5) were harvested at days 0–42. Samples were lysed in a single reaction vessel in 400 µl gDNA extraction buffer (10 mM Tris pH 8, 150 mM NaCl, 10 mM EDTA, 0.1% sodium dodecyl sulfate (SDS) and 0.2 mg ml$^{-1}$ Proteinase K) incubated at 55 °C O/N followed by 85 °C for 1 h. RNase A (1 µl; Thermo Fisher Scientific, EN0531) and RNase T1 (0.2 µl; Thermo Fisher Scientific, EN0541) were added, incubated at 37 °C for 2 h followed by phenol/chloroform extraction and EtOH precipitation. After 12 freeze/thaw cycles viral barcode inserts were amplified using lineage tracing genomic DNA primers with sequencing adaptors. Libraries were sequenced on Illumina HiSeq 2500 and Novaseq S1 lanes. At least three replicates per condition and time point were analysed. Out of 476 samples sequenced in total, 20 were excluded from further analysis due to inefficient PCR amplification. In spite of exclusion of those samples, at least two measurements per experimental condition were used for analysis in all cases[22].

### Dissociating organoids for flow cytometry analysis
For flow cytometry analysis organoids were dissociated in a 10:1 mixture of Accutase (Sigma-Aldrich, A6954) and 10× Trypsin (Gibco, 15400) shaking at 37 °C and filtered through a 35 µm strainer. Aliquots were counted on an Invitrogen Countess II cell counter. Antibody staining was performed in DMEM/F12 + 4% BSA using the following antibodies: anti-NCAM (BD Biosciences, 564058), anti-CXCR4 (BD Biosciences, 560936) and anti-TRA-1-60 (BD Biosciences, 563188).

### RNA extraction and qPCR
RNA samples and qPCR was performed as described previously[25]. In short, RNA samples were processed using the Qiagen RNeasy kit (Qiagen, 74004). cDNA was generated using Superscript (Invitrogen, 18080044) and qPCR was performed using Promegas qPCR master Mix (Promega, A6002). Primers used were published previously[25].

### Western blotting
For TP53 induction, Nutlin-3a (10 µM) was added to organoid culture media for 24 h prior to lysis. Organoids were lysed with radioimmunoprecipitation assay buffer (Thermo Fisher Scientific, 89900) with complete protease inhibitor (Roche, 05056489001). Protein was quantified by bicinchoninic acid assay (Thermo Fisher Scientific, 232250) and 25 µg of sample was denatured in reducing Laemmli buffer (Thermo Fisher Scientific, J61337) before protein separation by SDS–PAGE on 4–12% Bis-Tris gels in 3-(N-morpholino)propanesulfonic acid (MOPS) SDS buffer (Invitrogen, NP0323BOX, NP0001). Proteins were transferred to nitrocellulose membranes and incubated in 4% skimmed milk with primary antibody including anti-LaminB1 (1:2,000, GeneTex, 103292), and either anti-PAX6 (1:1,000, Biolegend, 901301) or anti-TP53 (1:1,000, AbCam, ab32389). Species-specific secondary antibodies (both 1:10,000, Li-cor, 68070, 32211) were used to detect bands by fluorescence (BioRad ChemiDoc MP).

### Cryosectioning and immunofluorescence
For histological staining, tissues were fixed overnight in 4% paraformaldehyde. For cryosectioning fixed organoids were embedded in OCT (Sakura, 4583), cut at 18–20 µm thickness and processed for immunofluorescence using standard methods. The following primary antibodies were used: anti-SOX2 (Abcam, ab97959, 1:500), anti-NeuN (Millipore, MAB377, 1:500), anti-DCX (Santa Cruz, sc8066, 1:1,000), anti-MAP2 (Abcam, ab5392, 1:1,000), anti-Nestin (BD, 611658, 1:1,000), anti bIII-tubulin (Abcam, ab18207, 1:1,000), anti-BrdU (Thermo Fisher Scientific, B35128, 1:500), anti-phospho-vimentin(Ser82) (MBL, D095-3S, 1:250), anti-TTR (AbD Serotec, ahp1837, 1:500), anti-aPKC

(SantaCruz, sc-17781, 1:100), anti-nestin (Abcam, ab105389, 1:1,000), anti-β-catenin (Sigma, C2206, 1:250), anti-N-cadherin (BD, 610920, 1:500), anti-ASPM (Bethyl Laboratories, IHC-00058, 1:100), anti-TBR2 (Abcam, ab23345, 1:250) and anti-TBR1 (Abcam, ab31940, 1:500). Secondary antibodies raised in donkey or goat were purchased from Thermo Fisher Scientific: donkey anti-mouse immunoglobulin G (IgG) (H+L), Alexa Fluor 488, A-21202, 1:1,000; donkey anti-rabbit IgG (H+L), Alexa Fluor 488, A-21206, 1:1,000; donkey anti-goat IgG (H+L), Alexa Fluor 488, A-11055, 1:1,000; donkey anti-sheep IgG (H+L), Alexa Fluor 488, A-11015, 1:1,000; goat anti-rabbit IgG (H+L), Alexa Fluor 488, A-11008, 1:1,000; goat anti-mouse IgG (H+L), Alexa Fluor 488, A-11001, 1:1,000; goat anti-mouse IgG1, Alexa Fluor 488, A-21121, 1:1,000; goat anti-rabbit IgG (H+L), Alexa Fluor 568, A-11011, 1:1,000; goat anti-mouse IgG (H+L), Alexa Fluor 568, A-11004, 1:1,000; donkey anti-goat IgG (H+L), Alexa Fluor 568, A-11057, 1:1,000; donkey anti-mouse IgG (H+L), Alexa Fluor 568, A-11037, 1:1,000; goat anti-mouse IgG2b, Alexa Fluor 568, A-21144, 1:1,000; donkey anti-chicken IgY (H+L), Alexa Fluor 568, A-78950, 1:1,000; donkey anti-rabbit IgG (H+L), Alexa Fluor 647, 31573, 1:1,000; donkey anti-goat IgG (H+L), Alexa Fluor 647, 21447, 1:1,000; goat anti-rabbit IgG (H+L), Alexa Fluor 647, 21245, 1:1,000; and goat anti-rabbit IgG (H+L), Alexa Fluor 647, 21244, 1:1,000. If possible, all secondary antibodies were of highly cross-adsorbed quality.

### Imaging and microscopy
Imaging was performed on Zeiss (LSM 780, 800, 880, Z1) or Olympus (IX83) microscopes. Channels were collected individually with excitation being delivered by lasers of 405, 488, 561 and 633 nm wavelength. Alternatively, imaging was performed on Panoramic250 slidescanners from 3DHistech. Images were processed in the respective companies' image suites and FIJI. All equipment was kept in immaculate condition by the IMP/IMBA biooptics facility.

### Transduction of cerebral organoids for live imaging
Cerebral organoids (week 6) were embedded in 3% low melting point agarose dissolved in DMEM/F-12. Organoids were sliced with a Leica VT1200S vibratome in ice-cold DMEM/F-12 into sections of 250 µm. Sections were transferred to a 24-well plate, transduced with a GFP-encoding retrovirus (MSCV-IRES-GFP: Addgene 20672) or adenovirus (Vector Biolabs, 1060) and incubated for 2 h at 37 °C. After four washes with DMEM/F-12, the slices were transferred to Millicell cell culture inserts (Merck) and placed in a six-well plate containing cortical culture medium (DMEM/F-12 containing B27, N2, 10 ng ml$^{-1}$ fibroblast growth factor, 10 ng ml$^{-1}$ epidermal growth factor, 5% fetal bovine serum and 5% normal horse serum). Slices were cultured for 2 days before live imaging.

### Live imaging of cerebral organoids
Prior to live imaging, culture inserts containing the organoid slices were transferred to a six-well glass bottom plate (Cellvis) and fresh cortical culture medium was added. GFP-positive cells were live imaged for 48 hours on a spinning disk wide microscope equipped with a Yokogawa CSU-W1 scanner unit. The microscope was equipped with a high working distance (WD 6.9–8.2, 414 mm) 20× Plan Fluor ELWD NA 0.45 dry objective (Nikon), and a Prime95B SCMOS camera. Z-stacks of 80–100 µm range were taken with a step size of 4–5 µm at intervals of 15 minutes. Temperature and CO$_2$ levels were controlled with a stage top incubator (Tokai Hit). Raw images were processed with NIS-Elements and FIJI ImageJ.

### Immunostaining after live imaging
Cerebral organoid slices were fixed in 4% PFA for 20 minutes. Slices were blocked and permeabilized in 5% normal donkey serum 0.25% Triton X-100 for 1 hour at room temperature. Primary and secondary antibody mix was prepared in 5% normal donkey serum 0.1% Triton X-100 and incubated overnight at 4 °C. Samples were mounted

in VECTASHIELD HardSet (Vector Laboratories) and mosaics (3 × 3 tiles) were acquired with CFI Apo LWD Lambda S ×40 objective (NA 1.15; WD 0.61–0.59, Nikon). Primary antibodies used included: sheep anti-EOMES (1:200, R&D Systems AF6166), chicken anti-GFP (1:500, Abcam Ab13970), mouse anti-Sox2 (1:500, Abcam Ab79351) and rabbit anti-NeuroD2 (1:500, Abcam Ab104430).

### Live-fixed correlative microscopy

Correlative microscopy was performed as described previously[28]. In brief, live and fixed images were segmented, paired and aligned using an ImageJ and MATLAB-based macro. Once positional information was retrieved and transferred to the fixed images, high-resolution mosaics (3 × 3 tiles) were acquired and cell fate was assessed using canonical cell fate markers.

### Barcoded scRNA-seq

A plasmid library was cloned as described above to insert a 20 bp semi-random barcode into the 3′ UTR (untranslated region) of a Pol II transcript encoding GFP (sequences in Supplementary Table 1). The purified PCR product containing the barcode was cloned into the backbone using EcoRI and NheI restriction sites. Infected hESCs were as described above. After 48 hours, GFP-barcoded cells were mixed with BFP-expressing cells, which had been transduced with an equivalent, non-barcoded retrovirus. Cells were mixed at a ratio of 5:95 (GFP:BFP) and were seeded to generate organoids.

Organoids were dissociated at day 18 or 42 as before, filtered through a 35 µm strainer followed by dilution with 0.1% BSA in DPBS −/− and stained with Draq7 viability dye (Biostatus; DR70250, 0.3 mM). Live, single, GFP-positive cells were sorted with a BD FACSAria III (70 nm nozzle) and collected in a 96-well PCR plate pre-loaded with 10 µl of 0.5% BSA in DPBS −/−. Single organoids were sorted to completion into a single well and the entire resulting cell suspensions were subjected to scRNA-seq analysis using the Chromium Single Cell 3′ Reagent Kit (v.3.1) following the 10x Genomics user guide with the following modification. During cDNA amplification (2.2), the PCR reaction was paused at 72 degrees after six cycles. An aliquot (75 µl) of sample is removed for targeted amplification and replaced with 37.5 µl of Amp Mix, 11.25 µl cDNA Primers and 26.25 µl H$_2$O before continuing for a total of 13 cycles. Targeted amplification to enhance barcode recovery was performed as described previously[38]. Briefly, sixth cycle cDNA from 10x libraries were PCR amplified in a hemi-nested manner using limited PCR cycles with AmpliTaq Gold 360 (Thermo Fisher Scientific, 4398876). Sequencing of 10x gene expression libraries and targeted amplification libraries was performed on an Illumina NovaSeq S4 lane.

### scRNA-seq analysis

Per sample, sequenced reads were aligned to the GRCh38 2020-A human reference genome with Cell Ranger version 7.01 (10x Genomics) using default parameters. Resulting cell-by-gene, unique molecular identifier (UMI) count matrices were analysed in R using Seurat version 4.3.0.1. We filtered for high-quality cells based on doublet detection performed with scrublet (version 0.2.3), number of uniquely detected genes ('nFeature'), number of UMIs ('nCount'), percentage of mitochondrial reads and percentage of ribosomal reads. The latter four thresholds were set for each sample separately and are listed in Supplementary Table 2. Subsequently, the remaining high-quality cells were merged for day 18 and day 42 samples. The following analysis was performed for both timepoints: counts were log-normalized cell-wise ('LogNormalize'), 2,000 highly variable genes were identified ('FindVariableFeatures') and scaled ('ScaleData'). Next, principal component analysis was performed ('RunPCA') on a custom gene list used previously[25] and based on the first 50 components a UMAP embedding was computed ('RunUMAP'). To cluster the data, the shared nearest neighbour graph was calculated ('FindNeighbors', also based on the first 50 principal components) and used as input for the Louvain

clustering algorithm ('FindClusters', resolution 1.2 for day 18 and 0.7 for 42). Clusters were annotated based on expression of known marker genes for radial glia, intermediate progenitors and neurons.

### Bioinformatic analyses

The following sequencing information was extracted from the raw reads: the 50-base-long viral barcode was extracted from read 1; the sample barcode was extracted from index 1 and 2. Sample barcodes were error-corrected by comparison to known input sequences, while base errors in viral barcodes were corrected by a clustering algorithm[42]. Lineage barcodes that were found in a different sample with a 20× or higher read count were removed from the sample with the lower read count. Lineage barcodes with a read count lower than a sample-specific threshold are removed. Sample-specific thresholds are chosen as suggested previously[43]. Samples for which lineage barcodes overlapped more often with lineage barcodes of other samples than expected or with low lineage or read count were removed as outliers.

### Optimal transport

To compare the lineage size distributions of two samples $\{x_i \mid i = 1, …, N\}$ and $\{y_i \mid i = 1, …, M\}$ we used an optimal transport approach. If two samples had a different read threshold, the higher read threshold was applied to both samples. Both samples were normalized to reads per million. If the number of lineages differed between the two samples, each lineage in sample $x_i$ was counted as lcm($N$, $M$)/$N$ lineages and each lineage in sample $y_i$ as lcm($N$, M)/$M$ lineages. We then find a transport map $T(x)$ that maps each lineage $x_i$ to a lineage $T(x_i) = y_i$ so that the sum of fold changes of the lineage sizes $\sum \log_2(x_i) - \log_2(T(x_i))$ is minimal. The solution to this optimal transport problem is given by mapping rankwise, so that the lineage with rank $i$ in sample $x_{(i)}$ is mapped to the lineage with rank $i$ in sample $T(x_{(i)}) = y_{(i)}$. For the fold-change curves, the top 1% lineages were removed due to high graph variability, after which the graph $\{(x_{(i)}, T(x_{(i)}))\}$ is resampled and smoothed with LOWESS regression and averaged over all replicates.

### Modelling

A lineage is modelled as its number of S cells ($s$), A cells ($a$) and N cells ($n$). Three events can modify the cell counts: symmetric division (+1 S cell), transition (−1 S cell, +1 A cell) and asymmetric division (+$k$ N cells) with corresponding rates $r_s$, $r_a$ and $r_n$. Each lineage is independently simulated stepwise. In each step the number of events is approximated by the binomial distribution with mean $rx$ where $r$ is the rate of the event and $x$ is the number of originating cells (that is, S cells for symmetric division, S cells for transition and A cells for asymmetric division). The time step is chosen as 0.01/max($r_s$, $r_a$, $r_n$) so that the probability of the same cell dividing twice in one time step (which the binomial distribution does not capture) is smaller than 1%. An organoid is simulated as 10,000 independent lineages.

### Statistics and reproducibility

No statistical method was used to predetermine sample size but our sample sizes are similar to those reported in previous publications[16,19,25]. No data were excluded from analyses with the exception of low-quality sequencing libraries as indicated in the corresponding Methods section. Experiments were not randomized, and investigators were not blinded to allocation during experiments and outcome assessment. Statistical testing in this study (Kolmogorov–Smirnov) does not require normal distribution of the data; therefore it was not formally tested. At least two independent biological experiments were performed including multiple replicates for most experiments with comparable results. Exceptions include some whole organoid lineage tracing experiments, which were conducted only once with three individual organoids per timepoint and conditions and four mice per timepoint. Barcoded scRNA-seq experiments were performed from one batch with one organoid per 10x library in organoid duplicates per two timepoints.

Micrographs show representative images from at least two experiments including multiple organoids.

## Reporting summary

Further information on research design is available in the Nature Portfolio Reporting Summary linked to this article.

## Data availability

Sequencing data generated for this study have been deposited in the Gene Expression Omnibus under accession code GSE214105. Previously published lineage tracing data that were re-analysed here are available under accession code GSE151384. Source data are provided with this paper. All other data supporting the findings of this study are available from the corresponding authors on reasonable request.

## Code availability

Code for stochastic modelling and optimal transport fold-change plots is available at https://github.com/Cibiv/pyrganoid.

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

## Acknowledgements

We thank all members of the Knoblich Lab for technical expertise and feedback. We thank M. Balmana for performing select immunofluorescence stainings. We thank IMP/IMBA biooptics services for expert microscopy and flow cytometry services. We thank B. Schroeder and D. Malzl (CIBIV) for sequencing analysis. We thank A. Sommer and the VBCF NGS unit (www.viennabiocenter.org/facilities) for consultation and sequencing. D.L., S.H., J.B.L., J.N. and D.R. are members of the VBC doctoral programme. F.G.P. and S.H. are members of the FWF SFB-F78 programme. C.E. received funding from an EMBO postdoctoral fellowship (ALTF 966-2010). J.B.L. is supported by a doctoral fellowship from Boehringer Ingelheim Fonds. Work in the laboratory of J.A.K. is supported by the Austrian Academy of Sciences, the Austrian Science Fund (FWF), SCORPION DOC 72-B27, Special Research Programme F7804-B and Stand-Alone grants P 35680 and P 35369, the Austrian Federal Ministry of Education, Science and Research, the City of Vienna, the Gilbert Family Foundation (award 923017) and the Simons Foundation Autism Research Initiative (SFARI, 724430). D.R. was supported by the ERC Advanced Grant (695642) under the European Union's Horizon 2020 programme, the FWF P 35369 grant and the Austrian Lotteries. Work in the laboratory of A.v.H. is supported by the Austrian Science Fund (Special Research Program SFB-F78, F 7811-B) and by the Austrian Science Fund (FWF, F78).

## Author contributions

D.L., S.H., C.E. and J.A.K. designed the study, analysed data and wrote the paper with input from all authors. D.L., C.E., J.B.L. and C.B.A. performed experiments with help from E.G.P.v.d.V. and D.R.; S.H., J.N. and F.G.P. performed bioinformatic analyses under the supervision of A.v.H.; J.A.K., A.v.H. and A.D.B. acquired funding.

## Competing interests

J.A.K. is on the supervisory and scientific advisory board of a:head bio AG (https://aheadbio.com) and is an inventor on several patents relating to cerebral organoids. The other authors declare no competing interests.

## Additional information

**Extended data** is available for this paper at https://doi.org/10.1038/s41556-024-01412-z.

**Correspondence and requests for materials** should be addressed to Christopher Esk or Jürgen A. Knoblich.

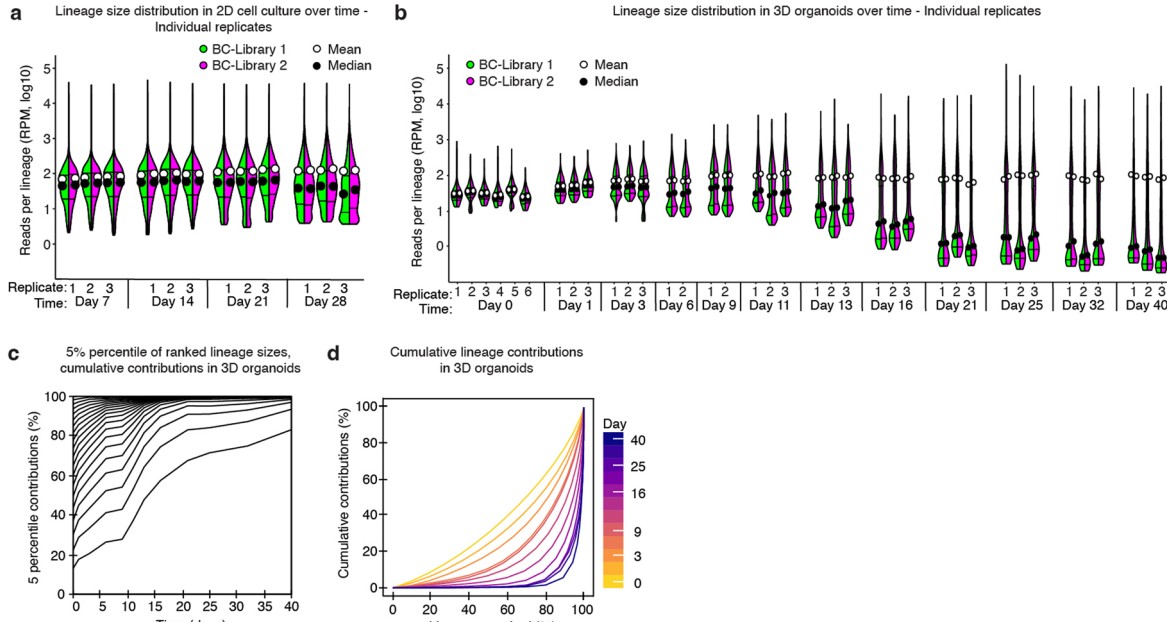

**Extended Data Fig. 1 | Details on whole organoid lineage tracing. a**, 2D barcoded lineage tracing data. Normalized reads per lineage (RPM) plotted for the indicated time points for each barcode. Individual Barcode Libraries 1 and 2 are indicated. (n = individual lineages of individual biological replicates from 1 experiment at each time point are shown). **b**, 3D lineage tracing data. Normalized reads per lineage (RPM) plotted for the indicated time points for each barcode. Individual Barcode Libraries 1 and 2 are indicated. (n = individual lineages of individual organoids as biological replicates from 1 differentiation experiment at each time point are shown). **c**, Contribution to organoids of ranked 5 percentiles of lineages over time in 3D organoids, normalized to whole organoids. (n = individual lineages from 3 organoids as biological replicates from 1 differentiation experiment at each time point). **d**, Cumulative contributions of lineages to whole 3D organoids over all time points. (n = individual lineages from 3 organoids as biological replicates from 1 differentiation experiment at each time point).

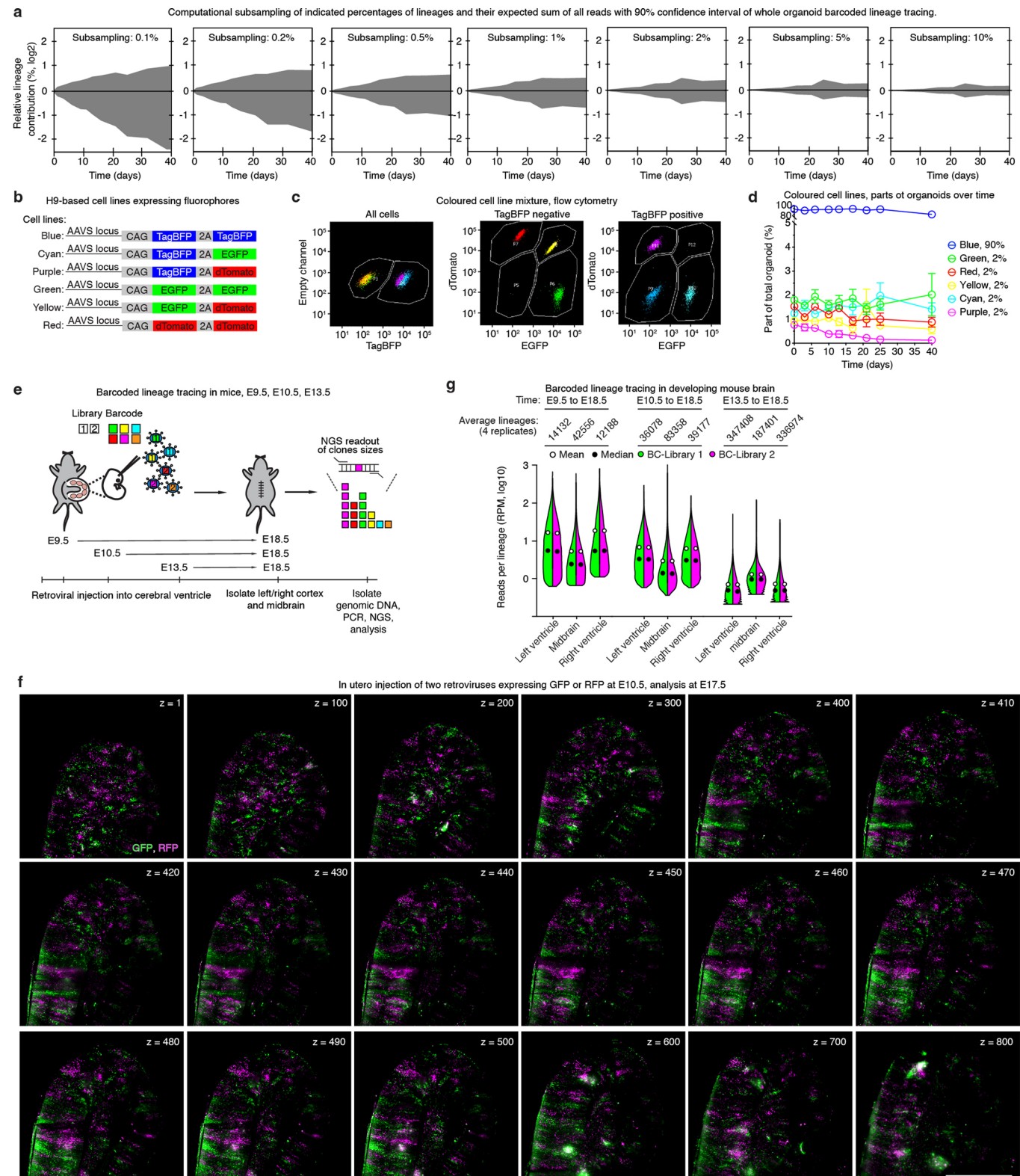

**Extended Data Fig. 2 | See next page for caption.**

**Extended Data Fig. 2 | Verification of whole organoid barcoded lineage tracing by FACS in organoids and barcoded lineage tracing in developing mouse embryos. a**, In silico subsampling of indicated percentages of lineages in organoids with 90% confidence range of subsampled results indicated. **b**, Scheme of fluorescent H9-based WT cells lines generated. **c**, Example of flow cytometry separation of fluorescent cell lines in equal mixtures. **d**, Cell line contribution of fluorescent cell lines in chimeric organoids mixed at 90:2:2:2:2:2 contribution of cell lines ('Blue', 'Cyan', 'Purple', 'Green', 'Yellow', 'Red') over time. (n = 3 organoids as biological replicates from 1 differentiation experiment at each time point). Data points represent mean ± SEM. **e**, Outline of barcoded lineage tracing in mice from E9.5, E10.5 and E13.5 to E18.5. **f**, In utero injection of two fluorescent retroviruses carrying GFP and RFP at E10.5 with analysis following on E17.5. One embryo hemisphere was cleared and imaged entirely for GFP and RFP. Individual z-sections are indicated. Note the lack of overlap of GFP and RFP signal indicating single viral integrations. **g**, Lineage size distributions in lineage traced mouse brains. Individual Barcode Libraries 1 and 2 as well as the number of detected lineages are indicated. Left and right hemispheres as well as midbrain of individual embryos were scored as indicated. (n = individual lineages from 4 mouse brains as biological replicates split into left ventricle, right ventricle and midbrain at each time point).

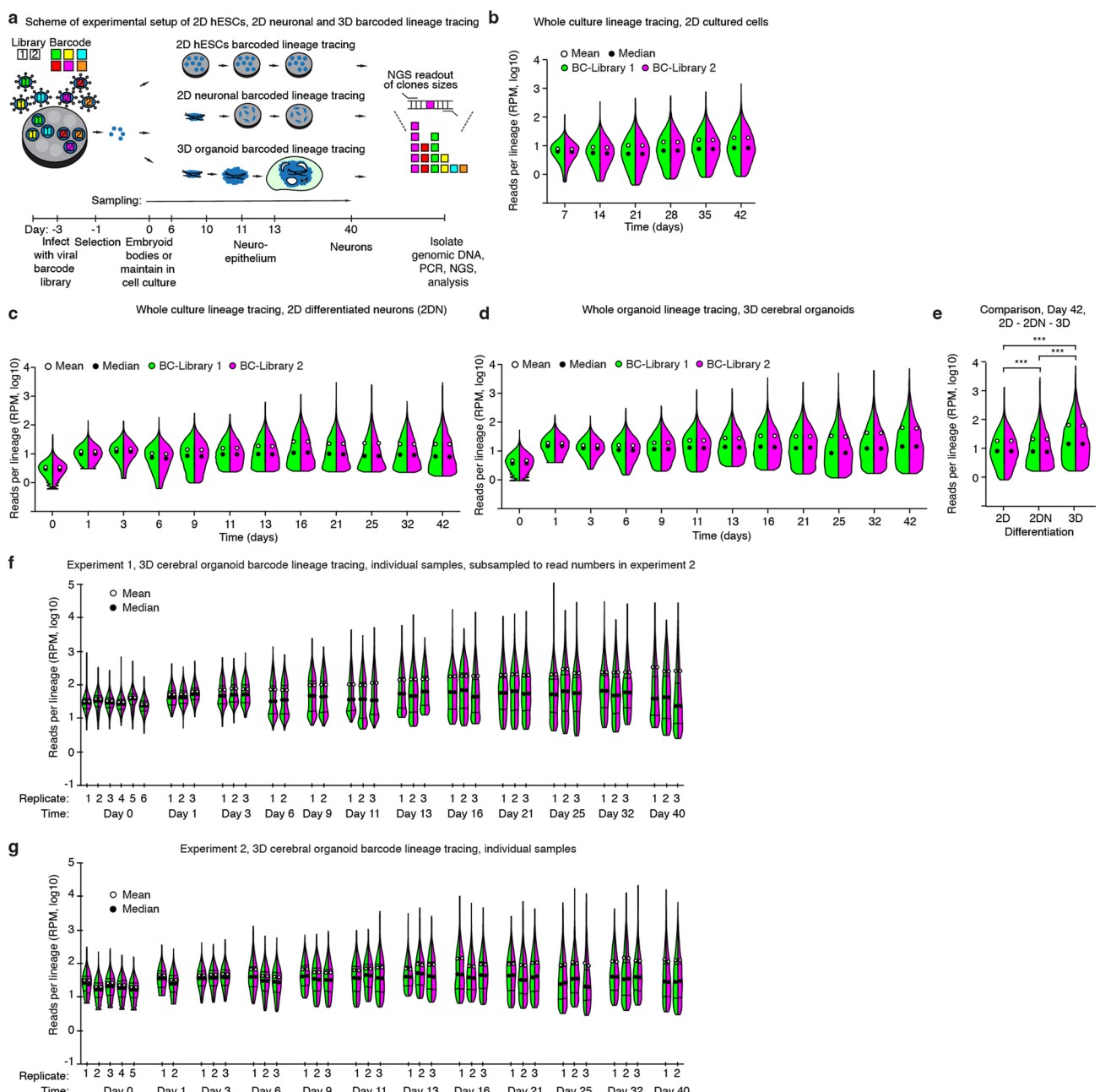

**Extended Data Fig. 3 | Lineage size distribution dependent on culture conditions in barcoded lineage tracing. a**, Outline for different culture conditions used for lineage tracing experiments. **b**, Whole culture lineage tracing of WT hESC in 2D cell culture. (n = individual lineages from 3 biological replicates from 1 experiment at each time point). **c**, Whole culture lineage tracing of WT hESC cells differentiated to neurons in 2D (2DN). (n = individual lineages from 3 biological replicates from 1 differentiation experiment at each time point). **d**, Whole culture lineage tracing of WT hESC cells grown to form 3D tissue cerebral organoids. Individual Barcode Libraries 1 and 2 are indicated. (n = individual lineages from 3 organoids as biological replicates from

1 differentiation experiment at each time point). **e**, Comparison of 2D, 2DN and 3D lineage size distributions at day 42. ***$P < 10^{-4}$ by two-sample Kolmogorov-Smirnov test. (P-values: 2D − 2DN = $1 \times 10^{-324}$, 2D − 3D = $3 \times 10^{-310}$, 2DN − 3D = $2 \times 10^{-84}$). (n = individual lineages from 3 biological replicates from 1 experiment for each 2D, 2DN and 3D).**f, g**, Comparability of 3D lineage tracing experiments performed. Reads of experiment 1 (e) were down sampled to match read numbers of experiment 2 (f). (n = individual lineages of individual organoids as biological replicates from 1 differentiation experiment at each time point for experiment 1 and experiment 2 are shown).

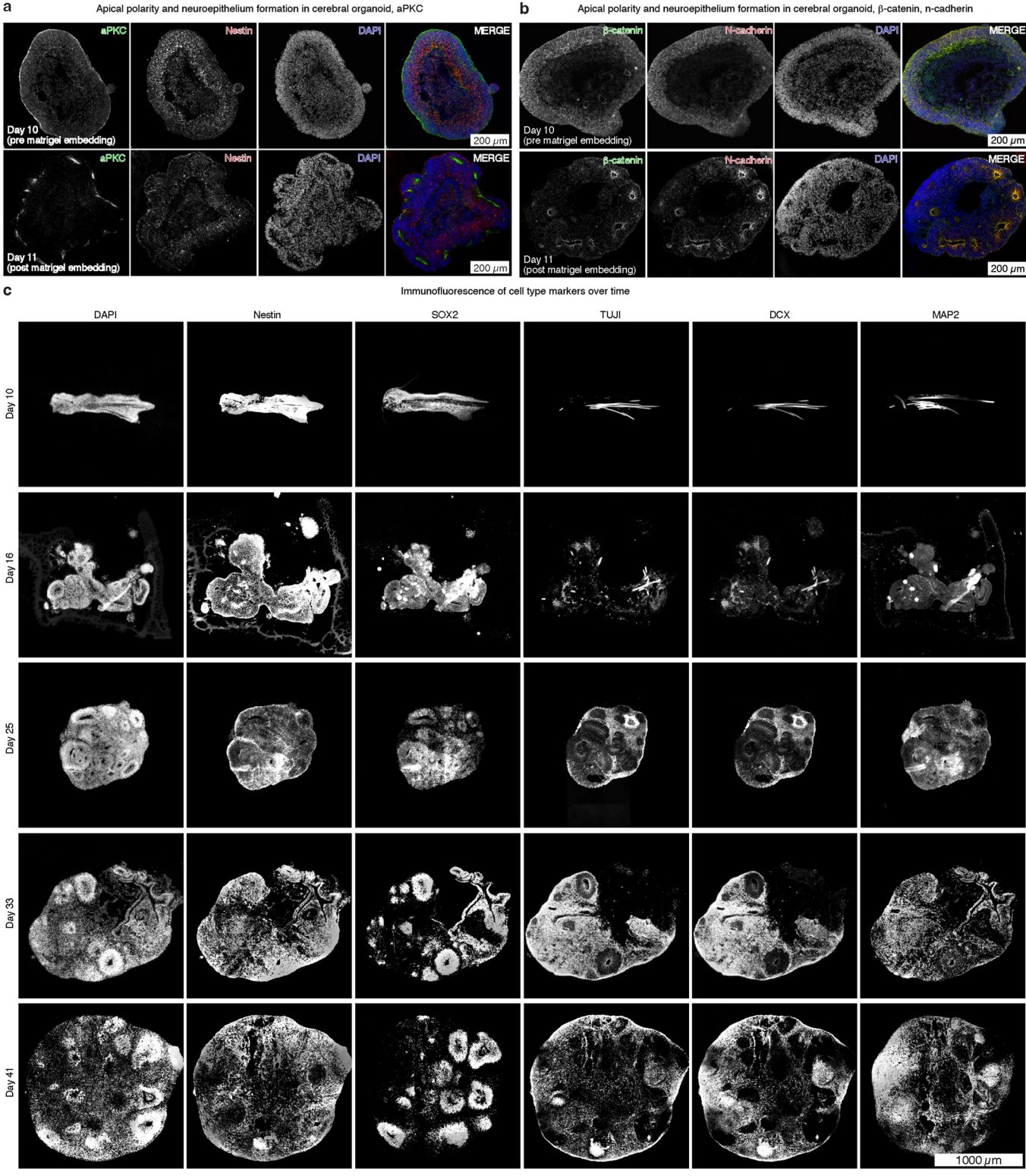

**Extended Data Fig. 4 | Timing of neuroepithelium formation at day 11 in cerebral organoids and immunofluorescent characterization of organoid cell composition over time. a**, Immunofluorescent characterization of organoids at indicated timepoints using apical (aPKC) and stem cell (Nestin) markers. **b**, Immunofluorescent characterization of organoids at indicated timepoints using adherens junction markers (b-catenin, n-cadherin).

**c**, Immunofluorescent characterization of cerebral organoids at indicated timepoints on the left and markers for nuclei (DAPI), neural stem cells (Nestin, SOX2) and neurons (TUJ1, DCX, MAP2) indicated on the top. Note that fibres in organoids display autofluorescence. Adjacent cryosection per timepoints were used.

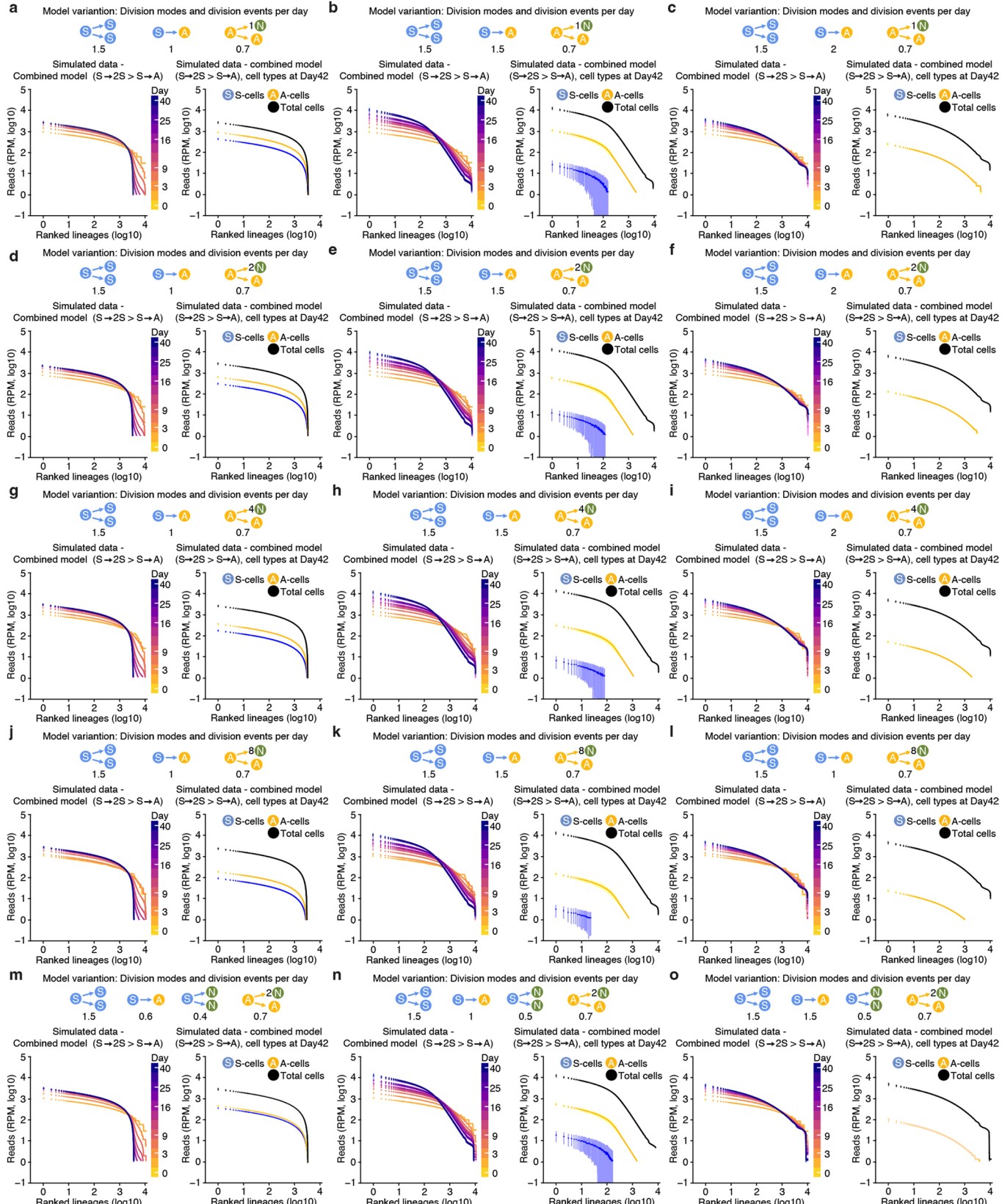

**Extended Data Fig. 5 | Variations of organoid growth model. a-o,** Rank plots of modelled lineage size distribution encompassing symmetric divisions, S-cell to A-cell transitions, asymmetric divisions and S-cell consuming divisions leading to generation of S-, A- and N-cells over time. Rates for individual model steps are indicated at the top. Left panels: Lineage size distribution over time as per legends. Right panels: Cell type compositions at endpoint, day 42.

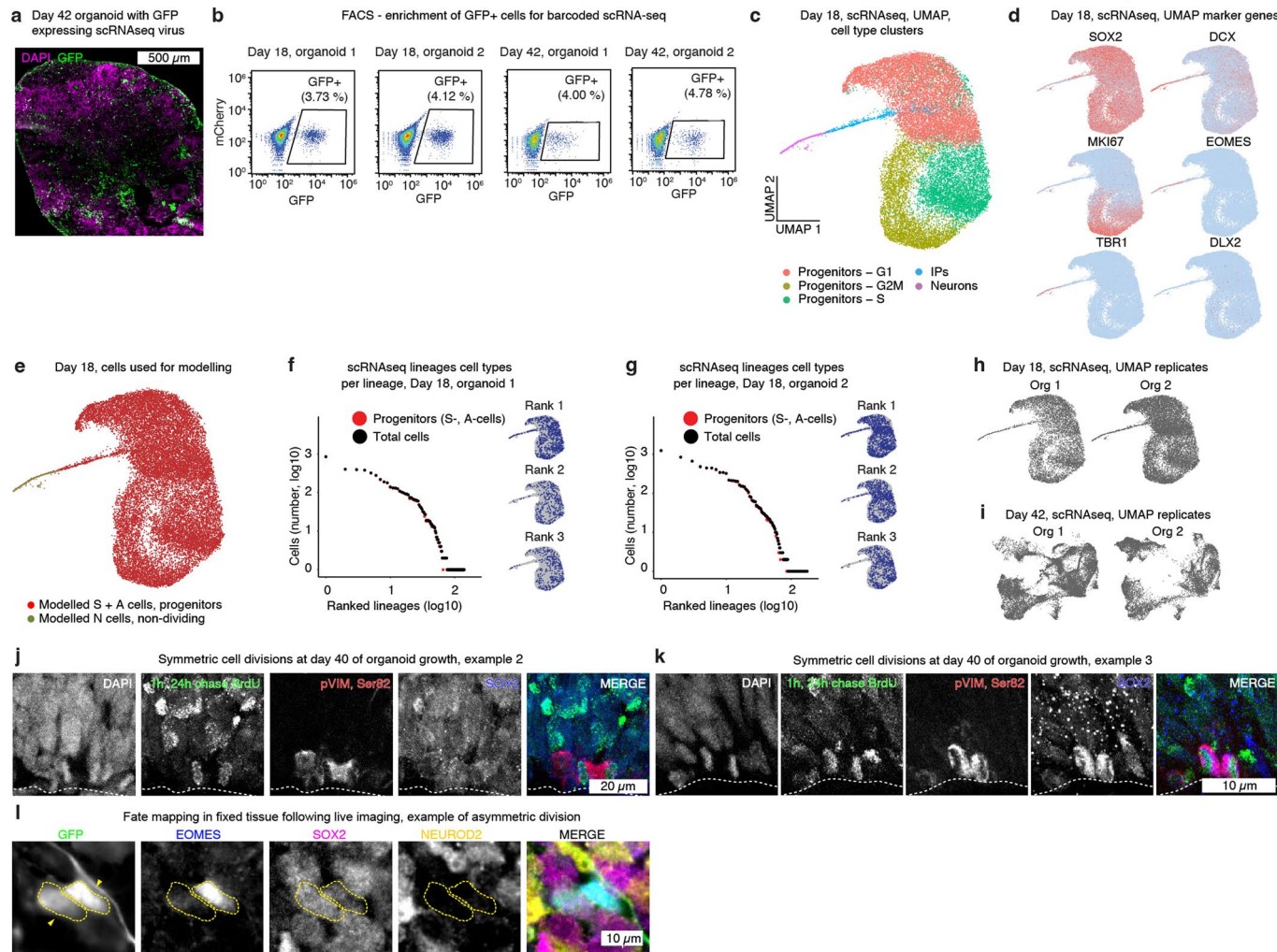

**Extended Data Fig. 6 | Symmetric cell divisions in late organoids.**
**a**, Representative example of an organoid at day 42 with lineage tracing virus (GFP, green) from the same experiment as used for scRNA-seq. **b**, FACS plots of sorted cells for scRNA-seq analysis per organoid. (n = 2 organoids as biological replicates from 1 differentiation experiment). **c-e**, UMAPs of day 18 organoid cells in scRNA-seq indicating cell types (b), select markers for cell type annotation (c) and cell classifications in S- plus A- and N-cells (d). Ips, intermediate progenitors.

(n = 2 organoids as biological replicates from 1 differentiation experiment). **f, g**, Lineage size rank plots for day 18 organoid 1 (f) and 2 (g). **h, i**, UMAP of individual organoids at day 18 (h) and day 42 (i). **j, k**, Immunofluorescent characterization of cerebral organoids at day 40 for nuclei (DAPI), incorporated BrdU (1 h pulse 24 h prior to fixation), dividing cells (pVIM, Ser 82) and neural stem cells (SOX2). Example 2 (j) and 3 (k). See Fig. 3g, h, for overview and example 1. **l**, Live imaging example 2. See Fig. 3i–m for overview and quantification.

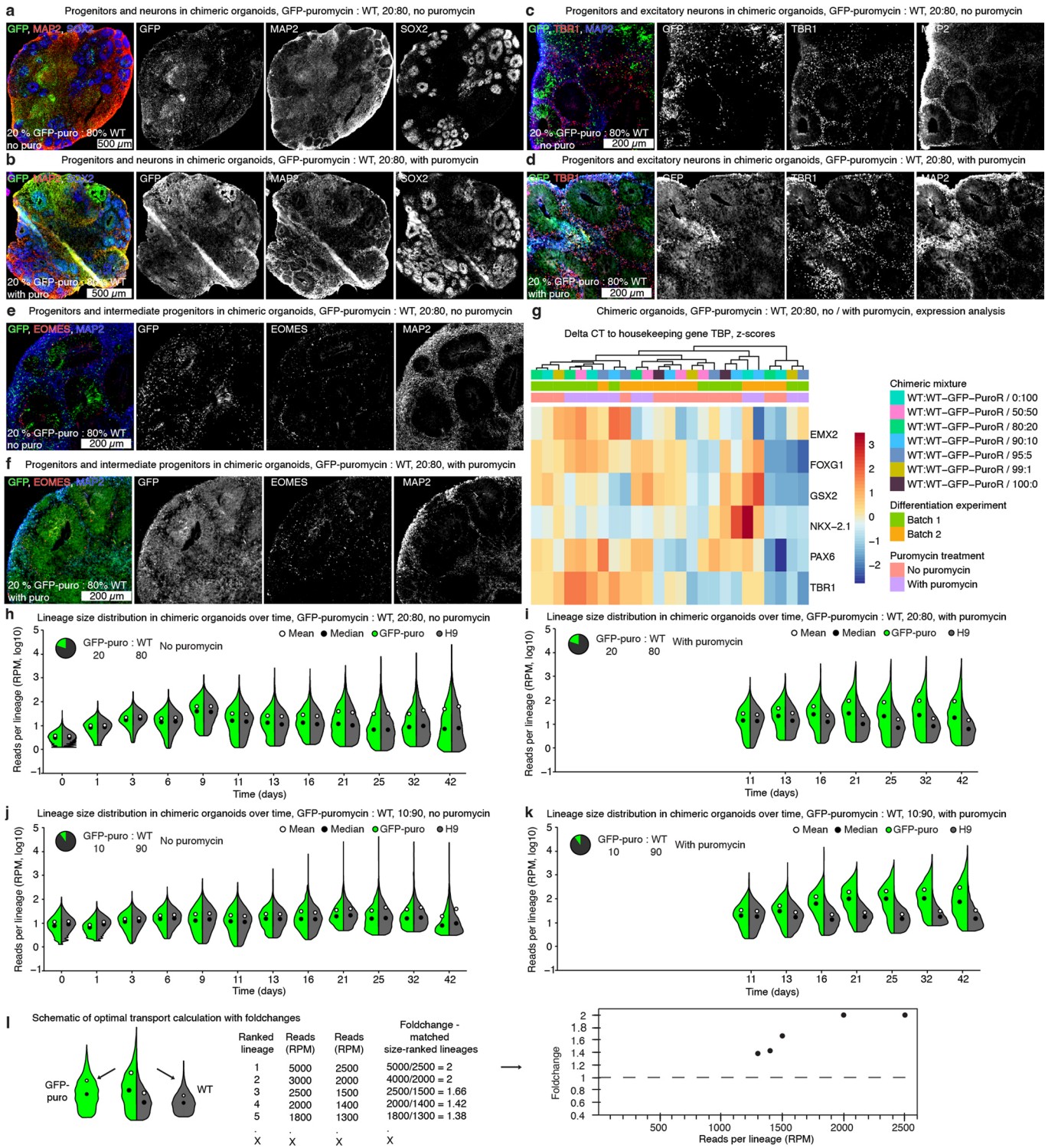

**Extended Data Fig. 7 | Replenishment in WT:GFP-puromycin-resistant chimeric organoids. a-f,** Immunofluorescent analysis of day 42 organoids chimeric for WT:GFP-puro (starting at 80:20 ratios) with (b, d, f) or without (a, c, e) puromycin treatment from day 11 on. Antibody labelling indicated in panels. **g,** qPCR analysis of various chimeric H9 WT and GFP-puromycin-resistant organoids (n = 6 organoids as biological replicates from 2 differentiation experiments). **h-k,** Whole organoid lineage tracing of WT:GFP-puro chimeric

organoids. Lineage size distribution in conditions indicated on the top (h, 80% WT: 20% GFP-puro, no puromycin; i, 80% WT: 20% GFP-puro, with puromycin; j, 90% WT: 10% GFP-puro, no puromycin; k, 90% WT: 10% GFP-puro, with puromycin. (n = individual lineages from 3 organoids as biological replicates from 1 differentiation experiment at each time point for each ± puromycin treatment). **l,** Schematic of optimal transport and foldchange calculation.

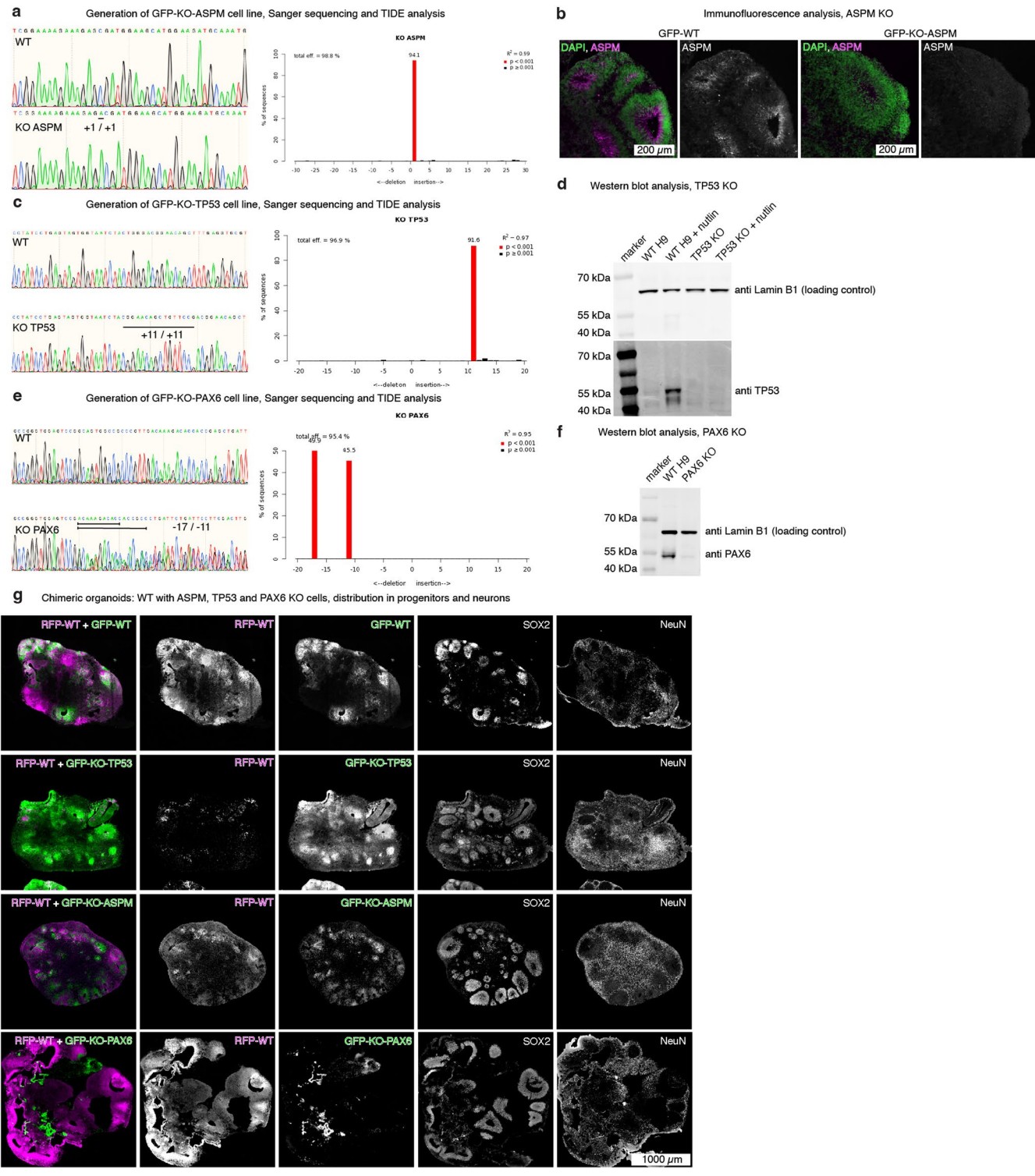

**Extended Data Fig. 8 | Generation GFP-KO cell lines and microscopic analyses of chimeric organoids. a-f**, Generation of GFP-KO-ASPM (a, b), GFP-KO-TP53 (c, d) and GFP-KO-PAX6 (e, f) cell lines. Sanger sequencing tracks on the left and TIDE analysis of genomic DNA (a, c, e). Protein expression analysis in (b, d, f). Immunoblotting was preferred for protein expression, however, ASPM antibody did not recognize blotted protein. **g**, Immunofluorescent images of chimeric organoids containing RFP-WT and indicated GFP-KO cells. RFP and GFP merged images on the left with individual signals shown as indicated. SOX2 and NeuN signal marks neural progenitors and neurons respectively.

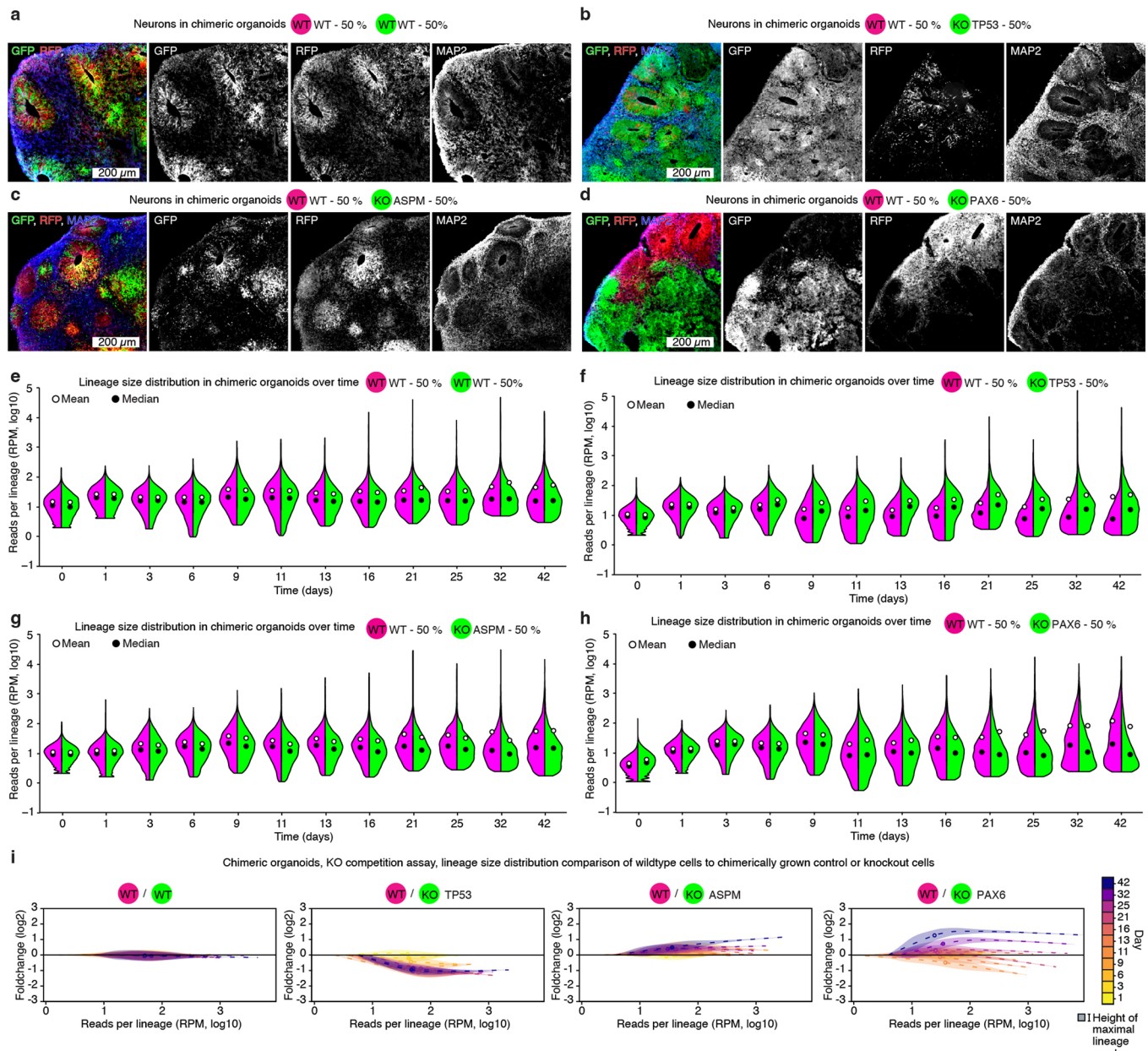

**Extended Data Fig. 9 | Replenishment in RFP-WT with GFP-KO chimeric organoids. a-d,** Immunofluorescent analysis of day 42 organoids chimeric for RFP-WT and GFP-WT (a), GFP-KO-TP53 (b), GFP-KO-ASPM (c) and GFP-KO-PAX6 (d). Labelling of fluorophores and neuronal marker MAP2 as indicated. **e-h,** Whole organoid lineage tracing of chimeric organoids containing 50% RFP-WT and 50% GFP-WT (e), GFP-KO-TP53 (f), GFP-KO ASPM (g) and GFP-KO-PAX6 (h) cells. (n = individual lineages from 3 organoids as biological replicates from 1 differentiation experiment at each time point). **i,** Optimal transport comparison of lineage populations of RFP-WT to chemically grown intra-organoid GFP-WT or selected GFP-KO cells over time. Colour code for time indicated at the right, thickness of curves indicates number of lineages of certain sizes. (n = individual lineages from 3 organoids as biological replicates from 1 differentiation experiment at each time point).

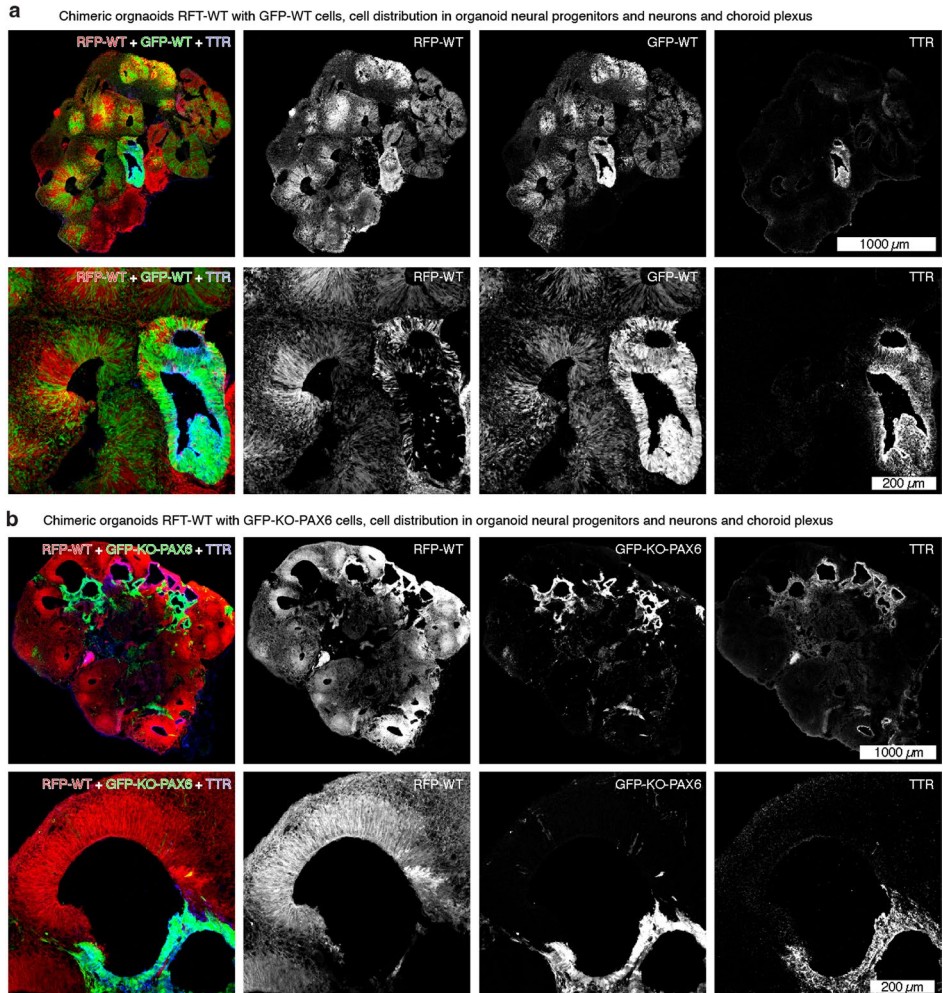

**Extended Data Fig. 10 | Immunofluorescent characterization of RFP-WT-GFP-WT and RFP-WT-GFP-KO-PAX6 chimeric organoids. a, b,** Immunofluorescence images of RFP and GFP signal alongside marker staining for choroid plexus tissue (TTR) in RFP-WT:GFP-WT (a) and RFP-WT:GFP-KO-PAX6 (b) chimeric organoids.

# Reporting Summary

## Statistics

For all statistical analyses, confirm that the following items are present in the figure legend, table legend, main text, or Methods section.

| n/a | Confirmed | |
|---|---|---|
| ☐ | ☒ | The exact sample size (*n*) for each experimental group/condition, given as a discrete number and unit of measurement |
| ☐ | ☒ | A statement on whether measurements were taken from distinct samples or whether the same sample was measured repeatedly |
| ☐ | ☒ | The statistical test(s) used AND whether they are one- or two-sided *Only common tests should be described solely by name; describe more complex techniques in the Methods section.* |
| ☒ | ☐ | A description of all covariates tested |
| ☒ | ☐ | A description of any assumptions or corrections, such as tests of normality and adjustment for multiple comparisons |
| ☒ | ☐ | A full description of the statistical parameters including central tendency (e.g. means) or other basic estimates (e.g. regression coefficient) AND variation (e.g. standard deviation) or associated estimates of uncertainty (e.g. confidence intervals) |
| ☐ | ☒ | For null hypothesis testing, the test statistic (e.g. *F*, *t*, *r*) with confidence intervals, effect sizes, degrees of freedom and *P* value noted *Give P values as exact values whenever suitable.* |
| ☒ | ☐ | For Bayesian analysis, information on the choice of priors and Markov chain Monte Carlo settings |
| ☒ | ☐ | For hierarchical and complex designs, identification of the appropriate level for tests and full reporting of outcomes |
| ☒ | ☐ | Estimates of effect sizes (e.g. Cohen's *d*, Pearson's *r*), indicating how they were calculated |

*Our web collection on statistics for biologists contains articles on many of the points above.*

## Software and code

Policy information about availability of computer code

| Data collection | Data was collected using Zeiss (LSM 780, 800, 880, Z1), Olympus (IX83) and Yokogawa (CSU-W1) microscopes and a slidescanner (3DHistech, Panoramic250) with the companies standard softwares. Flow cytometry was performed using BD Fortessa instruments running Diva (V9.0.1) software. Illumina sequencers (HiSeq and NovaSeq) were used for NGS experiments. Single cell RNA-seq experiments were performed using the 10x Genomics Chromium controller. |
|---|---|
| Data analysis | Data analysis was performed using Microsoft Excel (V. 16.83), Graphpad Prism (V.9 and newer), FlowJo, Seurat in R (V. 4.3.0.1.), Python (V. 3.9.), ImageJ (V. 1.5.3) and CellRanger (V. 7.01) . Code for stochastic modeling is available here: https://github.com/Cibiv/pyrganoid |

For manuscripts utilizing custom algorithms or software that are central to the research but not yet described in published literature, software must be made available to editors and reviewers. We strongly encourage code deposition in a community repository (e.g. GitHub). See the Nature Portfolio guidelines for submitting code & software for further information.

## Data

Policy information about availability of data

All manuscripts must include a data availability statement. This statement should provide the following information, where applicable:
- Accession codes, unique identifiers, or web links for publicly available datasets
- A description of any restrictions on data availability
- For clinical datasets or third party data, please ensure that the statement adheres to our policy

> NGS data used in this work is available under GEO accession no.: NCBI GSE214105, GSE151384. The human reference genome used was GRCh38 2020.

## Human research participants

Policy information about studies involving human research participants and Sex and Gender in Research.

| | |
|---|---|
| Reporting on sex and gender | Not applicable. |
| Population characteristics | Not applicable. |
| Recruitment | Not applicable. |
| Ethics oversight | Not applicable. |

Note that full information on the approval of the study protocol must also be provided in the manuscript.

# Field-specific reporting

Please select the one below that is the best fit for your research. If you are not sure, read the appropriate sections before making your selection.

☒ Life sciences    ☐ Behavioural & social sciences    ☐ Ecological, evolutionary & environmental sciences

For a reference copy of the document with all sections, see nature.com/documents/nr-reporting-summary-flat.pdf

# Life sciences study design

All studies must disclose on these points even when the disclosure is negative.

| | |
|---|---|
| Sample size | Samples sizes for microscopy, organoid size measurements and flow cytometry analysis and were chosen based on previous experience and literature. Lineage tracing studies were performed at least in triplicates for all timepoints in all conditions. This approach is in line with standards in the field (Lancaster et al, Nature Biotechnology, 2017; Bagley et al, Nature Methods, 2017; Esk et al, Science, 2020; Eichmüller et al, Science, 2020). |
| Data exclusions | 20 lineage tracing samples (out of 476) were discarded due to low / no PCR amplicon generation. In those cases duplicate (instead of planned triplicate) measurements were used. |
| Replication | At least two independent biological experiments were performed including multiple replicates for most experiments. Details are provided in figure legends. Exceptions include some whole organoid lineage tracing experiments, which were conducted only once with three individual organoids per timepoint and conditions and four mice per timepoint. Barcoded scRNA-seq experiments were performed from one batch with one organoid per 10X library in organoid duplicates per two timepoints. |
| Randomization | Individual organoids analyzed were randomly chosen from a running batch for each type of experiment. Randomization of subjects was not relevant in this study. |
| Blinding | The experimenters were not blinded. Analyses were performed predominantly using unbiased computers and results reported are mostly descriptive. |

# Reporting for specific materials, systems and methods

We require information from authors about some types of materials, experimental systems and methods used in many studies. Here, indicate whether each material, system or method listed is relevant to your study. If you are not sure if a list item applies to your research, read the appropriate section before selecting a response.

## Materials & experimental systems

| n/a | Involved in the study |
|-----|------------------------|
| ☐ | ☒ Antibodies |
| ☐ | ☒ Eukaryotic cell lines |
| ☒ | ☐ Palaeontology and archaeology |
| ☐ | ☒ Animals and other organisms |
| ☒ | ☐ Clinical data |
| ☒ | ☐ Dual use research of concern |

## Methods

| n/a | Involved in the study |
|-----|------------------------|
| ☒ | ☐ ChIP-seq |
| ☐ | ☒ Flow cytometry |
| ☒ | ☐ MRI-based neuroimaging |

# Antibodies

| | |
|---|---|
| Antibodies used | The following primary antibodies were used:<br><br>anti-SOX2 (Abcam, ab97959, 1:500),<br>anti-NEUN (Millipore, MAB377, 1:500),<br>anti-DCX (Santa Cruz, sc8066, 1:1000),<br>anti-MAP2 (Abcam, ab5392, 1:1000),<br>anti-Nestin (BD, 611658, 1:1000),<br>anti bIII-tubulin (Abcam, ab18207, 1:1000),<br>anti-BrdU (ThermoFisher, B35128, 1:500),<br>anti phospho-vimentin(Ser82)(MBL, D095-3S, 1:250),<br>anti-TTR (AbD Serotec, ahp1837, 1:500),<br>anti-aPKC (SantaCruz, sc-17781, 1:100),<br>anti nestin (Abcam, ab105389, 1:1000),<br>anti-b-catenin (Sigma, C2206, 1:250),<br>anti-n-cadherin (BD, 610920, 1:250),<br>anti-TBR1 (Abcam, ab31940, 1:500),<br>anti-TBR2 (Abcam, ab23345, 1:250),<br>anti-ASPM (Bethyl Laboratories, IHC-00058, 1:100),<br>anti-EOMES (R&D Sytems AF6166, 1:200),<br>chicken anti-GFP (Abcam Ab13970, 1:500),<br>mouse anti-Sox2 (Abcam Ab79351, 1:500),<br>rabbit anti-NeuroD2 (Abcam Ab104430, 1:500),<br>anti-LaminB1 (1:2000, GeneTex, cat #103292),<br>anti-PAX6 (1:1000, Biolegend, cat #901301),<br>anti-TP53 (1:1000, AbCam, #ab32389),<br>anti-NCAM (BD Biosciences, cat.: 564058, 1:500),<br>anti-CXCR4 (BD Biosciences, cat.: 560936, 1:250),<br>anti-TRA-1-60 (BD Biosciences, cat.: 563188, 1:250).<br><br>Secondary antibodies raised in donkey or goat were purchased from Invitrogen. If possible, all secondary antibodies were of highly cross-adsorbed quality.<br><br>Donkey anti-Mouse IgG (H+L), Alexa Fluor™ 488 (Thermo Fisher Scientific, cat. A-21202, 1:1000),<br>Donkey anti-Rabbit IgG (H+L), Alexa Fluor™ 488 (Thermo Fisher Scientific, cat. A-21206, 1:1000),<br>Donkey anti-Goat IgG (H+L), Alexa Fluor™ 488, (Thermo Fisher Scientific, cat. A-11055, 1:1000),<br>Donkey anti-Sheep IgG (H+L), Alexa Fluor™ 488 (Thermo Fisher Scientific, cat. A-11015, 1:1000),<br>Goat anti-Rabbit IgG (H+L), Alexa Fluor™ 488 (Thermo Fisher Scientific, cat. A-11008, 1:1000),<br>Goat anti-Mouse IgG (H+L), Alexa Fluor™ 488 (Thermo Fisher Scientific, cat. A-11001, 1:1000),<br>Goat anti-Mouse IgG1, Alexa Fluor™ 488 (Thermo Fisher Scientific, cat. A-21121, 1:1000),<br>Goat anti-Rabbit IgG (H+L), Alexa Fluor™ 568 (Thermo Fisher Scientific, cat. A-11011, 1:1000),<br>Goat anti-Mouse IgG (H+L), Alexa Fluor™ 568 (Thermo Fisher Scientific, cat. A-11004, 1:1000),<br>Donkey anti-Goat IgG (H+L), Alexa Fluor™ 568 (Thermo Fisher Scientific, cat. A-11057, 1:1000),<br>Donkey anti-Mouse IgG (H+L), Alexa Fluor™ 568 (Thermo Fisher Scientific, cat. A-11037, 1:1000),<br>Goat anti-Mouse IgG2b, Alexa Fluor™ 568 (Thermo Fisher Scientific, cat. A-21144, 1:1000),<br>Donkey anti-Chicken IgY (H+L), Alexa Fluor™ 568 (Thermo Fisher Scientific, cat. A-78950, 1:1000),<br>Donkey anti-Rabbit IgG (H+L), Alexa Fluor™ 647 (Thermo Fisher Scientific, cat. 31573, 1:1000),<br>Donkey anti-Goat IgG (H+L), Alexa Fluor™ 647 (Thermo Fisher Scientific, cat. 21447, 1:1000),<br>Goat anti-Rabbit IgG (H+L), Alexa Fluor™ 647 (Thermo Fisher Scientific, cat. 21245, 1:1000),<br>Goat anti-Rabbit IgG (H+L), Alexa Fluor™ 647 (Thermo Fisher Scientific, cat. 21244, 1:1000). |
| Validation | anti-SOX2 (Abcam, ab97959) has been validated by the company and referenced in 656 publications.<br>anti-NEUN (Millipore, MAB377) has been validated by the company and referenced in more than 100 publications.<br>anti-DCX (Santa Cruz, sc8066) has been validated by the company and referenced in 226 publications.<br>anti-MAP2 (Abcam, ab5392) has been validated by the company and referenced in 511 publications.<br>anti-Nestin (BD, 611658) has been validated by the company and referenced in 5 publications.<br>anti bIII-tubulin (Abcam, ab18207) has been validated by the company and referenced in 379 publications.<br>anti-BrdU (ThermoFisher, B35128) has been validated by the company and referenced in 60 publications.<br>anti phospho-vimentin(Ser82)(MBL, D095-3S) has been validated by the company and referenced in 10 publications. |

anti-TTR (AbD Serotec, ahp1837) has been validated by the company and referenced in 9 publications.
anti-aPKC (SantaCruz, sc-17781) has been validated by the company and referenced in 138 publications.
anti nestin (Abcam, ab105389) has been validated by the company and referenced in 40 publications.
anti-b-catenin (Sigma, C2206) has been validated by the company and referenced in 370 publications.
anti-n-cadherin (BD, 610920) has been validated by the company and referenced in 5 publications.
anti-TBR1 (Abcam, ab31940) has been validated by the company and referenced in 322 publications.
anti-TBR2 (Abcam, ab23345) has been validated by the company and referenced in 392 publications.
anti-ASPM (Bethyl Laboratories, IHC-00058) has been validated by the company and referenced in 7 publications.
anti-EOMES (R&D Sytems AF6166) has been validated by the company and referenced in 11publications.
chicken anti-GFP (Abcam Ab13970) has been validated by the company and referenced in 2531 publications.
mouse anti-Sox2 (Abcam Ab79351) has been validated by the company and referenced in 49 publications.
rabbit anti-NeuroD2 (Abcam Ab104430) has been validated by the company and referenced in 14 publications.
anti-LaminB1 (GeneTex, cat #103292) has been validated by the company and referenced in 39 publications.
anti-PAX6 (Biolegend, cat #901301) has been validated by the company and referenced in 308 publications.
anti-TP53 (AbCam, #ab32389) has been validated by the company and referenced in 53 publications.
anti-NCAM (BD Biosciences, cat.: 564058) has been validated by the company and referenced in 14 publications.
anti-CXCR4 (BD Biosciences, cat.: 560936) has been validated by the company and referenced in 5 publications.
anti-TRA-1-60 (BD Biosciences, cat.: 563188) has been validated by the company and referenced in 6 publications.

# Eukaryotic cell lines

Policy information about cell lines and Sex and Gender in Research

| | |
|---|---|
| Cell line source(s) | The HEK293 line was purchased from ATCC. The hESC line WA09 (H9) was obtained from WiCell. All other lines used in the study were generated in the lab. |
| Authentication | The hESC line WA09 (H9) obtained from WiCell was not independently authenticated. The line HEK293 purchased from ATCC was not independently authenticated. |
| Mycoplasma contamination | Cell cultures were routinely bi-monthly tested and confirmed negative for mycoplasma. |
| Commonly misidentified lines (See ICLAC register) | No commonly misidentified lines have been used in this study. |

# Animals and other research organisms

Policy information about studies involving animals; ARRIVE guidelines recommended for reporting animal research, and Sex and Gender in Research

| | |
|---|---|
| Laboratory animals | 10-12 week old timed pregnant C57BL/6J mice (The Jackson Laboratory) were used in this study. Mice were kept at 22C +/-1C, 55% +/- 5% humidity in a 14h light / 10h dark cycle under the care of the IMBA's animal care facility. |
| Wild animals | No wild animals were used in this study. |
| Reporting on sex | Mouse embryos analyzed at day E17.5 were not tested for sex. |
| Field-collected samples | No field collected samples were used in this study. |
| Ethics oversight | All experimental procedures were approved by the Austrian Federal Ministry of Education, Science and Research under the animal experiment license BMWF.66.015/0023.WF/3b/2017. 3R principles were followed and monitored by the IMBA Ethics and Biosafety department. |

Note that full information on the approval of the study protocol must also be provided in the manuscript.

# Flow Cytometry

## Plots

Confirm that:

☒ The axis labels state the marker and fluorochrome used (e.g. CD4-FITC).

☒ The axis scales are clearly visible. Include numbers along axes only for bottom left plot of group (a 'group' is an analysis of identical markers).

☐ All plots are contour plots with outliers or pseudocolor plots.

☒ A numerical value for number of cells or percentage (with statistics) is provided.

## Methodology

| | |
|---|---|
| Sample preparation | For flow cytometry analysis organoids were dissociated by incubating on a Thermo Shaker at 37 °C in a 10:1 mixture of Accutase (Sigma Aldrich, cat # A6954) and 10x Trypsin (Gibco, cat # 15400) to generate a single cell suspension, followed by filtering through a 35µm cell strainer. |

| Instrument | BD Fortessa |
| --- | --- |
| Software | FACS Diva |
| Cell population abundance | A minimum of 20000 single cells was analyzed for each condition at each timepoint. |
| Gating strategy | Single cells were gated using forward and side scatters. Amplifier settiings were chosen to clearly display negative and positive populations. Gating strategies are provided in the Extended Data. |

☒ Tick this box to confirm that a figure exemplifying the gating strategy is provided in the Supplementary Information.

