## [Peer Review File · Nature Cell Biology]

Peer Review Information

Journal: Nature Cell Biology

Manuscript Title: Cerebral organoids display dynamic clonal growth and tunable tissue replenishment

Corresponding author name(s): Professor Juergen Knoblich

Editorial Notes:

Reviewer Comments & Decisions:

Decision Letter, initial version:

Dear Jürgen,

Your manuscript, "Cerebral organoids display dynamic clonal growth with lineage replenishment", has now been seen by 3 referees, who are experts in cerebral organoids, tissue development (referee 1); cerebral organoids, barcoding and lineage tracing (referee 2); and human cerebral organoids, neurodevelopment (referee 3). As you will see from their comments (attached below) they find this work of potential interest, but have raised substantial concerns, which in our view would need to be addressed with considerable revisions before we can consider publication in Nature Cell Biology.

Nature Cell Biology editors discuss the referee reports in detail within the editorial team, including the chief editor, to identify key referee points that should be addressed with priority, and requests that are overruled as being beyond the scope of the current study. To guide the scope of the revisions, I have listed these points below. We are committed to providing a fair and constructive peer-review process, so please feel free to contact me if you would like to discuss any of the referee comments further.

In particular, it would be essential to:

(A) Provide single cell RNA-seq analyses of barcoded cell populations to address the following concerns by referee #1:

"Specifically, the main weakness of this study is that the major conclusions about the cell types generated, depleted, replenished or dominated are not supported by cell identity analysis at all. There is no evidence on which exact cell types are built, competed, lost and particularly restored during organoid generation, maintenance and regrowth...", "...All findings in this work would enormously benefit and warrant publication if barcoded cell populations would also been analyzed at the single cell RNA-Seq level. This requires identification of transcribed barcodes. With this added, specifically labeled cells are also attached with their cellular state and all statements can be faithfully tested".

"Abstract statement on symmetrically dividing cells (Line 35): "We find that these variations in lineage size distribution are driven by a subpopulation of lineages that retain symmetrically dividing cells". How are symmetrically dividing cells identified? How do authors conclude beyond correlation of simulation with lineage number and size that variable lineage size scale is contributed by those symmetrically dividing stem cells? A specific barcoded clone may contain stem cells, intermediate progenitors, neurons and more heterogeneous cell types at variable sizes and lineage relations that cannot be deduced by this method without knowing the cellular state of each barcoded cell".

"Symmetrical Vs. asymmetrical cell division (lines 136-138, 172-174). The authors do not demonstrate symmetrically (or asymmetrically) dividing cells. The computational simulation nicely suggests S-> A/N transition that and it is supported by the literature and data accumulated on organoids. But larger and less heterogeneous lineages are not necessarily symmetrically dividing neural stem cells. SOX2+PAX6+ cells for example can produce more SOX2+PAX6+ cells only, or SOX2+PAX6+ cells and TBR2+ cells, the latter which symmetrically divide as well to two TBR2+ cells that then again later divide symmetrically to give rise to two neurons. Again, RNA-Seq of barcoded cells is required for validation of cell types produced and assessing how powerful the method is".

"The authors do not provide full evidence that the original cell repertoire of organoids has been restored, but only the lineage size and number. Since cell type composition is not shown, it is not clear if cells at any point can restore what was depleted. The authors should provide single cell data of barcoded cells to show that cell types have been restored".

"The latter applies also to the defected cell line compensation experiments (line 237). Full restoration means that the organoid is fully compensated in cell composition, and the current experiments cannot answer this. Also, SOX2+ rosettes surrounded by neurons are not sufficient to prove of full repertoire compensation".

(B) Confirm the validity and robustness of your method, but also address further concerns regarding the methodological approaches and interpretations, as indicated by referees #1 and #2. We would advise you to address the concerns with experiments, whenever possible.

Referee #1 says:

"Barcode library recovery (Methods). The authors compare different samples / methods (in vivo and in vitro). Have they verified the barcode library coverage immediately after infection in each case? This is important to know that there is enough range of clones that can be generated at each method. Charts show that barcodes are in excess, but it will be good to confirm this by analyzing barcode recovery / loss after infection".

"Organoid reproducibility (Lines 94-95): This is quite interesting that two independent libraries produce nearly identical results in triplicates. Would the authors elaborate on how this is possible given the known heterogeneity among organoids under most derivation methods? Also, if they are that identical – would the authors explain the differences seen in lineage size distribution shown in Fig. 1f compared with that of Extended Data Fig. 3d? In general, confirming the validity and robustness of this method requires replication in different cell lines (preferably 2 iPSC lines)".

Referee #2:

"In the developing brain and in organoids alike, symmetrically and asymmetrically dividing neuroepithelial cells and radial glia are not the only progenitor populations. Intermediate progenitor cells (IPCs) can make up ~10% of cells in an organoid, depending on the time of sampling. IPCs arise from RG and perform a consumptive division to give rise to 2 neurons, following the differentiation pattern $S \rightarrow 2xN$ (borrowing the nomenclature from the manuscript). How does inclusion of these cells in your model affect its performance? Could this be used to explain differences between simulated and actual data? Please consider addressing this as a limitation in the text, or revise the model to include divisions expected from this cell type. Alternatively, if the authors stain for the IPC marker EOMES and find that, surprisingly, they make up a very small proportion of organoid cells, then these data can be added in the supplement to demonstrate that the model does not need to be revised".

"Excitatory neurons are historically particularly sensitive to dissociation. Are neurons selectively lost during dissociation using the authors' protocol? If so, clone sizes will be artificially low, especially those which contain proportionately more neurons. More importantly, estimates of proportion of symmetric to asymmetric division could be affected. The authors could strengthen their work by either demonstrating equivalent loss of all cell types during dissociation, or addressing this as a limitation of their study".

"Fig 1c+d and elsewhere: With this bulk method, how can the authors distinguish between a.) two lineages of the same cell count from b.) one smaller lineage with two or more integration sites of the same barcode? Relatedly, how can the authors distinguish between c.) one lineage which received two distinct barcodes and d.) two equally sized lineages, each with different barcodes? Conceptually, these questions need to be addressed. If these considerations are currently incorporated in the analysis pipelines, this should be stated clearly. Alternatively, these questions should be addressed as a major

caveat to the approach".

(C) Provide a more thorough description of the method and a more thorough discussion of the method applications and limitations, as indicated by referee #3:

"The technique of DNA barcoding used in this manuscript is interesting but complicated. It would help readers better understand this system if more description about technical details could be added. For example, why the authors used two DNA barcode libraries to label each starting cell, not one or three? Please add more explanation or related references".

"The manuscript is well-written but the discussion seems to be too short. This part would be more abundant and instructive if the limitations of this research and its application prospect are fully discussed".

(D) All other referee concerns pertaining to strengthening existing data, providing controls, methodological and statistical details, clarifications and textual changes, should also be addressed.

(F) Finally please pay close attention to our guidelines on statistical and methodological reporting (listed below) as failure to do so may delay the reconsideration of the revised manuscript. In particular please provide:

We would be happy to consider a revised manuscript that would satisfactorily address these points, unless a similar paper is published elsewhere, or is accepted for publication in Nature Cell Biology in the meantime.

In contrast, although we agree with referee #2 that the in vivo relevance of the findings here would provide valuable insights, we consider this point to be beyond the scope of the present study. Thus, while we would welcome such data, addressing this particular point experimentally will not be necessary for reconsideration of the manuscript at this journal and it would be sufficient if you discuss this issue within the manuscript, as suggested by the referee as an alternative.

- ensure that it conforms to our format instructions and publication policies (see below and <https://www.nature.com/nature/for-authors>).
- provide a point-by-point rebuttal to the full referee reports verbatim, as provided at the end of this letter.
- provide the completed Reporting Summary (found here <https://www.nature.com/documents/nr-reporting-summary.pdf>). This is essential for reconsideration of the manuscript will be available to editors and referees in the event of peer review. For more information see <http://www.nature.com/authors/policies/availability.html> or contact me.

When submitting the revised version of your manuscript, please pay close attention to our [href="https://www.nature.com/nature-portfolio/editorial-policies/image-integrity">Digital Image Integrity Guidelines](https://www.nature.com/nature-portfolio/editorial-policies/image-integrity). and to the following points below:

Nature Cell Biology is committed to improving transparency in authorship. As part of our efforts in this direction, we are now requesting that all authors identified as 'corresponding author' on published papers create and link their Open Researcher and Contributor Identifier (ORCID) with their account on the Manuscript Tracking System (MTS), prior to acceptance. ORCID helps the scientific community achieve unambiguous attribution of all scholarly contributions. You can create and link your ORCID from the home page of the MTS by clicking on 'Modify my Springer Nature account'. For more information please visit please visit www.springernature.com/orcid.

This journal strongly supports public availability of data. Please place the data used in your paper into a public data repository, or alternatively, present the data as Supplementary Information. If data can only be shared on request, please explain why in your Data Availability Statement, and also in the correspondence with your editor. Please note that for some data types, deposition in a public repository is mandatory - more information on our data deposition policies and available repositories appears below.

[Redacted]

*This url links to your confidential home page and associated information about manuscripts you may

have submitted or be reviewing for us. If you wish to forward this email to co-authors, please delete the link to your homepage.

We would like to receive a revised submission within six months.

We hope that you will find our referees' comments, and editorial guidance helpful. Please do not hesitate to contact me if there is anything you would like to discuss.

Best wishes,

Stelios

Stylianos Lefkopoulos, PhD
He/him/his
Associate Editor
Nature Cell Biology
Springer Nature
Heidelberger Platz 3, 14197 Berlin, Germany

E-mail: stylianos.lefkopoulos@springernature.com
Twitter: @s_lefkopoulos

Reviewers' Comments:

Reviewer #1:

Remarks to the Author:

Lindenhofer et al. devise a whole tissue lineage tracing method by genomic DNA barcoding in human cerebral organoids aiming to address limitations of imaging-based lineage tracing methodologies. The authors first introduce and analyze integrated barcodes into 2D and 3D cultures to show that while both cultures show stable lineage number over time, only 3D cultures exhibit significant clone size variation, implying that this system can trace the complex cell type diversity of the human cortex when mimicked in ES cell-based 3D cultures. Organoids composed of multiple fluorescently labelled ES cells combined at different proportions show maintained input percentages over time, indicating a lasting labelling of cell input. The authors apply their lineage tracing method to compare lineage number and size outcome in developing mouse brains in vivo, human 3D organoids and several 2D culture types, again to confirm that clonal heterogeneity in organoids is the highest, while 2D expanding and differentiating progenitor cells present poor lineage growth capacity. Based on decrease in organoid growth rate, divergence in lineage numbers and increase in neuronal differentiation markers after day 11, the authors suggest as a switch from symmetrically (S) to asymmetrically (A) dividing cells. Stochastic simulation of S, A and their neuronal progeny (N) predict limited lineage heterogeneity for S->S mode and increase in lineage size range for the S->A/N mode, fitting increased cell type diversity with differentiation. Authors correlate these findings in organoids showing more homogeneous and large lineage sizes until day 11 matching S->S mode, followed by greater heterogeneity in lineage size from day 13 to 40, more fitting the combined S->A/N simulated mode. Some rare symmetrically dividing stem cells are found in late organoid ventricles by immunostaining to support author computational-based conclusion that late-stage organoids are

composed of symmetrically dividing cells that remain and become rare throughout late organoid growth. Finally, the authors test the ability of minor cell populations in organoids to compensate for compromised organoid growth. They first show a proof concept of organoid entire regrowth and replenishment by a minority of puromycin resistant cells in mixed resistant and nonresistant cells. They then create genetic defects in fluorescently labeled cells and show that such defective cells can - depending on the mutation - either outgrow or be replenished by non-defected (and differently labeled) cells within organoids. They confirm this computationally by showing high contribution of the corresponding outgrowing/replenishing cell type to heterogeneous lineage sizes, and further support this by showing corresponding cell type domination within neural rosette cells (neural stem cells). Authors conclude that their method is able to assess clonal behavior of entire lineage populations, that organoid lineage heterogeneity relies on long lived symmetrically dividing cells, and that tissue replenishment can occur in plastic manner based on tissue needs and is not predetermined.

This study investigates a central question in stem cell biology: the capacity of clonal selection and replenishment of heterogeneous stem/progenitor cell types under normal in pathogenic conditions. Since this field of tracking and tracing cell types - particularly in 3D tissues - is still very challenging despite many years of work, the findings here are very interesting and provide new insights on the dynamic growth of heterogeneous tissues beyond brain organoid systems. With particular interest in cell type heterogeneity in 3D cultures, this method may hold promise to help addressing basic questions of dynamic tissue development, cellular growth and heterogeneity, and cell type homeostasis. But over all that, it seems to be a specifically strong tool to assess clonal domination events in health and disease.

However, the analyses in the current form of the study are general with a significant part of theoretical (the simulations) and computational (the lineage analysis) evidence and almost no investigation on cell type composition to elucidate the cell type nature of this dynamic growth or regrowth in organoids. Specifically, the main weakness of this study is that the major conclusions about the cell types generated, depleted, replenished or dominated are not supported by cell identity analysis at all. There is no evidence on which exact cell types are built, competed, lost and particularly restored during organoid generation, maintenance and regrowth. This leaves any statements on replenishment ability of entire organoid cell repertoire by minor stem cell populations poorly substantiated and narrows it down to organoid growth restoration only. In particular, the statements on the existence and dynamics of symmetrically dividing stem cells is one major aspect that needs to be addressed. The existence of such cells in this study is based on simulation and tracing lineage size over time, showing that bigger size lineages are progressively depleted. It is indeed established that early neural stem cells are symmetrically dividing at first and later transition to a smaller size pool of asymmetrically dividing stem/progenitor cells. But a proof for existence of rare symmetrical division is not trivial. Not only neural stem cells but also intermediate progenitors or transit amplifying like cells divide symmetrically and produce bursts of larger pools that later produce neurons and can be misdetected here as symmetrically dividing neural stem cells. In addition, basal radial glia consisting of a large portion of neurogenic neural stem cells during cortex expansion divide as well to produce themselves and/or neurons. So, the question of what is compensated here, only the growth - or also the entire cell type composition, remains open. The genetic defects introduced further underscore this, as any defect that can either be compensated for cell growth or that presents proliferative advantage needs to be assessed for the types of cells recovered. Are we getting the same organoid quality that was lost? The immunostaining shown is on partial organoids and cannot be comprehensive in its nature (significantly more full organoid image analyses and replicas are needed).

All findings in this work would enormously benefit and warrant publication if barcoded cell populations would also been analyzed at the single cell RNA-Seq level. This requires identification of

transcribed barcodes. With this added, specifically labeled cells are also attached with their cellular state and all statements can be faithfully tested. If this cannot be done, the work as much as novel in its essence, is more suitable as a method paper that provides important learning of the dynamics of cellular growth in general but not concrete realization of stem cell types, their properties, and regenerative potential.

Major points:

1. Abstract statement on symmetrically dividing cells (Line 35): "We find that these variations in lineage size distribution are driven by a subpopulation of lineages that retain symmetrically dividing cells". How are symmetrically dividing cells identified? How do authors conclude beyond correlation of simulation with lineage number and size that variable lineage size scale is contributed by those symmetrically dividing stem cells? A specific barcoded clone may contain stem cells, intermediate progenitors, neurons and more heterogeneous cell types at variable sizes and lineage relations that cannot be deduced by this method without knowing the cellular state of each barcoded cell.
2. Tissue replenishment / regrowth. (Line 36): "...stem cell output is tuneableis completely compensated for by unaffected lineages". How do authors conclude that the same cell type composition is achieved after ablation besides immunostaining of (partially shown) neural rosettes?
3. Barcode library recovery (Methods). The authors compare different samples / methods (in vivo and in vitro). Have they verified the barcode library coverage immediately after infection in each case? This is important to know that there is enough range of clones that can be generated at each method. Charts show that barcodes are in excess, but it will be good to confirm this by analyzing barcode recovery / loss after infection.
4. 2D culture lineage heterogeneity (Lines 82, 91-2, 118): Authors conclude that 2D cultures possess fewer and larger clones and thus are less heterogeneous compared to organoids. This conclusion is misleading. 2D neural stem/progenitor cultures are proliferative due to constant addition of mitogens, and this is the reason for the low diversity compared to organoids, which at any time reflect the total cumulative lineage range (as indicated in line 128-9, and hence the big advantage of organoids). In that sense, 2D neural stem cells are not that different from symmetrically dividing ESCs when assessed by the proposed lineage tracing method. Same for differentiating neurons – as they will reflect the differentiation competence of that specific time point when mitogens were withdrawn. This should be explained well in the text. It will be interesting to see whether multiple replicas of barcoded 2D cultures assessed for lineage size distribution at different times points of differentiation show together a larger cumulative lineage range.
5. Organoid reproducibility (Lines 94-95): This is quite interesting that two independent libraries produce nearly identical results in triplicates. Would the authors elaborate on how this is possible given the known heterogeneity among organoids under most derivation methods? Also, if they are that identical – would the authors explain the differences seen in lineage size distribution shown in Fig. 1f compared with that of Extended Data Fig. 3d? In general, confirming the validity and robustness of this method requires replication in different cell lines (preferably 2 iPSC lines).
6. Symmetrical Vs. asymmetrical cell division (lines 136-138, 172-174). The authors do not

demonstrate symmetrically (or asymmetrically) dividing cells. The computational simulation nicely suggests S-> A/N transition that and it is supported by the literature and data accumulated on organoids. But larger and less heterogeneous lineages are not necessarily symmetrically dividing neural stem cells. SOX2+PAX6+ cells for example can produce more SOX2+PAX6+ cells only, or SOX2+PAX6+ cells and TBR2+ cells, the latter which symmetrically divide as well to two TBR2+ cells that then again later divide symmetrically to give rise to two neurons. Again, RNA-Seq of barcoded cells is required for validation of cell types produced and assessing how powerful the method is.

7. Tissue replenishment (line 195-6, 218).

a. The authors do not provide full evidence that the original cell repertoire of organoids has been restored, but only the lineage size and number. Since cell type composition is not shown, it is not clear if cells at any point can restore what was depleted. The authors should provide single cell data of barcoded cells to show that cell types have been restored.

8. The latter applies also to the defected cell line compensation experiments (line 237). Full restoration means that the organoid is fully compensated in cell composition, and the current experiments cannot answer this. Also, SOX2+ rosettes surrounded by neurons are not sufficient to prove of full repertoire compensation.

Reviewer #2:

Remarks to the Author:

Review of: Cerebral organoids display dynamic clonal growth with lineage replacement

By: Lindenhofer et al 2022

Overall recommendation:

Cerebral organoids have become a staple model system for understanding human brain development and modeling disease. Here, Lindenhofer et al characterize the rules of clonal expansion of stem cells in the context of cerebral organoids, modeling symmetric and asymmetric divisions also observed in development. They utilize a barcoded lineage tracing strategy alongside a whole-culture or whole-organoid pooled sequencing readout to assess clonal dynamics. The authors claim that progressive loss of lineages demonstrates selective pressures during organoid growth. Additionally, using chimeric organoids, drug-based selection demonstrated adaptive lineage compensation. Finally, genetic perturbations to endogenous loci are shown to affect lineages and specification. The results included in this manuscript are on the whole thorough and appropriately analyzed. Further validation of organoid identity, a more thoughtful analysis of barcodes, and some concrete connection to in vivo work would strengthen the conclusions of the paper. However, the clever use of chimeric organoids, and particularly the functional assays presented in figures 3 and 4 are novel and contribute to our understanding of organoids as a model. As the authors point out, in the future, it would be very exciting to leverage these approaches in order to understand species differences in neural stem cell clonal expansion.

Major comments:

While organoids are a commonly utilized model, whether the results demonstrated in this paper could

be recapitulated in vivo is unknown. Some stronger connection to the in vivo setting would significantly strengthen the significance of the findings in the present manuscript. Alternatively, please address this as a caveat.

In the developing brain and in organoids alike, symmetrically and asymmetrically dividing neuroepithelial cells and radial glia are not the only progenitor populations. Intermediate progenitor cells (IPCs) can make up ~10% of cells in an organoid, depending on the time of sampling. IPCs arise from RG and perform a consumptive division to give rise to 2 neurons, following the differentiation pattern $S \rightarrow 2 \times N$ (borrowing the nomenclature from the manuscript). How does inclusion of these cells in your model affect its performance? Could this be used to explain differences between simulated and actual data? Please consider addressing this as a limitation in the text, or revise the model to include divisions expected from this cell type. Alternatively, if the authors stain for the IPC marker EOMES and find that, surprisingly, they make up a very small proportion of organoid cells, then these data can be added in the supplement to demonstrate that the model does not need to be revised.

Excitatory neurons are historically particularly sensitive to dissociation. Are neurons selectively lost during dissociation using the authors' protocol? If so, clone sizes will be artificially low, especially those which contain proportionately more neurons. More importantly, estimates of proportion of symmetric to asymmetric division could be affected. The authors could strengthen their work by either demonstrating equivalent loss of all cell types during dissociation, or addressing this as a limitation of their study.

Fig 1c+d and elsewhere: With this bulk method, how can the authors distinguish between a.) two lineages of the same cell count from b.) one smaller lineage with two or more integration sites of the same barcode? Relatedly, how can the authors distinguish between c.) one lineage which received two distinct barcodes and d.) two equally sized lineages, each with different barcodes? Conceptually, these questions need to be addressed. If these considerations are currently incorporated in the analysis pipelines, this should be stated clearly. Alternatively, these questions should be addressed as a major caveat to the approach.

Variability in cell type makeup within and across organoids could affect cell division modes, however this is not characterized formally in your study. How many replicates were stained for quality control? How much non-brain-like tissue is included in the organoids? Organoids are known to be highly heterogeneous, batch-dependent, and line-dependent.

Minor issues:

How was cell number measured in figure 2a? Consider including some details in the caption.

How were the coefficients for different types of divisions selected for your model (shown in Fig 2f)? Please include rationale.

Consider validating the knockout lines used late in the paper by qPCR or IHC to further demonstrate functional (not just genomic sequence-based) knockout as is displayed in Extended figure 8.

In methods and/or in extended figure 6 caption, please include the concentration of BrdU added.

It is not clear whether each "replicate" or "sample" (such as those reported in Extended Fig 3e, f, for

example) pertains to one organoid or one batch of organoids. This should be clarified.

Extended data 1f and 10 figures could be included in main figures as they help readers understand the experiments being performed, whereas some details in figure 2 and 3 could be moved to the extended figures, if desired.

Reviewer #3:

Remarks to the Author:

In this study, Dominik Lindenhofer et al. utilized a genomic DNA barcoding technique to perform whole-tissue lineage tracing on human organoids. In addition, they found that a group of stem cells retaining symmetrically dividing could lead to a great lineage size heterogeneity during the development process of organoids. Furthermore, the growth of neural stem cell lineage was plastic and possessed the self-healing ability when encountering cell-loss induced by either chemical and genetic ablation. Overall, these findings are highly innovative and intriguing, providing a platform for studying different clonal lineages in 3D organoids. There are a few aspects need further consideration for additional merits.

1. The technique of DNA barcoding used in this manuscript is interesting but complicated. It would help readers better understand this system if more description about technical details could be added. For example, why the authors used two DNA barcode libraries to label each starting cell, not one or three? Please add more explanation or related references.

2. In extended. Data Fig. 2e-g, the developing mice were injected with DNA-barcode virus in three timepoints and sampled at E18.5. What is the infection efficiency of the virus at the sampling time? The conclusion of smaller clonal heterogeneity in mouse seemed not be very convincing, since human organoids and mouse embryos were infected at different timepoints and with different infection approaches. A clarification is necessary.

3. In Extended Data Fig. 3, the authors tried to demonstrate that tissue context affects clonal lineage growth. Although there was a tendency of more divergent distribution in 3D organoids, statistic calculation was not found to support this conclusion? It would be more convincing if some statistical tests were added.

4. In Extended Data Fig. 6, the organoids were labeled with BrdU at day 40. But it seemed not suitable to define Day 40 organoids as late organoids. Please label organoids with BrdU at Day 80 or later.

5. In Fig. 4b and Fig. 4c, how many organoids did each point represent in each figure? Please add the number of analyzed organoids in the figure legends. In addition, the error bars of each figure are necessary.

6. In Fig.4d, the labels of SOX2 and NeuN are all red. Please label them with colors consistent with immunofluorescence images.

7. The manuscript is well-written but the discussion seems to be too short. This part would be more abundant and instructive if the limitations of this research and its application prospect are fully

discussed.

FINANCIAL AND NON-FINANCIAL COMPETING INTERESTS – the authors must include one of three

declarations: (1) that they have no financial and non-financial competing interests; (2) that they have financial and non-financial competing interests; or (3) that they decline to respond, after the Author Contributions section. This statement will be published with the article, and in cases where financial and non-financial competing interests are declared, these will be itemized in a web supplement to the article. For further details please see <https://www.nature.com/licenceforms/nrg/competing-interests.pdf>.

Methods should be written concisely, but should contain all elements necessary to allow interpretation and replication of the results. As a guideline, Methods sections typically do not exceed 3,000 words. The Methods should be divided into subsections listing reagents and techniques. When citing previous methods, accurate references should be provided and any alterations should be noted. Information must be provided about: antibody dilutions, company names, catalogue numbers and clone numbers for monoclonal antibodies; sequences of RNAi and cDNA probes/primers or company names and catalogue numbers if reagents are commercial; cell line names, sources and information on cell line identity and authentication. Animal studies and experiments involving human subjects must be reported in detail, identifying the committees approving the protocols. For studies involving human subjects/samples, a statement must be included confirming that informed consent was obtained. Statistical analyses and information on the reproducibility of experimental results should be provided in a section titled "Statistics and Reproducibility".

All Nature Cell Biology manuscripts submitted on or after March 21 2016 must include a Data availability statement as a separate section after Methods but before references, under the heading "Data Availability". For Springer Nature policies on data availability see <http://www.nature.com/authors/policies/availability.html>; for more information on this particular policy see <http://www.nature.com/authors/policies/data/data-availability-statements-data-citations.pdf>. The Data availability statement should include:

- Accession codes for primary datasets (generated during the study under consideration and designated as "primary accessions") and secondary datasets (published datasets reanalysed during the study under consideration, designated as "referenced accessions"). For primary accessions data should be made public to coincide with publication of the manuscript. A list of data types for which submission to community-endorsed public repositories is mandated (including sequence, structure, microarray, deep sequencing data) can be found here <http://www.nature.com/authors/policies/availability.html#data>.

- Unique identifiers (accession codes, DOIs or other unique persistent identifier) and hyperlinks for datasets deposited in an approved repository, but for which data deposition is not mandated (see here for details <http://www.nature.com/sdata/data-policies/repositories>).
- At a minimum, please include a statement confirming that all relevant data are available from the authors, and/or are included with the manuscript (e.g. as source data or supplementary information), listing which data are included (e.g. by figure panels and data types) and mentioning any restrictions on availability.
- If a dataset has a Digital Object Identifier (DOI) as its unique identifier, we strongly encourage including this in the Reference list and citing the dataset in the Methods.

We recommend that you upload the step-by-step protocols used in this manuscript to the Protocol Exchange. More details can found at www.nature.com/protocolexchange/about.

All imaging data should be accompanied by scale bars, which should be defined in the legend. Cropped images of gels/blots are acceptable, but need to be accompanied by size markers, and to retain visible background signal within the linear range (i.e. should not be saturated). The boundaries of panels with low background have to be demarked with black lines. Splicing of panels should only be considered if unavoidable, and must be clearly marked on the figure, and noted in the legend with a statement on whether the samples were obtained and processed simultaneously. Quantitative comparisons between samples on different gels/blots are discouraged; if this is unavoidable, it should only be performed for samples derived from the same experiment with gels/blots were processed in parallel, which needs to be stated in the legend.

- For line art, graphs, charts and schematics we prefer Adobe Illustrator (.AI), Encapsulated PostScript (.EPS) or Portable Document Format (.PDF). Files should be saved or exported as such directly from the application in which they were made, to allow us to restyle them according to our journal house style.
- We accept PowerPoint (.PPT) files if they are fully editable. However, please refrain from adding PowerPoint graphical effects to objects, as this results in them outputting poor quality raster art. Text used for PowerPoint figures should be Helvetica (preferred) or Arial.
- We do not recommend using Adobe Photoshop for designing figures, but we can accept Photoshop generated (.PSD or .TIFF) files only if each element included in the figure (text, labels, pictures, graphs, arrows and scale bars) are on separate layers. All text should be editable in 'type layers' and line-art such as graphs and other simple schematics should be preserved and embedded within 'vector smart objects' - not flattened raster/bitmap graphics.
- Some programs can generate Postscript by 'printing to file' (found in the Print dialogue). If using an application not listed above, save the file in PostScript format or email our Art Editor, Allen Beattie for advice (a.beattie@nature.com).

The total number of Supplementary Figures (not including the “unprocessed scans” Supplementary Figure) should not exceed the number of main display items (figures and/or tables (see our Guide to Authors and March 2012 editorial <http://www.nature.com/ncb/authors/submit/index.html#suppinfo>; <http://www.nature.com/ncb/journal/v14/n3/index.html#ed>). No restrictions apply to Supplementary Tables or Videos, but we advise authors to be selective in including supplemental data.

GUIDELINES FOR EXPERIMENTAL AND STATISTICAL REPORTING

REPORTING REQUIREMENTS – We are trying to improve the quality of methods and statistics reporting in our papers. To that end, we are now asking authors to complete a reporting summary that collects information on experimental design and reagents. The Reporting Summary can be found here <https://www.nature.com/documents/nr-reporting-summary.pdf>) If you would like to reference the guidance text as you complete the template, please access these flattened versions at <http://www.nature.com/authors/policies/availability.html>.

Information on how many times each experiment was repeated independently with similar results

needs to be provided in the legends and/or Methods for all experiments, and in particular wherever representative experiments are shown.

Author Rebuttal to Initial comments

Response to reviewers (NCB-A49497A, Lindenhofer et al.)

We thank the reviewers for their detailed and thorough examination of our manuscript, their valuable suggestion and criticism. In response to the reviewers' comments we have made substantial additions that support the key findings of our manuscript. This includes **additional experimental evidence for long-term maintenance of symmetrically dividing cells** in cerebral organoids, **updated and extended theoretical modelling** of whole organoid growth as well as several **control experiments** throughout the manuscript. The main concern of reviewers was whether symmetrically dividing cells exist in late organoids and we have **dedicated a new main figure** (figure 3, with new Extended Data figure 6) to address this question.

Please find below a summary of all changes that were made to figures as well as a point-by-point responses to all the reviewers' comments. Reviewers' comments are in black and our responses are in blue.

Main figures:

1. Figure 1: previous Figure 1.
2. Figure 2: previous Figure 2 with replaced panels from the improved model in 2c, d, f, h, i.
3. Figure 3: new Figure: cell type composition of lineages by barcoded lineage tracing scRNA-seq and live fate mapping of dividing radial glia showing abundant symmetric divisions.
4. Figure 4: previous Figure 3.
5. Figure 5: previous Figure 4.

Extended Data Figures:

1. Extended Data Figure 1: previous Extended Data figure 1 with added panel on library barcode diversity.
2. Extended Data Figure 2: previous Extended Data Figure 2.
3. Extended Data Figure 3: previous Extended Data Figure 3 with added panel for statistical comparison of different culture conditions.
4. Extended Data Figure 4: previous Extended Data Figure 4.
5. Extended Data Figure 5: replaces previous Extended Data Figure 5. Previously two model variants were shown but now replaced by 15 variations of our improved model (see below).
6. Extended Data Figure 6: new figure: details on barcoded scRNA-seq experiments to determine cell type composition of individual lineages and cell fate mapping details.
7. Extended Data Figure 7: previous Extended Data Figure 6 with added panels for cell type characterization (IF and qPCR) of replenished organoid tissue.
8. Extended Data Figure 8: previous Extended Data Figure 7 with added controls showing loss of protein in KO cell lines generated (ASPM, TP53, PAX6).

9. Extended Data Figure 9: previous Extended Data figure 8 with added panels for cell type characterization of compensating organoid tissue.
10. Extended Data Figure 10: previous Extended Data Figure 9.

Please note that during the review process review we had additional correspondence with reviewer 1 to clarify some of his / her suggestions. This correspondence was moderated by the handling NCB editor, Stylianos Lefkopoulos. We copy the correspondence (the handling editor / reviewer letters in black, our letters in green) below reviewer 1's initial comments since the initial comments were further detailed and we now reply with additional experimental evidence to the correspondence.

Please find our point-by-responses to the reviewer's comments below.

Reviewers' Comments:

Reviewer #1:

Remarks to the Author:

Lindenhofer et al. devise a whole tissue lineage tracing method by genomic DNA barcoding in human cerebral organoids aiming to address limitations of imaging-based lineage tracing methodologies. The authors first introduce and analyze integrated barcodes into 2D and 3D cultures to show that while both cultures show stable lineage number over time, only 3D cultures exhibit significant clone size variation, implying that this system can trace the complex cell type diversity of the human cortex when mimicked in ES cell-based 3D cultures. Organoids composed of multiple fluorescently labelled ES cells combined at different proportions show maintained input percentages over time, indicating a lasting labelling of cell input. The authors apply their lineage tracing method to compare lineage number and size outcome in developing mouse brains in vivo, human 3D organoids and several 2D culture types, again to confirm that clonal heterogeneity in organoids is the highest, while 2D expanding and differentiating progenitor cells present poor lineage growth capacity. Based on decrease in organoid growth rate, divergence in lineage numbers and increase in neuronal differentiation markers after day 11, the authors suggest a switch from symmetrically (S) to asymmetrically (A) dividing cells. Stochastic simulation of S, A and their neuronal progeny (N) predict limited lineage heterogeneity for S->S mode and increase in lineage size range for the S->A/N mode, fitting increased cell type diversity with differentiation. Authors correlate these findings in organoids showing more homogeneous and large lineage sizes until day 11 matching S->S mode, followed by greater heterogeneity in lineage size from day 13 to 40, more fitting the combined S->A/N simulated mode. Some rare symmetrically dividing stem cells are found in late organoid ventricles by immunostaining to support author computational-based conclusion that late-stage organoids are composed of symmetrically dividing cells that remain and become rare throughout late organoid growth. Finally, the authors test the ability of minor cell populations in organoids to compensate for compromised organoid growth. They first show a proof concept of organoid entire regrowth and replenishment by a minority of puromycin resistant cells in mixed resistant and nonresistant cells. They then create genetic defects in

fluorescently labeled cells and show that such defective cells can - depending on the mutation - either outgrow or be replenished by non-defected (and differently labeled) cells within organoids. They confirm this computationally by showing high contribution of the corresponding outgrowing/replenishing cell type to heterogeneous lineage sizes, and further support this by showing corresponding cell type domination within neural rosette cells (neural stem cells). Authors conclude that their method is able to assess clonal behavior of entire lineage populations, that organoid lineage heterogeneity relies on long lived symmetrically dividing cells, and that tissue replenishment can occur in plastic manner based on tissue needs and is not predetermined.

This study investigates a central question in stem cell biology: the capacity of clonal selection and replenishment of heterogeneous stem/progenitor cell types under normal in pathogenic conditions. Since this field of tracking and tracing cell types - particularly in 3D tissues - is still very challenging despite many years of work, the findings here are very interesting and provide new insights on the dynamic growth of heterogeneous tissues beyond brain organoid systems. With particular interest in cell type heterogeneity in 3D cultures, this method may hold promise to help addressing basic questions of dynamic tissue development, cellular growth and heterogeneity, and cell type homeostasis. But over all that, it seems to be a specifically strong tool to assess clonal domination events in health and disease.

We thank the reviewer for his / her interest in our work and pointing out useful future applications.

However, the analyses in the current form of the study are general with a significant part of theoretical (the simulations) and computational (the lineage analysis perse) evidence and almost no investigation on cell type composition to elucidate the cell type nature of this dynamic growth or re-growth in organoids. Specifically, the main weakness of this study is that the major conclusions about the cell types generated, depleted, replenished or dominated are not supported by cell identity analysis at all. There is no evidence on which exact cell types are built, competed, lost and particularly restored during organoid generation, maintenance and regrowth. This leaves any statements on replenishment ability of entire organoid cell repertoire by minor stem cell populations poorly substantiated and narrows it down to organoid growth restoration only. In particular, the statements on the existence and dynamics of symmetrically dividing stem cells is one major aspect that needs to be addressed. The existence of such cells in this study is based on simulation and tracing lineage size over time, showing that bigger size lineages are progressively depleted. It is indeed established that early neural stem cells are symmetrically dividing at first and later transition to a smaller size pool of asymmetrically dividing stem/progenitor cells. But a proof for existence of rare symmetrical division is not trivial. Not only neural stem cells but also intermediate progenitors or transit amplifying like cells divide symmetrically and produce bursts of larger pools that later produce neurons and can be mis detected here as symmetrically dividing neural stem cells. In addition, basal radial glia consisting of a large portion of neurogenetic neural stem cells during cortex expansion divide as well to produce themselves and/or neurons. So, the question of what is compensated here, only the growth - or also the entire cell type composition, remains open. The genetic defects introduced further underscore this, as any defect that can either be compensated for cell growth or that presents proliferative advantage needs to be assessed

for the types of cells recovered. Are we getting the same organoid quality that was lost? The immunostaining shown is on partial organoids and cannot be comprehensive in its nature (significantly more full organoid image analyses and replicas are needed). All findings in this work would enormously benefit and warrant publication if barcoded cell populations would also been analyzed at the single cell RNA-Seq level. This requires identification of transcribed barcodes. With this added, specifically labeled cells are also attached with their cellular state and all statements can be faithfully tested. If this cannot be done, the work as much as novel in its essence, is more suitable as a method paper that provides important learning of the dynamics of cellular growth in general but not concrete realization of stem cell types, their properties, and regenerative potential.

We thank the reviewer for his / her detailed criticisms. We have clarified these comments, added barcoded scRNA-seq data on wildtype organoids at two timepoints (Day 18 and Day 42) at the reviewer's suggestion, added live cell imaging coupled with fate mapping analyses for dividing progenitors and analyzed replenished tissue by immunohistochemistry and qPCR. This strengthens our key finding, the presence of symmetrically dividing neural progenitor cells in organoids that persist throughout the organoid differentiation protocol highlighting the biological implication of this study. In further correspondence with the reviewer and editor it was concluded that barcoded scRNA-seq of all experimental conditions are outside the scope of this paper. Instead, we added microscopy and qPCR data to address the question on the quality of replenished tissue on a lineage-independent level.

Major points:

1. Abstract statement on symmetrically dividing cells (Line 35): "We find that these variations in lineage size distribution are driven by a subpopulation of lineages that retain symmetrically dividing cells". How are symmetrically dividing cells identified? How do authors conclude beyond correlation of simulation with lineage number and size that variable lineage size scale is contributed by those symmetrically dividing stem cells? A specific barcoded clone may contain stem cells, intermediate progenitors, neurons and more heterogeneous cell types at variable sizes and lineage relations that cannot be deduced by this method without knowing the cellular state of each barcoded cell.

We thank the reviewer for his/her comment. We agree that clones contain many different cell types. We now include substantial additional data regarding cell type composition of clones, fate mapping of cell divisions via live-cell imaging in slice cultured organoids coupled with fate mapping and additional microscopy images as well as clarifications in the text to address these issues. Notably we can show the presence of symmetrically dividing cells in these organoids via live cell imaging and our modeling approaches only produce a lineage size distribution that is similar to the measured data if S cells are present throughout the differentiation protocol. Although multiple different cell types are present within an individual lineage, only radial glia progenitors retain the ability to divide symmetrically at this stage of organoid differentiation. Intermediate progenitors produce a finite number of neurons. In our improved version of the model, we show that modifying the output of A cells to include division behavior of intermediate progenitors does not alter our conclusion that a persistent S-cell population needs to be

present in order to generate the heterogeneous lineage size distribution. We want to thank the reviewer again for this important comment, the influence of intermediate progenitors on lineage size distribution was also raised by reviewer 2.

2. Tissue replenishment / regrowth. (Line 36): "...stem cell output is tuneable ...is completely compensated for by unaffected lineages". How do authors conclude that the same cell type composition is achieved after ablation besides immunostaining of (partially shown) neural rosettes?

We thank the reviewer for his/her comment, the respective sentence in the abstract might be misleading which was not intentional. In the original submission, we claim in the main text that replenishment is complete in terms of organoid cell number/organoid size, we do not claim this for quality, i.e. cell types. We found examples of conditions in which compensation appears largely similar also in cell type quality (i.e. puro chimeras) as well as conditions, in which cell type composition is altered (but not organoid size). The WT:PAX6-KO chimeric organoids are the most prominent example. To clarify this better in the abstract we modified the referenced sentence to: "We show that lineage sizes can adjust to tissue demands after growth perturbation via chemical ablation or genetic restriction of a subset of cells in chimeric organoids."

However, at the reviewer's request we have substantiated our analysis of organoids after compensatory growth. In addition to general markers for radial glia progenitors (Sox2) and neurons (NeuN) we now show an additional neuronal marker (Map2), a marker of dorsal excitatory neurons (Tbr1) as well as intermediate progenitors (Tbr2). All of these are expressed in both replenished and unaffected tissue. Additionally, we provide a comprehensive qPCR analysis quantifying various cell fate markers in chimeric puromycin resistant organoids of various replenishment states. At the time of full replenishment (day 42) samples are similar to each other indicating that tissue is also largely qualitatively replenished.

3. Barcode library recovery (Methods). The authors compare different samples / methods (in vivo and in vitro). Have they verified the barcode library coverage immediately after infection in each case? This is important to know that there is enough range of clones that can be generated at each method. Charts show that barcodes are in excess, but it will be good to confirm this by analyzing barcode recovery / loss after infection.

Having a large barcode variability in excess of labelled cells is crucial for barcoded lineage tracing. Therefore, this is an important point. Our theoretical barcode complexity is $2.81 \cdot 10^{14}$. We have previously generated complex barcode libraries (Esk et al, Science, 2020) and applied the same cloning and quality control strategies. Hamming distance analysis between replicates demonstrate the large barcode diversity (new Extended Data Fig. 1b). Therefore, we are confident that our experiments are using sufficient barcode complexities. A related question regarding potential double infection of cells has been raised by reviewer 2 and has been addressed in the corresponding section.

4. 2D culture lineage heterogeneity (Lines 82, 91-2, 118): Authors conclude that 2D cultures possess fewer and larger clones and thus are less heterogeneous compared to organoids. This conclusion is misleading. 2D neural stem/progenitor cultures are proliferative due to

constant addition of mitogens, and this is the reason for the low diversity compared to organoids, which at any time reflect the total cumulative lineage range (as indicated in line 128-9, and hence the big advantage of organoids). In that sense, 2D neural stem cells are not that different from symmetrically dividing ESCs when assessed by the proposed lineage tracing method. Same for differentiating neurons – as they will reflect the differentiation competence of that specific time point when mitogens were withdrawn. This should be explained well in the text. It will be interesting to see whether multiple replicas of barcoded 2D cultures assessed for lineage size distribution at different times points of differentiation show together a larger cumulative lineage range.

We would like to provide some clarifications to resolve this point. In both 2D (ESCs and 2D neuronal differentiation protocols) and 3D cultures the number of clones should be equivalent to the number of starting cells. Clone size distribution in 2D remains rather constant in 2D ESCs and only slightly increases in 2D neuronal differentiation. In 3D, however, the difference between large and small clones (the lineage size heterogeneity) increases with time. This is because in 3D cultures, some lineages maintain a pool of dividing cells (both symmetrically and asymmetrically), while in 2D neuronal differentiation cells are restricted in their potential to continuously produce neurons.

The key finding of our experiments is that, although within each clone in 3D organoids the majority of cells differentiates and stops dividing, some dividing progenitors are maintained much longer than in neuronal 2D cultures. This might mimic in vivo development and results in stronger heterogeneity between individual clone sizes.

5. Organoid reproducibility (Lines 94-95): This is quite interesting that two independent libraries produce nearly identical results in triplicates. Would the authors elaborate on how this is possible given the known heterogeneity among organoids under most derivation methods? Also, if they are that identical – would the authors explain the differences seen in lineage size distribution shown in Fig. 1f compared with that of Extended Data Fig. 3d? In general, confirming the validity and robustness of this method requires replication in different cell lines (preferably 2 iPSC lines).

As outlined in the main text, the two barcoded libraries were independently cloned but were used together in the same organoids. They were used as intra-organoid controls or to separate different genotypes in chimeric organoids. In control organoids the two barcode libraries behave similarly because they both represent 50% each of entire organoid. This was further detailed in the methods section.

Regarding intra-organoid variability, we find some limited variability between triplicates at each timepoint, however, over time organoids are reproducible in their behavior. For each experimental condition and timepoint we performed whole organoid lineage tracing in triplicates. Averaged data is presented in the main figures, but each individual measurement is shown in the extended data. The differences between Fig. 1f and Extended Data 3d is due to different sequencing depths of these two different rounds of experiments. In the WT lineage tracing experiments presented in Fig. 1f, we sequenced extremely deep while the second set of WT lineage tracing experiments was sequenced less deep (Extended Data Fig. 3f). The comparison across these experiments needs to take this into account, which is why we provide

the Extended Data Figures 3f, g in the manuscript in which this comparison is controlled for by sequencing depth. Our key conclusion, namely that lineage size heterogeneity in cerebral organoids at the latest timepoint of our analysis spans some 3-4 orders of magnitude, remains the same in both sets of experiments. This is also shown in the RFP-WT : GFP-WT mixtures in Extended Data Fig. 9. As this remains reproducible across several experiments in multiple monoclonal subclones, we believe that repeating the experiment in another iPSCs line is outside the scope of this study.

To test the observed variability in our organoid differentiation we performed qPCR analysis of patterning markers (Review Figure 1). The individual organoids we measured acquired dorsal patterning, while they do not cluster with ventral patterned organoids when compared to organoids from previous experiments (Esk et al., 2020, Bagley et al, 2017). This shows that we predominantly generate dorsal brain regions with minor variability, but we do not generate ventral brain tissue.

6. Symmetrical Vs. asymmetrical cell division (lines 136-138, 172-174). The authors do not demonstrate symmetrically (or asymmetrically) dividing cells. The computational simulation nicely suggests S-> A/N transition that and it is supported by the literature and data accumulated on organoids. But larger and less heterogeneous lineages are not necessarily symmetrically dividing neural stem cells. SOX2+PAX6+ cells for example can produce more SOX2+PAX6+ cells only, or SOX2+PAX6+ cells and TBR2+ cells, the latter which symmetrically divide as well to two TBR2+ cells that then again later divide symmetrically to give rise to two neurons. Again, RNA-Seq of barcoded cells is required for validation of cell types produced and assessing how powerful the method is.

We thank the reviewer for his/her comment. We performed live cell imaging experiments coupled with fate mapping to show the presence of both symmetrically and asymmetrically

dividing cells. This is part of the newly added Figure 3. Based on our predictive model and scRNA-seq data larger lineages don't exclusively contain S cells, but rather contain S, A and N cells.

We agree with the reviewer that assessing the role of intermediate progenitors in our models is important this point has also been raised by reviewer 2. Towards this goal, we improved our model to include the effects of TBR2+ intermediate progenitor cells which would only produce a limited output of cells. We find that they do not greatly influence the overall lineage size distribution even if they divide up to seven times. Even with those modifications, it is still the prolonged presence of symmetrically dividing cells that has the strongest influence on heterogeneity of lineage size distributions.

The second part of the reviewer's question concerns cellular heterogeneity within each lineage. Our model is stochastic and assigns S-, A- and N- cells to all lineages as would also be predicted from previously published data in mice (Gao et al., Cell 2014; Llorca et al., Elife 2019). As requested by the reviewer, we have now included an experiment where we combined lineage tracing with barcoded scRNA-seq in wildtype organoids. Indeed, this confirms that larger lineages are composed of both progenitors (S- and A- cells) as well as N-cells. We do not see large clones containing only S-cells – a theoretical possibility rightfully pointed out by the reviewer.

7. Tissue replenishment (line 195-6, 218). a. The authors do not provide full evidence that the original cell repertoire of organoids has been restored, but only the lineage size and number. Since cell type composition is not shown, it is not clear if cells at any point can restore what was depleted. The authors should provide single cell data of barcoded cells to show that cell types have been restored.

We thank the reviewer for his/her comment. In our original submission, we did not state that cell qualities (cell types) are fully restored in organoids under conditions of replenishment. We only measure and state this for organoid size and lineage size adjustment (cell quantity) in the main text. In fact, depending on experimental conditions and timing, cell type quality cannot be fully restored, most notably in chimeric organoids containing WT and PAX6 KO cells, in which PAX6 KO cannot attain dorsal progenitor fates.

In further correspondence with the reviewer and the editor it was concluded that barcoded scRNA-seq lineage tracing in chimeric organoids in all conditions is outside the scope of this manuscript. In the case of puromycin chimeric organoids, we now have added additional qPCR and microscopy analyses to address the question of tissue replenishment. We demonstrate that progenitors (Sox2), neurons (NeuN) as well as dorsal subclasses of cells, intermediate progenitors (Tbr2) and excitatory neurons (Tbr1) are present in replenished organoids, arguing that there is no / limited change in tissue quality in replenished organoids. This is in line with qPCR data for patterning markers also included in the revised manuscript.

8. The latter applies also to the defected cell line compensation experiments (line 237). Full restoration means that the organoid is fully compensated in cell composition, and the current experiments cannot answer this. Also, SOX2+ rosettes surrounded by neurons are not sufficient to prove of full repertoire compensation.

Please refer also to the previous sections. In the case of genetic chimeric compensation, we did not claim in the original submission, do not expect and do not see full cell type quality replenishment. Nonetheless, also for these conditions we now offer additional microscopic analyses to better assess the state of genetically chimeric organoids.

Letter 1 to handling editor / reviewer 1 (Jan. 12, 2023):

Dear Stelios,

(...)

The **key novel finding** reported in our manuscript is the identification of cells that continue dividing symmetrically even during the late stages of organoid development. We demonstrate that these persistent symmetrically dividing cells result in a tunable lineage output allowing organoids to grow to their correct final size even when a fraction of cells is ablated during their development.

Reviewer 1 had some concerns about our methodology and **requested that we provide single cell RNA Seq data** of the barcoded organoids. The reviewer was concerned about the cell type composition within individual lineages and questioned whether not only cell number but also cell-type composition would be correctly restored. Specifically, the reviewer asks about (A) **the cell types that perform symmetric and asymmetric divisions** in the developing organoid and (B) **the replenishment potential within individual lineages** in terms of cell types that are generated. The reviewer seems to be under the impression that replenishment of perturbed lineages leads to full cell-type replenishment in chimeric organoids (quote: "The authors do not provide full evidence that the original cell repertoire of organoids has been restored, but only the lineage size and number. Since cell type composition is not shown, it is not clear if cells at any point can restore what was depleted. The authors should provide single cell data of barcoded cells to show that cell types have been restored"). Therefore, the reviewer suggested barcoded scRNA-seq experiments to resolve these questions and in your response you asked us to include those data in the revision.

In fact, however, **the suggested experiments will not help addressing those questions** for several reasons:

(A) There is no published transcriptomic signature that separates symmetrically dividing cells from asymmetrically dividing cells (both in radial glia and intermediate progenitor scRNA-seq clusters). Therefore, the proposed experiments will not allow us to determine, which cells divide symmetrically and which ones divide asymmetrically. The best way to address this would be to do image-based lineage tracing (live imaging of clones with subsequent staining). In fact, however, this experiment is included in a preprint by Alexandre Baffet's lab, in which both symmetric and asymmetric divisions are demonstrated for basal radial glia and intermediate progenitor cells (Coquand et al, BioRX, 2022). We can discuss this preprint in the manuscript, while it had also been co-submitted with ours and addresses the question better than any scRNA Seq data would.

(B) The reviewer wants us to demonstrate correct representation of cell types after compensatory growth. However, we do not want to make this claim at all. In fact, we even show that organoids vary in their cell type composition in some chimeric conditions from pure wildtype organoids (i.e., RFP-WT, chimeric with GFP-PAX6-KO cells, figure 4d and Extended Data figure 10). This is expected based on the experimental condition, as PAX6-KO cells divert into alternative fates, including choroid plexus tissue, thereby changing the cell type composition of chimeric organoids. Thus, we feel it would not be productive for us to carry out an experiment that we already know will not give the expected result. Instead, we would like to discuss the limitation of the study concerning cell type composition within individual lineages in detail, which was also requested by reviewers 2 and 3. This includes to clarify claims of full compensation and put them in perspective with regards to the complementarity of our whole organoid lineage tracing method to scRNAseq approaches.

As an **alternative**, we propose to **include scRNAseq experiments of our organoids at early and late timepoints (day 23 and day 42)**. We would use those data to identify sub-populations of radial glia and intermediate progenitor cells that – based on what is known from the literature – could correspond to symmetrically vs asymmetrically dividing populations. We feel this would be a more productive approach towards resolving point A raised by reviewer 1.

Again, we thank you for your and the reviewers' interest in our work. We look forward to hearing your thoughts on what we propose and would be happy to discuss further if necessary.

Best regards,
Jürgen

Editor / reviewer 1: response to our letter 1 (Feb. 16, 2023):

“Symmetric cell division cannot be simply proven at the single cell single molecule level and this was not asked explicitly from the authors. Imaging-based lineage tracing, as also presented for oRGs and IPs by Baffett and colleagues and as mentioned by authors, was also not requested. This is because of the realization by the reviewer that the power of the proposed method is in its molecular and high-throughput like nature.

What needs to be done to strongly support the validity of this method to predict symmetrical / asymmetrical cell division of NSCs is showing that the multiple uniquely barcoded large lineages during early development, which indicate low clonal heterogeneity, indeed contain NSCs only - and not NSCs and their progeny.

According to our model, clones of all sizes contain NSCs and neurons. We did not claim in the submitted manuscript that clones of only NSCs (and potentially large size) exist in organoids. However, the reviewer pointed out a possibility for such clones, even if not suggested by our model. We have now addressed this possibility with the barcoded scRNA-seq lineage experiment suggested by the reviewer. We can confirm that even the largest clones contain neurons, besides NSCs. We thank the reviewer for prompting us to perform this control.

In current study it seems that such conclusions rely on the fact that barcoded clones have reached a large size, hence representing symmetrically replicating NSCs. However, a given pool bearing an identical barcode can contain a family of NSCs and their progeny and not necessarily only NSCs, such as mixture of NSCs and IPs, IPs only, a mix of massive wave of differentiating IPs and their symmetrically derived neurons, and so on - all can produce considerably large clones.

In addition to the barcoded scRNA-seq experiment discussed above we have now also improved our model to test for intermediate progenitors. We find that INP divisions, even in extreme cases of up to 7 divisions do not match the observed lineage size distribution. Hence, our model points towards a subset of NSCs maintaining the potential to and actually dividing symmetrically.

Showing exclusive NSC identity (Sox2, Nestin and additional genes associated with NSC signature and identified by scRNA-Seq) with no (or a minority of) differentiated progeny (Tbr2, Tbr1, Dcx etc) for any of these large uni-barcoded clones will at least indicate that these NSCs are in an expansion phase and hence are at high probability neural stem cells that replicate via symmetrical cell division. This may not tell whether there are several types of NSCs contained within these large pools such as apical RG, oRG, early RG or late RG NSC types – all possibly bearing similar (although probably not identical) NSC signature. Yet, as long as they can be shown to contain an exclusive NSC identity, that will be fine with proving the concept of large NSC clones that only produce cells of their kind.

In that sense, based on the above it will be interesting to find out whether late organoids also contain clusters exhibiting exclusive NSC identity - as sign for ongoing symmetrical cell division at late stage.

We would like to again point out that we do not claim that large lineages only contain NSCs. The reviewer is correct that we do not know what exact subtype of radial glia perform symmetric divisions. Based on literature, it is worth pointing out that all radial glia types mentioned would be theoretically capable of symmetric divisions and that transcriptional profiles of NSCs are not predictive of division behaviour.

For the replenishment part, it is clear that depending on the mutation, RFP (WT) and GFP (mutated) cells exhibit different pattern dynamics within chimeric organoids. Also, it is not expected that WT and choroid plexus (Pax6 KO) containing organoids would look similar to WT/WT organoids. The authors should investigate whether the major, compensating cell type (WT or mutated, depending on the case) and not the entire chimeric organoid as a whole can restore organoid cell composition in addition to organoid size. This will demonstrate the tunability of NSCs to provide intact growth compensation that is also meaningful at the cell type level – i.e. NSCs that can restore adequate differentiation potential”.

In the chimeric organoids only the wildtype population can replenish any defects caused by defective KO cells. Whatever phenotype KO cells have cannot be compensated for by KO cells as we analyze whole organoids and the sum of individual KO lineage effects is the overall KO effect that is compensated for by WT cells. To this point, the whole organoid lineage tracing data allows for analyses of WT and KO lineages in a chimeric organoid as a population. While

ASPM-KO lineages are smaller as a whole population to WT lineages, TP53-KO lineages are larger than WT lineages – again as a population. This is indicated in the corresponding violin plots (old Extended Data Fig. 8, new Extended Data Fig. 9), the optimal transport plots (old Figure 3 and 4, new Figure 4 and 5) and density plot (old Figure 4f, new Figure 5f). Changes in lineage size population distributions are seen in PAX6-KO lineages and correspondingly in chimeric WT lineages.

Our letter 2 to handling editor / reviewer 1 (Feb. 21, 2023):

Dear Stelios,

(...)

Below, we outline the complications arising from reviewer 1's comments:

We are in principle prepared to perform the suggested barcoded scRNAseq experiments suggested by the reviewer. Nonetheless, we are **puzzled by the reviewer's expectations, which we will not be able to demonstrate with the suggested experiment.** In the first version of the review, the reviewer had asked about the exact **cell types that divide symmetrically** at late stages of brain organoid development – the existence of which are one of the key findings in our manuscript.

In the new version of their response, the reviewer is **now interested in the cell type composition of individual clones.** The reviewer expects large clones to contain NSCs with little neurons in the same clones (quote: "Showing exclusive NSC identity (Sox2, Nestin and additional genes associated with NSC signature and identified by scRNA-Seq) with no (or a minority of) differentiated progeny (Tbr2, Tbr1, Dcx etc) for any of these large uni-barcoded clones will at least indicate that these NSCs are in an expansion phase and hence are at high probability neural stem cells that replicate via symmetrical cell division.).

In fact, however, **we already know that large clones contain a large number of neurons in addition to a few symmetrically dividing NSCs.** We show this in our imaging efforts of individual clones (extended data 1f) and in our modelling of clones containing symmetrically dividing S-cells (figure 2i). According to our analysis, such clones contain 1-50 symmetrically dividing S-cells, alongside 100+ asymmetrically dividing cells and oftentimes 1000+ non-dividing cells (figure 2i). So, the reviewer's notion that clones contain predominantly NSCs is not supported by our imaging and modelling data. Also, in contrast to our whole organoid lineage tracing method, which takes information of all 1M+ cells in an organoid into account, scRNAseq sampling of ca. 10000 cells per sample will not exhaustively cover clones. We can discuss these differing foci of our method and scRNAseq in the text in more detail as also recognized by reviewers 2 and 3.

One experiment to address the reviewer's question would be to **induce clones at later stages of organoid development** and then follow them for just short periods of time (4-7 days) to possibly see divergent clone composition (clones consisting of symmetrically dividing cells

only, symmetrically dividing cells and differentiating cells, individual already differentiated cells etc). However, induction of clones labelled late to be ready for high throughput deep sequencing is technically not feasible. The best way to do this experiment is to use microscopy approaches. This limits the number of analyzed clones to 10s to 100s. Furthermore, this is exactly the experiment contributed by Baffett et al. in a very beautiful way. For us, publishing the two papers back to back is way superior to duplicating the experiments reported in that paper.

In principle, we could perform clonal analysis using barcoded scRNAseq, but the data already provided in the manuscript show beyond doubt that the experiment will not – and should not– give the result expected by the reviewer. If the reviewer insists or if the reviewer’s intention is different from what they have outlined in their comment, we can provide a neutral description of clone composition, knowing that we will miss a lot of information due to shallow sampling of scRNAseq.

Concerning the **reviewer’s second point** about cell type composition of replenished organoids, we agree that it would be interesting to observe if next to organoid and lineage size also cell type composition is restored in these organoids. **We already have several lines of experiments in the manuscript (various genetic chimeric organoid settings as well as various ratios of drug-induced chimeric cell death assays)**. Given the complexity of the experimental approach, with the difficulty of sub-sampling organoids and the cost intensive nature of these experiments it would be great to have input on whether this is really what is expected of us. Happy to discuss via phone or zoom to elucidate which experiments should be prioritized. If we strictly follow the reviewer, the entire suite of experiments should be repeated using barcoded scRNAseq, which in our view vastly exceeds the current scope of the manuscript.

Reviewer 1 / handling editor: response to our letter 1 (Apr. 11, 2023):

As discussed, we have contacted the referee again, who went through your points and discussed the issues raised with us. The referee notes that they are not convinced about symmetric cell division from extended data figure 1f. To the reviewer, if it is found by scRNA-Seq that clones contain stem cells and their progeny, as you suggest, then it cannot be claimed definitively that NSCs divided only symmetrically as they could also self-renew asymmetrically to give rise to the progeny. To better substantiate such claims, the referee suggests that (in addition to your suggestion to “induce clones at later stages of organoid development and then follow them for just short periods of time (4-7 days) to possibly see divergent clone composition with microscopy approaches”) you also induce and show scRNA-Seq at least in early organoids; if among diverse clones at early stage you find mostly or some NSC-only containing clones, this will strongly support existence of expanding NSCs at early stages in organoids as indicated by barcode behavior at that stage. To the reviewer, for late stages, it can also be speculated that the growing diversity in clonal size indicated by the barcode is due to decrease in stem cells and increase in progeny. This can be added to the clonal pattern by microscopy that you suggest to perform.

We thank the reviewer for this additional explanation. As mentioned previously, inducing clones for short periods of time during organoid growth for high-throughput clonal analysis would be an ideal experiment, but it is technically not feasible. Instead, we have teamed up with Alexandre Baffet's lab to provide data on cell division behavior in organoids. His lab established clone labelling of individual dividing cells in slice cultures and analysis of subsequent fate establishment in daughter cells using a retroviral labelling approach followed by live imaging / fixed fate analysis. We find abundant symmetric cell divisions in neural rosettes / ventricular zones, further supporting the existence of such division behavior in organoids.

Additionally, we now provide barcoded scRNA-seq data of wildtype organoids at two timepoint (days 18, 42). As expected at day 18 organoid lineages contain predominantly progenitors. The day 42 organoid large clones contain more neurons, while maintaining progenitors. As discussed above, this experiment has been important, because it shows that large clones modelled to contain symmetrically dividing cells also contain neurons. This data further corroborates our imaging data of individual clones in the previous submission.

Finally, regarding the reviewer's second point about cell type composition of replenished organoids, the referee believes that since the barcode analysis shows both organoid size and lineage size clonal pattern restoration, this would suffice and scRNAseq would not be required here.

We thank the reviewer for agreeing that the initial suite of experiments lies beyond the scope of the current manuscript and that the current imaging, qPCR, flow cytometry and whole organoid lineage tracing data sufficiently supports replenishment.

Reviewer #2:

Remarks to the Author:

Review of: Cerebral organoids display dynamic clonal growth with lineage replacement
By: Lindenhofer et al 2022

Overall recommendation:

Cerebral organoids have become a staple model system for understanding human brain development and modeling disease. Here, Lindenhofer et al characterize the rules of clonal expansion of stem cells in the context of cerebral organoids, modeling symmetric and asymmetric divisions also observed in development. They utilize a barcoded lineage tracing strategy alongside a whole-culture or whole-organoid pooled sequencing readout to assess clonal dynamics. The authors claim that progressive loss of lineages demonstrates selective pressures during organoid growth. Additionally, using chimeric organoids, drug-based selection demonstrated adaptive lineage compensation. Finally, genetic perturbations to endogenous loci are shown to affect lineages and specification. The results included in this manuscript are on the whole thorough and appropriately analyzed. Further validation of

organoid identity, a more thoughtful analysis of barcodes, and some concrete connection to in vivo work would strengthen the conclusions of the paper. However, the clever use of chimeric organoids, and particularly the functional assays presented in figures 3 and 4 are novel and contribute to our understanding of organoids as a model. As the authors point out, in the future, it would be very exciting to leverage these approaches in order to understand species differences in neural stem cell clonal expansion.

We thank the reviewer for his / her enthusiasm for in our manuscript and whole organoid lineage tracing methodology.

Major comments:

While organoids are a commonly utilized model, whether the results demonstrated in this paper could be recapitulated in vivo is unknown. Some stronger connection to the in vivo setting would significantly strengthen the significance of the findings in the present manuscript. Alternatively, please address this as a caveat.

The reviewer is correct in that organoids do not reflect the in-vivo situation in full detail. We have discussed the limited available sparse human in-vivo data now in more detail in the text. We also point out differences between murine and human lineage sizes in the text and the relationship to the organoid data. We also discuss more details of the advantages (lineage tracing of all organoid lineages in parallel, taking all cells into account) and limitations (labeling timing, in-vitro model) in more depth. We do not provide additional experimental data to this point in this manuscript but want to point out a preprint by Alexandre Baffet's lab (Coquant et al, BioRX, 2021) that was co-submitted with this manuscript. Their paper studies cell division behavior of progenitors in both organoids and human brain slices which reveals symmetrically dividing cells in those model systems. This study finds very similar division behaviors in organoids and human brain slices.

In the developing brain and in organoids alike, symmetrically and asymmetrically dividing neuroepithelial cells and radial glia are not the only progenitor populations. Intermediate progenitor cells (IPCs) can make up ~10% of cells in an organoid, depending on the time of sampling. IPCs arise from RG and perform a consumptive division to give rise to 2 neurons, following the differentiation pattern $S \rightarrow 2 \times N$ (borrowing the nomenclature from the manuscript). How does inclusion of these cells in your model affect its performance? Could this be used to explain differences between simulated and actual data? Please consider addressing this as a limitation in the text, or revise the model to include divisions expected from this cell type. Alternatively, if the authors stain for the IPC marker EOMES and find that, surprisingly, they make up a very small proportion of organoid cells, then these data can be added in the supplement to demonstrate that the model does not need to be revised.

We thank the reviewer for these important comments and constructive criticism. We have now examined the role in intermediate progenitor cells (IPCs, also called intermediate neural progenitors, INPs) in more detail and find that they do not explain the lineage size distribution in whole organoids we observe.

Intermediate progenitor cells were measured with IHC (EOMES) and scRNA-seq confirming the presence of this cell type in our organoids. The influence of this cell type was included in an improved version of the model. In this improved version of the model, we directly test the effect of INP divisions on lineage size heterogeneity. Notably, INPs are considered to divide predominantly once into two neurons in mouse and possibly a few more times in humans. In any case INPs are considered to divide with limited capacity. To incorporate transit amplifying INPs to our model, we adjusted the division mode for asymmetrically dividing cells for $A \rightarrow A+N$ (old) to $A \rightarrow A+kN$ (new). The factor k covers direct neurogenesis ($A \rightarrow A+1N$), symmetric consuming divisions of INPs as seen in the mouse ($A \rightarrow A+2N$) as well as suggested additional INP divisions in humans ($A \rightarrow A+kN$). We provide model runs for up to $A \rightarrow A+8N$ covering several theoretical division modes including potentially 6 asymmetric INP divisions to generate neurons followed by a symmetric consuming division. This data is provided in the expanded Extended Data Fig. 5. We find that the inclusion of INP in the improved model influences the number of total cells present in the organoid, but does not greatly influence the lineage size distribution. To best model the observed lineage size distribution it is vital to maintain a constant pool of S-cells capable of dividing symmetrically. As in our previous model, this S-cell population may not be depleted or allowed to overgrow to explain the measured data. Again, we thank the reviewer for prompting us to explore this important variation.

Excitatory neurons are historically particularly sensitive to dissociation. Are neurons selectively lost during dissociation using the authors' protocol? If so, clone sizes will be artificially low, especially those which contain proportionately more neurons. More importantly, estimates of proportion of symmetric to asymmetric division could be affected. The authors could strengthen their work by either demonstrating equivalent loss of all cell types during dissociation, or addressing this as a limitation of their study.

Thank you for pointing out the vulnerability of excitatory neurons which is problematic for most scRNA-seq approaches. We see similar difficulties in some assays that involve dissociation of organoids into a single-cell suspension. One strength of our whole organoid lineage tracing methodology is that neurons are not lost during the protocol as no single-cell suspension needs to be generated. Each organoid is lysed as a whole (including all neurons) in a single reaction tube directly for genomic DNA preparation and subsequent PCR-based measurements of integrated barcode abundance. Therefore, we are confident, that all cells in an organoid are taken into account. We adapted the corresponding method section to further clarify this in more detail.

We agree that loss of vulnerable cell types (i.e. excitatory neurons) in other assays such as scRNA-seq is a concern as it skews the ratio of cell types observed in a sample. We find this likely to be the case in our new barcoded scRNA-seq experiments and discuss this in the text in the corresponding section.

Fig 1c+d and elsewhere: With this bulk method, how can the authors distinguish between a.) two lineages of the same cell count from b.) one smaller lineage with two or more integration sites of the same barcode? Relatedly, how can the authors distinguish between c.) one lineage which received two distinct barcodes and d.) two equally sized lineages, each with different barcodes? Conceptually, these questions need to be addressed. If these considerations are

currently incorporated in the analysis pipelines, this should be stated clearly. Alternatively, these questions should be addressed as a major caveat to the approach.

Thank you for asking about these important quality checks. The barcode variability is extremely high in our experiments. Our theoretical barcode complexity is 2.81×10^{14} . We have previously generated complex barcode libraries (Esk et al, Science, 2020) and applied the same cloning and quality control strategies. We now show the Hamming distance frequencies between all library combinations within two replicates at day 0 (organoid start day) in Extended Data Fig. 1b. In line with our previously published barcode library experiments having twice the same barcode in a single cell is extremely rare (below 0,001% chance for a clash in a single cell in an organoid containing 10000 starter cells).

We cannot rule out the possibility that a clone may be labelled by two barcodes, however, such events would be rare and do not significantly change the interpretation of whole lineage organoid tracing experiments. The infection rate for whole tissue lineage tracing experiments ranged from 4 -7 % (Esk et al, Science, 2020), therefore double infection rates are low representing a minor fraction of all lineages. As a test case, we have modelled the effect on lineage size distributions over time for a double infection rate of 20% (Review Fig. 2). Our models run on 10000 independent lineages. In the theoretical case of 20% double infections, the sequencing reads would be assigned to 8333 lineages ($8333 \times 1.2 = 10000$). We find that the overall lineage size distribution does not change either in terms of largest to smallest clone nor in the bimodal curves emerging. The only difference is that reads are spread over 8333 datapoints instead of 10000 datapoints.

Review Fig. 2. Modelling the effect of barcode double infections (double infection rate: 20%) on lineage size distributions. a,b, Rank plots of modelled lineage size distribution encompassing symmetric divisions, S-cell to A-cell transitions, asymmetric divisions and S-cell consuming divisions. Rates for individual model steps are indicated at the top. Left panel (a): Lineage size distribution of 8333 single infected lineages over time. Right panel (b): Lineage size distribution of 8333 lineages with 20% double infections over time, resulting in 10000 modelled lineages.

Therefore, we conclude that the possibilities for potentially misinterpreted barcode information as pointed out by the reviewer are probably occurring at low rates in our experiment but lead to negligible effects on the interpretation of the results. Essentially, this misinterpreted information is minor noise inherent in our approach. We now mention these issues in the text and added the Hamming distance data to the Extended Data Fig.1b. Again, thank you to the reviewer for these important technical clarifications.

Variability in cell type makeup within and across organoids could affect cell division modes, however this is not characterized formally in your study. How many replicates were stained for quality control? How much non-brain-like tissue is included in the organoids? Organoids are known to be highly heterogeneous, batch-dependent, and line-dependent.

We agree, that organoids display a certain degree of variability. In the lab, we quality control batches using a qPCR panel. We find that organoid used in this study behave reasonably consistent (reviewer fig. 1). We also provide additional stainings in the manuscript. scRNAseq and stainings do not show a large proportion of non-brain tissue.

Additionally, we would like to point out that all whole organoid lineage tracing measurements in all conditions at all timepoints were carried out in triplicates. Aggregated data are shown in the main figures and individual measurements in the Extended Data figures. We find clone size distributions remarkably consistent in our experiments.

Minor issues:

How was cell number measured in figure 2a? Consider including some details in the caption.

Cells were counted in a cell counter (Countess II) in a defined volume after dissociation. This is also described in detail in the corresponding method section (Organoid cell numbers and flow cytometry).

How were the coefficients for different types of divisions selected for your model (shown in Fig 2f)? Please include rationale.

Cell division rates have been chosen for 1.5 events per day for symmetrically and asymmetrically dividing cells. This equals a cell cycle time of 16h, roughly in line with data for mouse radial glia.

Of note, we have run the model with different cell cycle rates. This changes the total amount of cells created but not the relative number of cells. The deciding factor for lineage size distribution is the ratio between $S \rightarrow 2S$ divisions and $S \rightarrow A$ transitions. This rate needs to be equal to maintain a consistent population of S-cells, which is key to best fit the observed data. If S-cells are lost ($S \rightarrow 2S < S \rightarrow A$) or increased ($S \rightarrow 2S > S \rightarrow A$) the resulting cell distribution does not match observed data (see Extended Data Fig. 5).

Consider validating the knockout lines used late in the paper by qPCR or IHC to further demonstrate functional (not just genomic sequence-based) knockout as is displayed in Extended figure 8.

We have now confirmed that KO lines (ASPM, TP53, PAX6) indeed have lost the protein by western blot (TP53, PAX6) and immunofluorescence (ASPM) in Extended Data Fig. 8. Unfortunately, our ASPM antibody did not work in western blotting.

In methods and/or in extended figure 6 caption, please include the concentration of BrdU added.

We have added the concentration of BrdU used in the respective method section. This is in line with published concentrations that are used in the cerebral organoids.

It is not clear whether each “replicate” or “sample” (such as those reported in Extended Fig 3e, f, for example) pertains to one organoid or one batch of organoids. This should be clarified.

We thank the reviewer his comment. We clarified this now throughout the manuscript and refer to a batch of organoids grown as differentiation experiment while individual organoid replicates are described as organoids. An example is now: $n = 3$ organoids from 1 differentiation experiment at each time point.

Each whole organoid lineage tracing measurement represents one organoid in the Extended Data Figures. Each timepoint in each condition was lineage traced in triplicate and the averages analyzed in the main figures. Wild type organoids were lineage traced at all timepoints in triplicates in two independent batches / experiments. Results were very comparable and the comparison is shown in the mentioned Extended Data Figure 3 e, f.

Extended data 1f and 10 figures could be included in main figures as they help readers understand the experiments being performed, whereas some details in figure 2 and 3 could be moved to the extended figures, if desired.

We opted to keep the mentioned Extended Data figures 1f and 10 in the Extended Data section, because 1f supports the whole lineage tracing data in main figure 1 without adding a new biological aspect. Similarly, Extended Data Figure 10 supports the data in the old figure 4 (now figure 5), without adding strong additional insight. The key finding, the absence of PAX6-KO in neural rosettes and their replenishment by wildtype cells is already visualized in the main figure 5. Additionally, both panels would require a significant amount of space in the main figures.

Reviewer #3:

Remarks to the Author:

In this study, Dominik Lindenhofer et al. utilized a genomic DNA barcoding technique to perform whole-tissue lineage tracing on human organoids. In addition, they found that a group of stem cells retaining symmetrically dividing could lead to a great lineage size heterogeneity during the development process of organoids. Furthermore, the growth of neural stem cell lineage was plastic and possessed the self-healing ability when encountering cell-loss induced by either chemical and genetic ablation. Overall, these findings are highly innovative and intriguing, providing a platform for studying different clonal lineages in 3D organoids. There are a few aspects need further consideration for additional merits.

We thank the reviewer for his / her interest in our manuscript and recognition of the importance of our work!

1. The technique of DNA barcoding used in this manuscript is interesting but complicated. It would help readers better understand this system if more description about technical details

could be added. For example, why the authors used two DNA barcode libraries to label each starting cell, not one or three? Please add more explanation or related references.

We have now expanded the method section regarding the lineage tracing technology and added additional information. Two viral barcode libraries were cloned independently, mixed, retrovirus was generated and used to infect H9 stem cells. Of note, each starting cell in a wildtype organoid is labelled with only one barcode out of two barcode libraries. Thereby one half of starter cell clones can be compared internally to the other half of starter clones. Both populations act highly similarly in WT conditions, serving as an internal control. In the chimeric organoids that contain two different genotypes the two independently cloned viral barcode libraries were used to distinguish these genotypes via sequencing in whole organoids. In this case the plasmid libraries were not mixed prior to making retrovirus, but used to infect the two different genotypes. As they acted highly similar in the WT lineage tracing experiments, we concluded that there would be no need for additional independently cloned libraries in the subsequent chimeric organoid experiments.

2. In extended. Data Fig. 2e-g, the developing mice were injected with DNA-barcode virus in three timepoints and sampled at E18.5. What is the infection efficiency of the virus at the sampling time? The conclusion of smaller clonal heterogeneity in mouse seemed not be very convincing, since human organoids and mouse embryos were infected at different timepoints and with different infection approaches. A clarification is necessary.

We typically achieve infection rates of 2-4% of mouse cells lining the ventricle. Such low infection rates ensure that a single cell receives only a single barcode for lineage tracing. This is corroborated by the images provided in Extended Data Fig. 2f in which a mixture of RFP and GFP expressing virus is imaged in a mouse brain. Only GFP or RFP-positive cells are seen, with extremely rare exceptions while the vast majority of cells were uninfected (non-visible space).

We agree with the reviewer that the mouse data is difficult to compare to the organoid data and that was not our purpose. The purpose of the experiment was to validate the sequencing-based lineage tracing approach by applying it to a well-characterized paradigm of image-based lineage tracing in a brain context in mice. Results are very comparable to such previous studies (ie Llorca et al, *Elife*, 2019; Gao et al, *Cell*, 2014).

3. In Extended Data Fig. 3, the authors tried to demonstrate that tissue context affects clonal lineage growth. Although there was a tendency of more divergent distribution in 3D organoids, statistic calculation was not found to support this conclusion? It would be more convincing if some statistical tests were added.

We thank the reviewer for his/her comment. We added a panel dedicated to the direct comparison of the lineage size distributions at day 42 for the different culture or differentiation systems (Extended Data Fig. 3e) with the corresponding statistics confirming that the distributions are significantly different from each other.

4. In Extended Data Fig. 6, the organoids were labeled with BrdU at day 40. But it seemed not suitable to define Day 40 organoids as late organoids. Please label organoids with BrdU at Day 80 or later.

We agree with the reviewer that analysis of possible symmetrically dividing cells at later stages of organoids will be an interesting question moving forward. In the current manuscript we concentrate all our efforts towards 6 weeks old organoids and provide an array of assays (whole organoid lineage tracing, microscopy, live imaging etc.). We opted to do so to thoroughly characterize up to this timepoint. However, future efforts should be directed at the questions how long into organoid development symmetrically dividing cells persist. In a co-submitted study Alexandre Baffet's lab has imaged symmetrically dividing cells in organoid up to day 105 (Coquand et al, BioRX, 2022) strongly suggesting such cells exist well past day 80.

5. In Fig. 4b and Fig. 4c, how many organoids did each point represent in each figure? Please add the number of analyzed organoids in the figure legends. In addition, the error bars of each figure are necessary.

We thank the reviewer his comment. This is now clarified throughout the manuscript in the respective sections. We refer to a batch of organoids grown as differentiation experiment while individual organoid replicates are described as organoids. An example is now: $n = 3$ organoids from 1 differentiation experiment at each time point. For panel 4b (now 5b) error bars are indicated, although they are sometimes small which makes them hardly visible.

6. In Fig.4d, the labels of SOX2 and NeuN are all red. Please label them with colors consistent with immunofluorescence images.

Thanks for catching this inconsistency. We rectified the error.

7. The manuscript is well-written but the discussion seems to be too short. This part would be more abundant and instructive if the limitations of this research and its application prospect are fully discussed.

Thanks for this encouragement. We expanded the discussion accordingly.

Decision Letter, first revision:

Dear Juergen,

Your manuscript, "Cerebral organoids display dynamic clonal growth with lineage replenishment", has now been seen by all of our original referees, who are experts in cerebral organoids, tissue development (referee 1); cerebral organoids, barcoding and lineage tracing (referee 2); and human cerebral organoids, neurodevelopment (referee 3). As you will see from their comments (attached below) they find this work of interest, but have raised some important points. Although we are also very interested in this study, we believe that some of their concerns should be addressed before we can consider publication in Nature Cell Biology.

Nature Cell Biology editors discuss the referee reports in detail within the editorial team, including the chief editor, to identify key referee points that should be addressed with priority, and requests that are overruled as being beyond the scope of the current study. Please note that, for points raised by referee #2, we contacted a subset of the referees to get further input and, taking everything into account, we have decided that the following points will need to be addressed:

(A) All remaining points by referee #1.

(B) The following points by referee #2:

Point #1: "The authors argue that "cerebral organoids have emerged as a faithful self-organizing model system to study human brain development in a tissue context", but this is a deeply misleading statement. There are several studies noting that organoids fail to recapitulate cell types present in the human developing brain in vivo (Bhaduri et al. 2020 Nature and He et al. 2023 Biorxiv from Treutlein lab) due to mechanisms that include metabolic cell stress and other factors. In addition, the organoids used in this study do not incorporate non-neural cell types such as microglia or endothelial cells, which have been shown to have profound impact on brain development (Park et al. 2023 Nature). Similarly, contrary to the authors statements, cerebral organoids fail to recapitulate neuronal morphology and physiological maturation, because transplantation experiments conducted by the Pasca lab have shown only rudimentary cellular features can be derived in vitro (Revah et al. 2022 Nature). Given these limitations, it is clear that cerebral organoids do not faithfully recapitulate the developing brain and the authors should acknowledge these limitations throughout the manuscript".

Point #4: "Addition of timelapse imaging data from cerebral organoid slices offers a notable improvement and validation to the experimental system implemented by the authors, but is not as extensive as the study by Baffet et al. (ref. 25). It is also of limited novelty as the only conclusion from this new data is the demonstration of the presence of self-renewing divisions. The authors use that to argue in a qualitative way that this observation validates their model, but this offers no quantitative validation. From simple theory not specific to the current method, presence of self-renewing progenitor cells would be expected. The added data thus provide limited value to the validation of the method, and falls short of providing quantitative insights, as the experiment appears

to have been conducted only two times based on the data points indicated in Figure 3. The recommendation is therefore to conduct a third replication of the experiment, and to conduct a quantitative comparison of the proportions of self-renewing vs neurogenic divisions to determine how closely those estimates compare to the predictions from the in silico model. Such comparison would constitute a more informative validation of the in silico model”.

Point #5: “Robustness of claims made in this study are limited by the utilization of a single embryonic stem cell line and this should be openly acknowledged as a limitation of this study in Discussion. This is important because in the introduction to the manuscript, the authors make strong statements about the robustness and versatility of the organoid platform and advantages of over methods such as DNA mutation sequencing, but themselves do not fully take advantage of the potential versatility of the method to comment on the robustness of the findings outside of the one embryonic stem cell line. This should be acknowledged”.

Point #7: “On a technical side, the manuscript also lacks novelty with respect to demonstrating the ability of their model to assess the consequences of genetic mutations using organoids. A recent study from the Treutlein lab has shown that introducing loss of function mutations in another patterning gene can be conducted successfully using organoids and recapitulates known biology of this gene. See Fleck et al. 2022 Nature for details. The authors should at least cite this highly relevant paper”.

(C) Point #2 raised by referee #2, but only for the part referring to a better description of the method (you can follow or not the advice regarding the part about rearranging the figures):

“Introduction to the results section should incorporate longer description of the method. This is currently limited to stating that the barcodes are unique and retrovirally inserted, but provide virtually no description of the library complexity or steps taken to limit barcode collision events”.

(D) Finally, please pay close attention to our guidelines on statistical and methodological reporting (listed below) as failure to do so may delay the reconsideration of the revised manuscript. In particular please provide:

- a Supplementary Figure including unprocessed images of all gels/blots in the form of a multi-page pdf file. Please ensure that blots/gels are labeled and the sections presented in the figures are clearly indicated.
- a Supplementary Table including all numerical source data in Excel format, with data for different figures provided as different sheets within a single Excel file. The file should include source data giving rise to graphical representations and statistical descriptions in the paper and for all instances where the figures present representative experiments of multiple independent repeats, the source data of all repeats should be provided.

In contrast, although we agree with referee #2 that their following points would provide valuable insights, it is our policy to avoid raising new points after the first round of review (unless they

are related to new data added after revision) and we consider these issues to be beyond the scope of the present study at this stage. Thus, addressing them experimentally will not be necessary for reconsideration of the manuscript at this journal:

Point #3: "One analysis that seems to be missing from this study is a deeper analysis of cell number per clone emerging within an organoid, and how that relates to estimates from primary cell lineage tracing experiments conducted by the Kriegstein lab using single cell clonal expansion experiments, which have shown that a single radial glia cell can produce 900 cells per clone (Pollen et al. 2015 Cell). On the other hand, single cell clonal lineage tracing in organoids has been previously conducted in cerebral organoids by the Treutlein lab (He et al. 2021 Nature Methods), while limiting the novelty claims made in this manuscript, has shown that average clone size in cerebral organoids is only 2 cells per clone. In the current manuscript, the authors briefly mention the He et al. study noting that the need for dissociation has likely led to an underestimate of clonal cell potential within an organoid, but reading the current manuscript it is almost impossible to grasp where the results fall in terms of the number of cells per clone derived from a single neural progenitor cell, which would be highly informative to studies of brain development".

Point #6: "On a conceptual level, the authors do not provide substantial novel insights into human brain development. The effects of Pax 6 loss of function have been studied for decades and this study does not extend these findings in any meaningful way beyond showing that they can be conducted using brain organoids. In fact, if anything, the results in this study do not agree with findings from in vivo experiments using chimeric Pax6 +/+ <-> Pax6 -/- animals, where Pax6-/- contributed to the cortical neuroepithelium (Quinn et al. 2007 Developmental Biology), whereas in their experiments using organoids, the authors report no contribution of Pax6 KO cells to neural rosettes. This discrepancy reflects either a fundamental limitation of the model system, contrary to the main claim of the paper, or cross-species difference. Either outcome is informative, and highly addressable experimentally, and resolution of this discrepancy, which after all reflects one of the main findings and points of novelty of the study, would be important for a high-quality paper".

We therefore invite you to take these points into account when revising the manuscript. In addition, when preparing the revision please:

- ensure that it conforms to our format instructions and publication policies (see below and www.nature.com/nature/authors/).
- provide a point-by-point rebuttal to the full referee reports verbatim, as provided at the end of this letter.
- provide the completed Editorial Policy Checklist (found here <https://www.nature.com/authors/policies/Policy.pdf>), and Reporting Summary (found here <https://www.nature.com/authors/policies/ReportingSummary.pdf>). This is essential for reconsideration of the manuscript and these documents will be available to editors and referees in the event of peer review. For more information see <http://www.nature.com/authors/policies/availability.html> or contact me.

Nature Cell Biology is committed to improving transparency in authorship. As part of our efforts in this direction, we are now requesting that all authors identified as 'corresponding author' on published papers create and link their Open Researcher and Contributor Identifier (ORCID) with their account on

the Manuscript Tracking System (MTS), prior to acceptance. ORCID helps the scientific community achieve unambiguous attribution of all scholarly contributions. You can create and link your ORCID from the home page of the MTS by clicking on 'Modify my Springer Nature account'. For more information please visit www.springernature.com/orcid.

[Redacted]

We would like to receive the revision within four weeks. Given the holiday season, we could extend this to five weeks, if needed (let us know). If submitted within this time period, reconsideration of the revised manuscript will not be affected by related studies published elsewhere, or accepted for publication in Nature Cell Biology in the meantime. We would be happy to consider a revision even after this timeframe, but in that case we will consider the published literature at the time of resubmission when assessing the file.

We hope that you will find our referees' comments, and editorial guidance helpful. Please do not hesitate to contact me if there is anything you would like to discuss.

Best wishes,

Stelios

Stylianos Lefkopoulos, PhD
He/him/his
Senior Editor, Nature Cell Biology
Springer Nature
Heidelberger Platz 3, 14197 Berlin, Germany

E-mail: stylianos.lefkopoulos@springernature.com
Twitter: @s_lefkopoulos
LinkedIn: [linkedin.com/in/stylianos-lefkopoulos-81b007a0](https://www.linkedin.com/in/stylianos-lefkopoulos-81b007a0)

Reviewers' Comments:

Reviewer #1:

Remarks to the Author:

Lindenhofer et al. have now submitted a comprehensive and satisfactory revised version of their work. This should be published with no further delay.

Main comments:

1. Investigation on cell type composition to elucidate the cell type nature of this dynamic growth or re-growth in organoids:

This has now been done in two modes, scRNAS-Seq and imaging. The finding by the former assay shows that clones are cell-type heterogeneous with or without relation to their size. As discussed in earlier correspondences it is expected that clones do not contain solely symmetrically dividing RG cells, but rather exhibit natural growth dynamics. While existence of such clones would have provided an unequivocally strong statistical proof for both the presence/type of symmetrically dividing cells and the robustness of the lineage tracing method, the live imaging provided now presents satisfactory indication that such cells do exist throughout organoid growth. Together with gene functional data, these findings now link fundamental principles together: organoids recapitulate important developmental features; to do this they rely on perpetuating progenitors throughout growth; these features can be detected by the new lineage tracing method. The work is well appreciated and now the main issue of the study is resolved.

2: Distinguishing apart RG from IP cells in interpretations regarding symmetrical divisions shown in the lineage tracing:

It is very interesting and that S cells must be included in the model so that it is able to predict organoid growth dynamics. Likewise, it is interesting, though bit puzzling that IP cells not play significant role in changing the division / growth landscape in organoids given their significant role in determining neuronal output number. It may be that organoids in these observed stages are less representing a strong IP growth wave. Nonetheless, the requirement of S cell state at any stage is new and sheds interesting light on in vivo development. The lack of IP dynamics in the prediction / measured models should be well discussed in the ms.

3. Replenishment capacity in perturbed organoids:

The point we will take in both the authors' response and new provided data now showing that tissue is also largely qualitatively preserved after 6 weeks of perturbed development. This well supports the claim on replenishment capacity measured by the tracing method.

4. 2D Vs 3D existence of dividing stem cells.

The clarification by authors that clonal output is restrictive in 2D is understood and it makes sense that diversity is restricted compared to 3D. However, it should be emphasized that as long as these 2D cultures are supplemented with mitogens it is obvious that the clonal output tested is of constantly supported proliferation rather than the natural transition between cell states present in 3D. These cultures can be grown for ages and certainly contain stem cells although not necessarily expanding, presumably just self-renewing asymmetrically. A way to detect clonal variability in these 2D cells is culturing them without any growth factors. But then, this will eventually drive complete exit from cell cycle and only neuronal clonal output, not stem cell output diversity, will be measured. No experiments are required here, just a visible clarification / discussion portion in text, as the 2D experiments here are an important reference for the 3D results and the use of this reference should be clear of doubts.

Reviewer #2:

Remarks to the Author:

The revised manuscript by Lindenhofer has been submitted for evaluation. The authors have incorporated substantial new data to strengthen the conclusions of the paper, including live cell imaging from organoid slices. However, despite these improvements, the manuscript requires additional revisions to merit publication in a high-impact journal. I list my remaining concerns below:

1. The authors argue that “cerebral organoids have emerged as a faithful self-organizing model system to study human brain development in a tissue context”, but this is a deeply misleading statement. There are several studies noting that organoids fail to recapitulate cell types present in the human developing brain in vivo (Bhaduri et al. 2020 Nature and He et al. 2023 Biorxiv from Treutlein lab) due to mechanisms that include metabolic cell stress and other factors. In addition, the organoids used in this study do not incorporate non-neural cell types such as microglia or endothelial cells, which have been shown to have profound impact on brain development (Park et al. 2023 Nature). Similarly, contrary to the authors statements, cerebral organoids fail to recapitulate neuronal morphology and physiological maturation, because transplantation experiments conducted by the Pasca lab have shown only rudimentary cellular features can be derived in vitro (Revah et al. 2022 Nature). Given these limitations, it is clear that cerebral organoids do not faithfully recapitulate the developing brain and the authors should acknowledge these limitations throughout the manuscript.

2. Introduction to the results section should incorporate longer description of the method. This is currently limited to stating that the barcodes are unique and retrovirally inserted, but provide virtually no description of the library complexity or steps taken to limit barcode collision events. I would highly recommend relocating some of the excellent material featured in Extended Data Figure 1 to main text Figure 1 and providing a little bit more description of the method in the results section. This will help readers who are novice to the nuances of the technology to better understand its implementation.

3. One analysis that seems to be missing from this study is a deeper analysis of cell number per clone emerging within an organoid, and how that relates to estimates from primary cell lineage tracing experiments conducted by the Kriegstein lab using single cell clonal expansion experiments, which have shown that a single radial glia cell can produce 900 cells per clone (Pollen et al. 2015 Cell). On the other hand, single cell clonal lineage tracing in organoids has been previously conducted in cerebral organoids by the Treutlein lab (He et al. 2021 Nature Methods), while limiting the novelty claims made in this manuscript, has shown that average clone size in cerebral organoids is only 2 cells per clone. In the current manuscript, the authors briefly mention the He et al. study noting that the need for dissociation has likely led to an underestimate of clonal cell potential within an organoid, but reading the current manuscript it is almost impossible to grasp where the results fall in terms of the number of cells per clone derived from a single neural progenitor cell, which would be highly informative to studies of brain development.

4. Addition of timelapse imaging data from cerebral organoid slices offers a notable improvement and validation to the experimental system implemented by the authors, but is not as extensive as the study by Baffet et al. (ref. 25). It is also of limited novelty as the only conclusion from this new data is the demonstration of the presence of self-renewing divisions. The authors use that to argue in a qualitative way that this observation validates their model, but this offers no quantitative validation.

From simple theory not specific to the current method, presence of self-renewing progenitor cells would be expected. The added data thus provide limited value to the validation of the method, and falls short of providing quantitative insights, as the experiment appears to have been conducted only two times based on the data points indicated in Figure 3. The recommendation is therefore to conduct a third replication of the experiment, and to conduct a quantitative comparison of the proportions of self-renewing vs neurogenic divisions to determine how closely those estimates compare to the predictions from the in silico model. Such comparison would constitute a more informative validation of the in silico model.

5. Robustness of claims made in this study are limited by the utilization of a single embryonic stem cell line and this should be openly acknowledged as a limitation of this study in Discussion. This is important because in the introduction to the manuscript, the authors make strong statements about the robustness and versatility of the organoid platform and advantages of over methods such as DNA mutation sequencing, but themselves do not fully take advantage of the potential versatility of the method to comment on the robustness of the findings outside of the one embryonic stem cell line. This should be acknowledged.

6. On a conceptual level, the authors do not provide substantial novel insights into human brain development. The effects of Pax 6 loss of function have been studied for decades and this study does not extend these findings in any meaningful way beyond showing that they can be conducted using brain organoids. In fact, if anything, the results in this study do not agree with findings from in vivo experiments using chimeric Pax6 +/+ \leftrightarrow Pax6 -/- animals, where Pax6-/- contributed to the cortical neuroepithelium (Quinn et al. 2007 Developmental Biology), whereas in their experiments using organoids, the authors report no contribution of Pax6 KO cells to neural rosettes. This discrepancy reflects either a fundamental limitation of the model system, contrary to the main claim of the paper, or cross-species difference. Either outcome is informative, and highly addressable experimentally, and resolution of this discrepancy, which after all reflects one of the main findings and points of novelty of the study, would be important for a high-quality paper.

7. On a technical side, the manuscript also lacks novelty with respect to demonstrating the ability of their model to assess the consequences of genetic mutations using organoids. A recent study from the Treutlein lab has shown that introducing loss of function mutations in another patterning gene can be conducted successfully using organoids and recapitulates known biology of this gene. See Fleck et al. 2022 Nature for details. The authors should at least cite this highly relevant paper.

Reviewer #3:

Remarks to the Author:

In this revision, Dominik Lindenhofer et al. evaluated clonal growth behavior of the whole cerebral organoids tissue dynamically through lineage tracing. The data suggest that the presence of consistently symmetrical dividing cells during the growth and development of human brain organoids may contribute to higher heterogeneity compared to 2D cells or mouse brains. And the growth of neural stem cells in organoids is adaptable to the needs of the whole tissue and has plasticity. The revised paper has addressed major concerns that raised by reviewers, and the manuscript has been better improved. The method that established in this manuscript is well written and would benefit this field. One minor point is the author could describe the result more clearly. For example, in the paragraph "Whole-tissue barcoded lineage tracing", in the sentence of "First, in-silico subsampling of

lineages from the whole organoid lineage tracing ...", the meaning of Extended Data Figure 2a and the ratio of 1%-2% could be explain more clearly.

GUIDELINES FOR SUBMISSION OF NATURE CELL BIOLOGY ARTICLES

ARTICLE FORMAT

ABSTRACT – should not exceed 150 words and should be unreferenced. This paragraph is the most visible part of the paper and should briefly outline the background and rationale for the work, and accurately summarize the main results and conclusions. Key genes, proteins and organisms should be specified to ensure discoverability of the paper in online searches.

TEXT – the main text consists of the Introduction, Results, and Discussion sections and must not exceed 3500 words including the abstract. The Introduction should expand on the background relating to the work. The Results should be divided in subsections with subheadings, and should provide a concise and accurate description of the experimental findings. The Discussion should expand on the findings and their implications. All relevant primary literature should be cited, in particular when discussing the background and specific findings.

REFERENCES – are limited to a total of 70 in the main text and Methods combined,. They must be numbered sequentially as they appear in the main text, tables and figure legends and Methods and must follow the precise style of Nature Cell Biology references. References only cited in the Methods should be numbered consecutively following the last reference cited in the main text. References only associated with Supplementary Information (e.g. in supplementary legends) do not count toward the total reference limit and do not need to be cited in numerical continuity with references in the main text. Only published papers can be cited, and each publication cited should be included in the numbered reference list, which should include the manuscript titles. Footnotes are not permitted.

Methods should be written concisely, but should contain all elements necessary to allow interpretation and replication of the results. As a guideline, Methods sections typically do not exceed 3,000 words. The Methods should be divided into subsections listing reagents and techniques. When citing previous methods, accurate references should be provided and any alterations should be noted. Information must be provided about: antibody dilutions, company names, catalogue numbers and clone numbers for monoclonal antibodies; sequences of RNAi and cDNA probes/primers or company names and catalogue numbers if reagents are commercial; cell line names, sources and information on cell line identity and authentication. Animal studies and experiments involving human subjects must be reported in detail, identifying the committees approving the protocols. For studies involving human subjects/samples, a statement must be included confirming that informed consent was obtained. Statistical analyses and information on the reproducibility of experimental results should be provided in a section titled "Statistics and Reproducibility".

All Nature Cell Biology manuscripts submitted on or after March 21 2016, must include a Data availability statement as a separate section after Methods but before references, under the heading "Data Availability". For Springer Nature policies on data availability see <http://www.nature.com/authors/policies/availability.html>; for more information on this particular policy see <http://www.nature.com/authors/policies/data/data-availability-statements-data-citations.pdf>. The Data availability statement should include:

- Accession codes for primary datasets (generated during the study under consideration and designated as "primary accessions") and secondary datasets (published datasets reanalysed during the study under consideration, designated as "referenced accessions"). For primary accessions data

should be made public to coincide with publication of the manuscript. A list of data types for which submission to community-endorsed public repositories is mandated (including sequence, structure, microarray, deep sequencing data) can be found here <http://www.nature.com/authors/policies/availability.html#data>.

- Unique identifiers (accession codes, DOIs or other unique persistent identifier) and hyperlinks for datasets deposited in an approved repository, but for which data deposition is not mandated (see here for details <http://www.nature.com/sdata/data-policies/repositories>).
- At a minimum, please include a statement confirming that all relevant data are available from the authors, and/or are included with the manuscript (e.g. as source data or supplementary information), listing which data are included (e.g. by figure panels and data types) and mentioning any restrictions on availability.
- If a dataset has a Digital Object Identifier (DOI) as its unique identifier, we strongly encourage including this in the Reference list and citing the dataset in the Methods.

We recommend that you upload the step-by-step protocols used in this manuscript to the Protocol Exchange. More details can be found at www.nature.com/protocolexchange/about.

DISPLAY ITEMS – main display items are limited to 6-8 main figures and/or main tables. For Supplementary Information see below.

FIGURES – Colour figure publication costs \$395 per colour figure. All panels of a multi-panel figure must be logically connected and arranged as they would appear in the final version. Unnecessary figures and figure panels should be avoided (e.g. data presented in small tables could be stated briefly in the text instead).

All imaging data should be accompanied by scale bars, which should be defined in the legend. Cropped images of gels/blots are acceptable, but need to be accompanied by size markers, and to retain visible background signal within the linear range (i.e. should not be saturated). The boundaries of panels with low background have to be demarked with black lines. Splicing of panels should only be considered if unavoidable, and must be clearly marked on the figure, and noted in the legend with a statement on whether the samples were obtained and processed simultaneously. Quantitative comparisons between samples on different gels/blots are discouraged; if this is unavoidable, it has to be performed for samples derived from the same experiment with gels/blots were processed in parallel, which needs to be stated in the legend.

- For line art, graphs, charts and schematics we prefer Adobe Illustrator (.AI), Encapsulated PostScript (.EPS) or Portable Document Format (.PDF). Files should be saved or exported as such directly from the application in which they were made, to allow us to restyle them according to our journal house style.
- We accept PowerPoint (.PPT) files if they are fully editable. However, please refrain from adding PowerPoint graphical effects to objects, as this results in them outputting poor quality raster art. Text used for PowerPoint figures should be Helvetica (preferred) or Arial.
- We do not recommend using Adobe Photoshop for designing figures, but we can accept Photoshop generated (.PSD or .TIFF) files only if each element included in the figure (text, labels, pictures, graphs, arrows and scale bars) are on separate layers. All text should be editable in 'type layers' and line-art such as graphs and other simple schematics should be preserved and embedded within 'vector smart objects' - not flattened raster/bitmap graphics.
- Some programs can generate Postscript by 'printing to file' (found in the Print dialogue). If using an application not listed above, save the file in PostScript format or email our Art Editor, Allen Beattie for advice (a.beattie@nature.com).

Regardless of format, all figures must be vector graphic compatible files, not supplied in a flattened raster/bitmap graphics format, but should be fully editable, allowing us to highlight/copy/paste all text and move individual parts of the figures (i.e. arrows, lines, x and y axes, graphs, tick marks, scale bars etc). The only parts of the figure that should be in pixel raster/bitmap format are photographic images or 3D rendered graphics/complex technical illustrations.

SUPPLEMENTARY INFORMATION – Supplementary information is material directly relevant to the conclusion of a paper, but which cannot be included in the printed version in order to keep the manuscript concise and accessible to the general reader. Supplementary information is an integral part of a Nature Cell Biology publication and should be prepared and presented with as much care as the main display item, but it must not include non-essential data or text, which may be removed at the editor's discretion. All supplementary material is fully peer-reviewed and published online as part

of the HTML version of the manuscript. Supplementary Figures and Supplementary Notes are appended at the end of the main PDF of the published manuscript.

Unprocessed scans of all key data generated through electrophoretic separation techniques need to be presented in a supplementary figure that should be labeled and numbered as the final supplementary figure, and should be mentioned in every relevant figure legend. This figure does not count towards the total number of figures and is the only figure that can be displayed over multiple pages, but should be provided as a single file, in PDF or TIFF format. Data in this figure can be displayed in a relatively informal style, but size markers and the figures panels corresponding to the presented data must be indicated.

The total number of Supplementary Figures (not including the “unprocessed scans” Supplementary Figure) should not exceed the number of main display items (figures and/or tables (see our Guide to Authors and March 2012 editorial <http://www.nature.com/ncb/authors/submit/index.html#suppinfo>; <http://www.nature.com/ncb/journal/v14/n3/index.html#ed>). No restrictions apply to Supplementary Tables or Videos, but we advise authors to be selective in including supplemental data.

GUIDELINES FOR EXPERIMENTAL AND STATISTICAL REPORTING

REPORTING REQUIREMENTS – To improve the quality of methods and statistics reporting in our papers we have recently revised the reporting checklist we introduced in 2013. We are now asking all life sciences authors to complete two items: an Editorial Policy Checklist (found here <https://www.nature.com/authors/policies/Policy.pdf>) that verifies compliance with all required editorial policies and a Reporting Summary (found here <https://www.nature.com/authors/policies/ReportingSummary.pdf>) that collects information on experimental design and reagents. These documents are available to referees to aid the evaluation of the manuscript. Please note that these forms are dynamic ‘smart pdfs’ and must therefore be downloaded and completed in Adobe Reader. We will then flatten them for ease of use by the reviewers. If you would like to reference the guidance text as you complete the template, please access these flattened versions at <http://www.nature.com/authors/policies/availability.html>.

STATISTICS – Wherever statistics have been derived the legend needs to provide the n number (i.e. the sample size used to derive statistics) as a precise value (not a range), and define what this value represents. Error bars need to be defined in the legends (e.g. SD, SEM) together with a measure of centre (e.g. mean, median). Box plots need to be defined in terms of minima, maxima, centre, and percentiles. Ranges are more appropriate than standard errors for small data sets. Wherever

statistical significance has been derived, precise p values need to be provided and the statistical test used needs to be stated in the legend. Statistics such as error bars must not be derived from $n < 3$. For sample sizes of $n < 5$ please plot the individual data points rather than providing bar graphs. Deriving statistics from technical replicate samples, rather than biological replicates is strongly discouraged. Wherever statistical significance has been derived, precise p values need to be provided and the statistical test stated in the legend.

Author Rebuttal, first revision:

Response to reviewers (NCB-A49497A, Lindenhofer et al.)

We thank the reviewers for their repeated efforts and their thorough examination of our revised manuscript. In response to the reviewers' comments we have introduced several text changes and extensions to clarify points of criticism as well as reanalyzed some data for better representation.

Please find our comments as point-by point responses below in blue. Reviewer comments are in black.

Reviewers' Comments:

Reviewer #1:

Remarks to the Author:

Lindenhofer et al. have now submitted a comprehensive and satisfactory revised version of their work. This should be published with no further delay.

We thank the reviewer for the endorsement of our manuscript.

Main comments:

1. Investigation on cell type composition to elucidate the cell type nature of this dynamic growth or re-growth in organoids:

This has now been done in two modes, scRNAS-Seq and imaging. The finding by the former assay shows that clones are cell-type heterogeneous with or without relation to their size. As discussed in earlier correspondences it is expected that clones do not contain solely symmetrically dividing RG cells, but rather exhibit natural growth dynamics. While existence of such clones would have provided an unequivocally strong statistical proof for both the presence/type of symmetrically dividing cells and the robustness of the lineage tracing method, the live imaging provided now presents satisfactory indication that such cells do exist throughout organoid growth. Together with gene functional data, these findings now link fundamental principles together: organoids recapitulate important developmental features; to do this they rely on perpetuating progenitors throughout growth; these features can be detected by the new lineage tracing method. The work is well appreciated and now the main issue of the study is resolved.

We thank the reviewer for raising this critical point to show the presence of symmetrically dividing cells in 3D cerebral organoids and for acknowledging that we sufficiently addressed this. We agree that this added data improves the quality of the manuscript.

2: Distinguishing apart RG from IP cells in interpretations regarding symmetrical divisions shown in the lineage tracing:

It is very interesting and that S cells must be included in the model so that it is able to predict organoid growth dynamics. Likewise, it is interesting, though bit puzzling that IP cells not play

significant role in changing the division / growth landscape in organoids given their significant role in determining neuronal output number. It may be that organoids in these observed stages are less representing a strong IP growth wave. Nonetheless, the requirement of S cell state at any stage is new and sheds interesting light on in vivo development. The lack of IP dynamics in the prediction / measured models should be well discussed in the ms.

We agree with the reviewer, that the relationship between INPs and symmetrically dividing progenitors is an interesting subject. Both populations are present in developing organoids. We have now discussed this in the text as suggested.

3. Replenishment capacity in perturbed organoids:

The point we will take in both the authors' response and new provided data now showing that tissue is also largely qualitatively preserved after 6 weeks of perturbed development. This well supports the claim on replenishment capacity measured by the tracing method.

We thank the reviewer for raising this important point and acknowledging that we addressed this caveat.

4. 2D Vs 3D existence of dividing stem cells.

The clarification by authors that clonal output is restrictive in 2D is understood and it makes sense that diversity is restricted compared to 3D. However, it should be emphasized that as long as these 2D cultures are supplemented with mitogens it is obvious that the clonal output tested is of constantly supported proliferation rather than the natural transition between cell states present in 3D. These cultures can be grown for ages and certainly contain stem cells although not necessarily expanding, presumably just self-renewing asymmetrically. A way to detect clonal variability in these 2D cells is culturing them without any growth factors. But then, this will eventually drive complete exit from cell cycle and only neuronal clonal output, not stem cell output diversity, will be measured. No experiments are required here, just a visible clarification / discussion portion in text, as the 2D experiments here are an important reference for the 3D results and the use of this reference should be clear of doubts.

We thank the reviewer for his/her comment and incorporated the suggested text changes. We agree that both agents for proliferation and differentiation are necessary in culture media to drive heterogenous lineage size distributions. Only agents necessary for 2D hESCs that stimulate proliferation of the entire population results in homogenous lineage size distributions. If all cells were driven to complete exit and differentiation at the same time a similar picture would emerge. Differences that are observed in 2D neuronal and 3D cerebral organoid differentiation might be attributed to an extended presence of proliferative cells in neural rosettes in a tissue context. This is now resolved in the text in the corresponding sections.

Reviewer #2:

Remarks to the Author:

The revised manuscript by Lindenhofer has been submitted for evaluation. The authors have incorporated substantial new data to strengthen the conclusions of the paper, including live cell imaging from organoid slices. However, despite these improvements, the manuscript requires additional revisions to merit publication in a high-impact journal. I list my remaining concerns below:

1. The authors argue that “cerebral organoids have emerged as a faithful self-organizing model system to study human brain development in a tissue context”, but this is a deeply misleading statement. There are several studies noting that organoids fail to recapitulate cell types present in the human developing brain in vivo (Bhaduri et al. 2020 Nature and He et al. 2023 Biorxiv from Treutlein lab) due to mechanisms that include metabolic cell stress and other factors. In addition, the organoids used in this study do not incorporate non-neural cell types such as microglia or endothelial cells, which have been shown to have profound impact on brain development (Park et al. 2023 Nature). Similarly, contrary to the authors statements, cerebral organoids fail to recapitulate neuronal morphology and physiological maturation, because transplantation experiments conducted by the Pasca lab have shown only rudimentary cellular features can be derived in vitro (Revah et al. 2022 Nature). Given these limitations, it is clear that cerebral organoids do not faithfully recapitulate the developing brain and the authors should acknowledge these limitations throughout the manuscript.

We thank the reviewer for these comments and agree that it should be made clear that organoids are not 100% like the actual human brain. We changed the corresponding section in the manuscript. We also updated references including the ones suggested by the reviewer, some of which were only published after the initial submission of our manuscript.

However, we strongly disagree with the possible deeper notion of this reviewer that brain organoids are fundamentally flawed as a model to study brain development. It became abundantly clear in the past few years that brain organoids are a key tool to study important aspects of human brain development. This is evidenced by numerous recent reviews on the topic from leading groups (i.e. Arlotta and Gage labs, Arlotta and Gage, *Biol Psychiatry*, 2023; Pasca lab, Kelley and Pasca, *Cell*, 2022; our lab, Eichmüller et al, *Nature Rev. Neurol.*, 2022).

The reviewer correctly points out papers that describe weaknesses in specific organoid protocols. Specifically, the reviewer references a paper by the Kriegstein laboratory (Bhaduri et al., *Nature*, 2020). Among other things, this paper discussed metabolic stress in cortical organoids. Of note, however, even in this paper, the vast majority of organoid cells correspond well to in vivo cells. Moreover, the paper has since been put in perspective by newer and better culture protocols that have improved organoid quality since then (i.e. Arlotta lab, Uzquiano et al, *Cell*, 2022; Pasca lab, Trevino et al, *Science*, 2020, our lab, Vertesy et al, *EMBO J*, 2022). The other paper cited by the reviewer from the Treutlein lab (He et al, *BioRX*, 2023) is a beautiful recent compendium of available scRNA-seq data from multiple organoids, organoid protocols and labs. This is very informative in general and particularly for the issue at hand here. (1) The data from Bhaduri et al, *Nature*, 2020 is directly compared to more advanced organoid data (also compare Vertesy et al, *EMBO J*, 2022). (2) The authors show by comparison to in vivo data that indeed not all organoid protocols result in all cell types found in the brain. The authors point out erythrocytes and immune cells for example. This is expected as these cell types have a very different developmental origin. (3) He et al. also calculate a glycolysis score as a measure of metabolic activity in organoids compared to the in vivo reference. While they can identify a weak change in organoids compared to reference data, the authors conclude that "metabolic changes in organoids has only limited impact on the core molecular identity of neuronal cell types as defined by TF activity". Thus, while it is clear that organoids are not a 100% replica of the human brain, key aspects are well recapitulated without introducing significant artificial biases in the confounds of the respective protocol details.

We would also like to point out that brain organoids have since been used to model developmental diseases. These include for example Timothy syndrome (Birey et al, *Cell Stem Cell*, 2022), Miller-

Dieker Syndrome (Bershtyn et al, Cell Stem Cell, 2017), microcephaly (Lancaster et al., Nature, 2013), tuberous sclerosis (Eichmüller, Science, 2022) or autism (Paulsen et al., Nature, 2022; Jourdon et al., Nat. Neurosci., 2023; Li et al., Nature, 2023). The latter paper in particular is a good example for the ability of brain organoids to model normal and diseased brain phenotypes. Based on the observation that ARID1B malfunction leads to an increase in interneuron progenitors in brain organoids, it was predicted and confirmed in ARID1B patient MRIs that patient ganglionic eminences, the brain area harbouring ventral interneuron progenitors, were enlarged.

So, in our view, it is clear that brain organoids recapitulate a number of human brain developmental features, including correct cell type generation, the core of the current study. In fact, current organoid culture systems have reached a level of quality that they may serve as a predictive tool for brain diseases, even though it has to be pointed out that in agreement with the reviewer, brain organoids are not 100% like the actual human brain.

2. Introduction to the results section should incorporate longer description of the method. This is currently limited to stating that the barcodes are unique and retrovirally inserted, but provide virtually no description of the library complexity or steps taken to limit barcode collision events. I would highly recommend relocating some of the excellent material featured in Extended Data Figure 1 to main text Figure 1 and providing a little bit more description of the method in the results section. This will help readers who are novice to the nuances of the technology to better understand its implementation.

We thank the reviewer for suggesting these clarifications. We have now expanded the corresponding section in the main text, adding details about the technology and highlighting the importance of barcode variability and controls thereof. As these are technical details, important for a successful experiment, but not essential to the biological finding we opted to leave these controls in the Extended Data Figure 1; compare also library protocols in Esk et al., 2020. We believe that readers interested not just in the biological findings but applying our method for their biological questions will find the data in the Extended Data section.

3. One analysis that seems to be missing from this study is a deeper analysis of cell number per clone emerging within an organoid, and how that relates to estimates from primary cell lineage tracing experiments conducted by the Kriegstein lab using single cell clonal expansion experiments, which have shown that a single radial glia cell can produce 900 cells per clone (Pollen et al. 2015 Cell). On the other hand, single cell clonal lineage tracing in organoids has been previously conducted in cerebral organoids by the Treutlein lab (He et al. 2021 Nature Methods), while limiting the novelty claims made in this manuscript, has shown that average clone size in cerebral organoids is only 2 cells per clone. In the current manuscript, the authors briefly mention the He et al. study noting that the need for dissociation has likely led to an underestimate of clonal cell potential within an organoid, but reading the current manuscript it is almost impossible to grasp where the results fall in terms of the number of cells per clone derived from a single neural progenitor cell, which would be highly informative to studies of brain development.

We thank the reviewer for these comments. We want to point out that absolute clone sizes are not a topic in our paper, but rather the description of all clones making up organoids as a population and the discovery of long-lived symmetrically dividing cells in the largest of organoid clones. Therefore, we do not elaborate on this topic in the main text.

However, for interested readers the absolute sizes of clones measured in lineage traced scRNAseq is given in figures 3a, b (day 42 organoids) and Extended Data Figures 6h, i (day 18 organoids). Clone sizes span up to several hundred cells per clone. Comparing this number to the data by Pollen et al, Cell, 2015 - even though they are in a similar range - is very difficult as the experiments are very different. Lineage-traced scRNAseq in our manuscript versus single oRG cell clones in vitro in 2D.

Comparing our data to the data from He et al, Nature Methods, 2021 is also difficult, because the experimental setups are quite different. We labelled and isolated only a small subset of starter cells in an organoid for lineage traced scRNA-seq in an attempt to isolate these clones fully before performing scRNAseq (see corresponding experimental scheme and data in Extended Data Fig. 6a-c). Individual scRNA-seq preparations from 10x Genomics typically contain some 10000 cells per lane. He et al. used much more variable input populations and collected from full organoids

thereby subsampling individual clones as not all cells are sequenced from an organoid. One can speculate that this allowed them to assess possibly greater cell fate transition possibilities but may also have lead to an underestimation of individual clone sizes.

Clearly more work is required in the future, if absolute clone sizes are to be reliably investigated.

4. Addition of timelapse imaging data from cerebral organoid slices offers a notable improvement and validation to the experimental system implemented by the authors, but is not as extensive as the study by Baffet et al. (ref. 25). It is also of limited novelty as the only conclusion from this new data is the demonstration of the presence of self-renewing divisions. The authors use that to argue in a qualitative way that this observation validates their model, but this offers no quantitative validation. From simple theory not specific to the current method, presence of self-renewing progenitor cells would be expected. The added data thus provide limited value to the validation of the method, and falls short of providing quantitative insights, as the experiment appears to have been conducted only two times based on the data points indicated in Figure 3. The recommendation is therefore to conduct a third replication of the experiment, and to conduct a quantitative comparison of the proportions of self-renewing vs neurogenic divisions to determine how closely those estimates compare to the predictions from the in silico model. Such comparison would constitute a more informative validation of the in silico model.

We thank the reviewer for his / her thoughts on this point. However, we strongly disagree with the reviewer's notion that the presence of self-renewing cells would be expected at this stage. We describe symmetrically-dividing cells at least 28 days after the onset of neurogenesis in human cerebral organoids. In contrast, in mice all radial glia divide asymmetrically 3 days after the onset of neurogenesis (Gao et al, Cell, 2014; Llorca et al, ELife, 2019). To our knowledge, with the exception of the BioRX paper by the Baffet lab cited, this manuscript shows the prolonged existence of symmetrically dividing cells in a human brain development model for the first time. In the current manuscript we not only show that symmetrically dividing cells drive lineage size heterogeneity but also that their progeny output is plastic and tunable to tissue demands. The

existence of long-lived symmetrically dividing cells has been found and characterised by applying our modelling approaches to the novel whole organoid lineage tracing method introduced in this manuscript. We then wanted to validate this key finding and teamed up with the Baffet lab to demonstrate such cell divisions in our system. The confirmation of this key finding provides additional experimental proof to the model by validating the key insight generated. We would like to note that the reviewer acknowledged the significance of the manuscript in the first round of review.

The live cell imaging data presented in the manuscript were kindly provided through a collaboration with the Baffet lab with the intention to qualitatively show that symmetrically dividing cells are present in the organoids at this stage. For the experimental setup of the live cell imaging an organoid slice culture is required, in which proliferative zones are disrupted and exposed to mitogens in the culture media (see methods and Coquand et al, BioRX, 2022). This could have a strong impact on the relative division behaviour of progenitors and is in line with our chimeric organoid experiments where we show that stem cell output is tunable to tissue demands (Fig. 4 and 5; Extended Data Fig. 7-10). Additionally, the field of view of slice cultures under the microscope is limited. We concentrated on neural rosette progenitor zones to analyze the possible division behaviors of progenitor cells. Taken together, this makes data obtained from slice culture imaging difficult to incorporate in modelling approaches compared to data obtained from whole organoids (see below). We want to once again point out that our whole organoid lineage tracing method based on sequencing takes the entire organoid into account with no experimental procedures such as slicing and culturing the tissue, biases of field of views or single cell suspensions.

Regardless of the technical difficulties and selective nature of live cell imaging, we nonetheless took the reviewer's advice and modelled organoid growth based on the live imaging data (Reviewer Fig. 1.2 a, and c,). To compare division rates and lineage size distribution we used a simplified model only consisting of S and N cells. This is done because we show in Extended Data Figure 5 that the size of neuronal output does not greatly impact lineage size heterogeneity. Taking the rates from the live cell imaging and transferring this directly to the model we do observe a lineage size heterogeneity that resembles exponential growth. This does not correspond to

measured organoid sizes (Fig. 2a) at this stage where we know that overall growth of the organoid at this stage is not exponential. The reasons for this result might be they physical break-up of proliferative zones with combined mitogen exposure and the selection of fields of view of progenitor zones in an attempt to document different possible progenitor division modes – including symmetric divisions.

Reviewer Figure 1.2.

Reviewer Figure 1.2. Models with rates of live-cell imaging data. a,b, Model of lineage size distribution over time with rates from live cell imaging data (a) and with factor that removes progenitors from pool (b). c,d, Cell type composition resulting in lineage size model with rates from live cell imaging data (c) and with factor that removes progenitors from pool (d).

Running an alternative model and introducing a factor that removes progenitors from the pool by having them stop dividing can reproduce both the imaging division rates and observed lineage size heterogeneity (Reviewer Fig. 1.2 b, and d,). This progenitor removal would not be observed in live cell imaging based on retroviral labelling as it does not include a cell division. Consistent with what is reported in the manuscript the crucial factor for the observed lineage size heterogeneity is to have a near equal rate of symmetrical proliferation and differentiation. Although cell types and rates of individual cell divisions differ between different modelling approaches we can conclude a number of findings that are consistent to explain the observed heterogeneous lineage size distribution in cerebral organoids: (i) the rate of symmetrical proliferative divisions and differentiation is near equal, (ii) clones that contain symmetrically dividing cells are restricted to be a subset of the entire organoid (iii) clones that grow very large still contain symmetrically dividing cells (iv) clones progressively lose symmetrically dividing cells during organoid development – those clones remain smaller. This shows that rather than focusing on individual division rates of cells in those models these findings are the key messages from the presented modelling. Nonetheless, in the context of this review, it is informative and we thank the reviewer for prompting us to test this.

We believe that providing an additional replicate of the live cell imaging would not lead to any additional insights as the data provided shows very little heterogeneity and would delay finalization of the manuscript. However, each data point that was shown initially originated from a distinct differentiation experiment which contained two organoid replicates and multiple field of views for which imaging was performed. We now replotted the data to contain all individual organoid replicates confirming little heterogeneity and updated the respective information in the

figure legend. This analysis is in line with other experiments in the manuscript and underlines the high organoid reproducibility in the experiments.

5. Robustness of claims made in this study are limited by the utilization of a single embryonic stem cell line and this should be openly acknowledged as a limitation of this study in Discussion. This is important because in the introduction to the manuscript, the authors make strong statements about the robustness and versatility of the organoid platform and advantages of over methods such as DNA mutation sequencing, but themselves do not fully take advantage of the potential versatility of the method to comment on the robustness of the findings outside of the one embryonic stem cell line. This should be acknowledged.

We thank the reviewer for this comment and agree that the use of only H9 hESCs is a limitation of the study. This is now included in the discussion section.

We would like to point out that in the context of the H9 background, several independent lines have been used i.e. the GFP and RFP-expressing lines.

We do not see our method as superior to DNA mutation sequencing as insinuated by the reviewer. In fact, we see them as complimentary as both methods have different strengths with regard to throughput and complexity of obtained results. See corresponding section of main text:

“In human brain only limited data is available of clonal output heterogeneity, suggesting potentially expanded clone size dispersion compared to mouse^{3–6}. These pioneering studies rely on whole-genome sequencing of individual neurons from post mortem brains and back-tracking lineage relationships based on single nucleotide polymorphisms accumulated during a donor’s brain development. These approaches are time and costs intensive and have so far been limited to the study of several thousand cells.”

6. On a conceptual level, the authors do not provide substantial novel insights into human brain development. The effects of Pax 6 loss of function have been studied for decades and this study does not extend these findings in any meaningful way beyond showing that they can be conducted using brain organoids. In fact, if anything, the results in this study do not agree with findings from in vivo experiments using chimeric Pax6 +/+ <-> Pax6 -/- animals, where Pax6-/- contributed to the cortical neuroepithelium (Quinn et al. 2007 Developmental Biology), whereas in their experiments using organoids, the authors report no contribution of Pax6 KO cells to neural rosettes. This discrepancy reflects either a fundamental limitation of the model system, contrary to the main claim of the paper, or cross-species difference. Either outcome is informative, and highly addressable experimentally, and resolution of this discrepancy, which after all reflects one of the main findings and points of novelty of the study, would be important for a high-quality paper.

PAX6 is a transcription factor that is conserved between primate and non-primate species but there are major differences in expression patterns and functionality (reviewed for example in Manuel et al., Front Cell Neurosci, 2015). This is also highlighted by a study from Zhang et al., Cell Stem Cell, 2010, suggesting that knockdown of PAX6 blocks neuroectoderm specification in hESCs but not in mESCs. In addition, PAX6 expression levels differ depending on brain region and have an impact on patterning. This motivated us to use PAX6-KO cells to test wildtype cell behaviour against loss of function cells. The focus of this manuscript was not PAX6 functionality in particular, but rather to explore a systemic response of lineages on a neuroectoderm specification defect in chimeric organoids. This has not been explored in any other study to our knowledge.

7. On a technical side, the manuscript also lacks novelty with respect to demonstrating the ability of their model to assess the consequences of genetic mutations using organoids. A recent study from the Treutlein lab has shown that introducing loss of function mutations in another patterning gene can be conducted successfully using organoids and recapitulates known biology of this gene. See Fleck et al. 2022 Nature for details. The authors should at least cite this highly relevant paper.

We thank the reviewer for this comment. We would like to emphasize again that our study concerns the behaviour of normal wildtype cells and their adjustment to surrounding tissue being disturbed.

This contrasts the study by the Treutlein lab cited (Fleck et al, Nature, 2022), which studies individual genes' functions to understand GRNs in development and complements our own lab's collaborative study with the Treutlein lab on autism candidate gene function (Li et al, Nature, 2023).

Fleck et al. also perturb PAX6 in their mosaic pooled perturbation screen. Of note is that their screening is done in mosaic organoids which contain many different perturbed genes and that not all cells in their setting are expected to receive a complete loss of function mutation (Inferred KO probabilities for PAX6 are shown). Although they do not highlight PAX6 in the main text, they do see an enrichment of PAX6 targeted cells in the ventral telencephalon compared to the dorsal one, consistent with the patterning impact of PAX6 expression.

At the time of the initial submission of this manuscript the paper from Fleck et al. was not published yet. We cite this study now in the respective section of the manuscript.

Reviewer #3:

Remarks to the Author:

In this revision, Dominik Lindenhofer et al. evaluated clonal growth behavior of the whole cerebral organoids tissue dynamically through lineage tracing. The data suggest that the presence of consistently symmetrical dividing cells during the growth and development of human brain organoids may contribute to higher heterogeneity compared to 2D cells or mouse brains. And the growth of neural stem cells in organoids is adaptable to the needs of the whole tissue and has plasticity. The revised paper has addressed major concerns that raised by reviewers, and the manuscript has been better improved. The method that established in this manuscript is well written and would benefit this field. One minor point is the author could describe the result

more clearly. For example, in the paragraph “Whole-tissue barcoded lineage tracing”, in the sentence of “First, in-silico subsampling of lineages from the whole organoid lineage tracing ...”, the meaning of Extended Data Figure 2a and the ratio of 1%-2% could be explain more clearly.

We thank the reviewer for his / her support of our work. We have taken the reviewer comment into account and clarified the passage in question in the main text. Again, thank you for your interest in this manuscript.

Decision Letter, second revision:

1st February 2024

Dear Juergen,

Thank you for submitting your revised manuscript "Cerebral organoids display dynamic clonal growth with lineage replenishment" (NCB-A49497C). It has now been seen by the original referee #2 and their comments are below. The reviewer finds that the paper has improved in revision, and therefore we'll be happy in principle to publish it in Nature Cell Biology, pending minor revisions to comply with our editorial and formatting guidelines.

If the current version of your manuscript is in a PDF format, please email us a copy of the file in an editable format (Microsoft Word or LaTeX)-- we cannot proceed with PDFs at this stage.

Thank you again for your interest in Nature Cell Biology. Please do not hesitate to contact me if you have any questions.

Best wishes,
Stelios

Stylianos Lefkopoulos, PhD
He/him/his
Senior Editor, Nature Cell Biology
Springer Nature
Heidelberger Platz 3, 14197 Berlin, Germany

E-mail: stylianos.lefkopoulos@springernature.com
Twitter: @s_lefkopoulos

LinkedIn: [linkedin.com/in/stylianos-lefkopoulos-81b007a0](https://www.linkedin.com/in/stylianos-lefkopoulos-81b007a0)

Reviewer #2 (Remarks to the Author):

In their revised submission, the authors have addressed my concerns, including acknowledgment of study limitations, which will be of service to the community.

In its current form, I am supportive of publication.

Decision Letter, final checks:

Our ref: NCB-A49497C

20th February 2024

Dear Dr. Knoblich,

Thank you for your patience as we've prepared the guidelines for final submission of your Nature Cell Biology manuscript, "Cerebral organoids display dynamic clonal growth with lineage replenishment" (NCB-A49497C). Our sincere apologies for the delay in sending over the following checklist. Please carefully follow the step-by-step instructions provided in the attached file, and add a response in each row of the table to indicate the changes that you have made. Ensuring that each point is addressed will help to ensure that your revised manuscript can be swiftly handed over to our production team.

In recognition of the time and expertise our reviewers provide to Nature Cell Biology's editorial process, we would like to formally acknowledge their contribution to the external peer review of your manuscript entitled "Cerebral organoids display dynamic clonal growth with lineage replenishment". For those reviewers who give their assent, we will be publishing their names alongside the published article.

Nature Cell Biology offers a Transparent Peer Review option for new original research manuscripts submitted after December 1st, 2019. As part of this initiative, we encourage our authors to support increased transparency into the peer review process by agreeing to have the reviewer comments, author rebuttal letters, and editorial decision letters published as a Supplementary item. When you

submit your final files please clearly state in your cover letter whether or not you would like to participate in this initiative. Please note that failure to state your preference will result in delays in accepting your manuscript for publication.

Cover suggestions

COVER ARTWORK: We welcome submissions of artwork for consideration for our cover. For more information, please see our guide for cover artwork.

Nature Cell Biology has now transitioned to a unified Rights Collection system which will allow our Author Services team to quickly and easily collect the rights and permissions required to publish your work. Approximately 10 days after your paper is formally accepted, you will receive an email in providing you with a link to complete the grant of rights. If your paper is eligible for Open Access, our Author Services team will also be in touch regarding any additional information that may be required to arrange payment for your article.

Please note that *Nature Cell Biology* is a Transformative Journal (TJ). Authors may publish their research with us through the traditional subscription access route or make their paper immediately open access through payment of an article-processing charge (APC). Authors will not be required to make a final decision about access to their article until it has been accepted. Find out more about Transformative Journals

Please use the following link for uploading these materials, ideally within one week:
[Redacted]

Best regards,

Kendra Donahue
Staff
Nature Cell Biology

On behalf of

Stylianos Lefkopoulos, PhD
He/him/his
Senior Editor, Nature Cell Biology
Springer Nature
Heidelberger Platz 3, 14197 Berlin, Germany

E-mail: stylianos.lefkopoulos@springernature.com
Twitter: @s_lefkopoulos
LinkedIn: [linkedin.com/in/stylianos-lefkopoulos-81b007a0](https://www.linkedin.com/in/stylianos-lefkopoulos-81b007a0)

Reviewer #2:

Remarks to the Author:

In their revised submission, the authors have addressed my concerns, including acknowledgment of study limitations, which will be of service to the community.

In its current form, I am supportive of publication.

Final Decision Letter:

Dear Juergen,

I am pleased to inform you that your manuscript, "Cerebral organoids display dynamic clonal growth and tunable tissue replenishment", has now been accepted for publication in Nature Cell Biology. Congratulations!

Please note that *Nature Cell Biology* is a Transformative Journal (TJ). Authors may publish their research with us through the traditional subscription access route or make their paper immediately open access through payment of an article-processing charge (APC). Authors will not be required to make a final decision about access to their article until it has been accepted. Find out more about Transformative Journals

If your paper includes color figures, please be aware that in order to help cover some of the additional cost of four-color reproduction, Nature Portfolio charges our authors a fee for the printing of their color

figures. Please contact our offices for exact pricing and details.

If you have not already done so, we strongly recommend that you upload the step-by-step protocols used in this manuscript to the Protocol Exchange (www.nature.com/protocolexchange), an open online resource established by Nature Protocols that allows researchers to share their detailed experimental know-how. All uploaded protocols are made freely available, assigned DOIs for ease of citation and are fully searchable through nature.com. Protocols and Nature Portfolio journal papers in which they are used can be linked to one another, and this link is clearly and prominently visible in the online versions of both papers. Authors who performed the specific experiments can act as primary authors for the Protocol as they will be best placed to share the methodology details, but the Corresponding Author of the present research paper should be included as one of the authors. By uploading your Protocols to Protocol Exchange, you are enabling researchers to more readily reproduce or adapt the methodology you use, as well as increasing the visibility of your protocols and papers. You can also establish a dedicated page to collect your lab Protocols. Further information can be found at www.nature.com/protocolexchange/about

With kind regards,
Stelios

Stylianos Lefkopoulos, PhD
He/him/his
Senior Editor, Nature Cell Biology
Springer Nature
Heidelberger Platz 3, 14197 Berlin, Germany

E-mail: stylianos.lefkopoulos@springernature.com
Twitter: @s_lefkopoulos
LinkedIn: [linkedin.com/in/stylianos-lefkopoulos-81b007a0](https://www.linkedin.com/in/stylianos-lefkopoulos-81b007a0)

** Visit the Springer Nature Editorial and Publishing website at www.springernature.com/editorial-and-publishing-jobs for more information about our career opportunities. If you have any questions please click here.**